# Emergence of large-scale cell death through ferroptotic trigger waves

Hannah K. C. Co[1,2,5], Chia-Chou Wu[2,3,5], Yi-Chen Lee[2] & Sheng-hong Chen[1,2,3,4 ✉]

Large-scale cell death is commonly observed during organismal development and in human pathologies[1–5]. These cell death events extend over great distances to eliminate large populations of cells, raising the question of how cell death can be coordinated in space and time. One mechanism that enables long-range signal transmission is trigger waves[6], but how this mechanism might be used for death events in cell populations remains unclear. Here we demonstrate that ferroptosis, an iron- and lipid-peroxidation-dependent form of cell death, can propagate across human cells over long distances (≥5 mm) at constant speeds (around 5.5 µm min⁻¹) through trigger waves of reactive oxygen species (ROS). Chemical and genetic perturbations indicate a primary role of ROS feedback loops (Fenton reaction, NADPH oxidase signalling and glutathione synthesis) in controlling the progression of ferroptotic trigger waves. We show that introducing ferroptotic stress through suppression of cystine uptake activates these ROS feedback loops, converting cellular redox systems from being monostable to being bistable and thereby priming cell populations to become bistable media over which ROS propagate. Furthermore, we demonstrate that ferroptosis and its propagation accompany the massive, yet spatially restricted, cell death events during muscle remodelling of the embryonic avian limb, substantiating its use as a tissue-sculpting strategy during embryogenesis. Our findings highlight the role of ferroptosis in coordinating global cell death events, providing a paradigm for investigating large-scale cell death in embryonic development and human pathologies.

Large-scale cell death is a common occurrence in embryonic development and diseases. During embryogenesis, massive cell death acts as a tissue-sculpting process to shape cavities and tubules for organ formation[1,7,8]. Under pathological conditions, excessive tissue damage frequently occurs in ischaemia–reperfusion and degenerative diseases[3–5,9]. These cell death events usually occur contiguously across the tissue over considerable distances, eliminating large cell populations. Although it is known that a cell death event can affect neighbouring cells through diffusive signals such as cytotoxic molecules (for example, calcium and ROS)[10,11] or signalling factors (including TNF)[12], this bystander effect is quickly abrogated in space and time, curtailing cell death extent (to around 20 µm or 2–3 cells). Thus, it is unclear how a localized and short-lived death signal overcomes the spatial limitations of simple diffusion to cause a global cell death event.

Collective cell death requires robust signal transmission, which is made possible by the mechanism of trigger waves. In contrast to simple diffusion, whereby signals dissipate quickly in space, trigger waves are self-regenerating chemical fronts that propagate over long distances without compromising speed or amplitude[6,13]. Trigger waves are ubiquitous in nature, being responsible for neuron firing[14], mitotic and apoptotic waves in *Xenopus* eggs[15,16], and infectious disease epidemics[17]. Although they differ in molecular details, all of these examples rely on excitable or bistable media over which signals are regenerated and transmitted[6,18]. Considering the prevalence of large-scale cell death, we postulated that human cells could be primed to become bistable cell populations that permit the propagation of death signals via trigger waves.

## Propagative nature of ferroptosis

Ferroptosis, a form of cell death through iron-catalysed lipid peroxidation, has recently been shown to exhibit profound bystander effects that allow cell death to spread over relatively long distances (up to 600 µm)[19]. In support of that observation, we found that ferroptosis, caused by either cystine deprivation or treatment with ferroptosis inducers (such as erastin (an inhibitor of the cystine–glutamate antiporter xCT) and RSL3 (an inhibitor of glutathione peroxidase 4; GPX4)), spreads across human retinal pigment epithelial (RPE) cells as a series of spatially coupled death events (Extended Data Fig. 1a,b,d,f and Supplementary Video 1). Consistently, we found that ferroptotic death kinetics presented high spatiotemporal order, as reflected by the lower entropies of cell death vectors compared with those of apoptotic and calcium-induced death (Extended Data Fig. 1c–j). These well-ordered ferroptotic death kinetics were also observed after xCT suppression in 16 additional human cell lines derived from 12 tissue types (Extended Data Fig. 1k,m and Supplementary Video 1). By contrast, these ordered

[1]Molecular and Cell Biology, Taiwan International Graduate Program, Academia Sinica and Graduate Institute of Life Science, National Defense Medical Center, Taipei, Taiwan. [2]Laboratory for Cell Dynamics, Institute of Molecular Biology, Academia Sinica, Taipei, Taiwan. [3]National Center for Theoretical Sciences, Physics Division, Taipei, Taiwan. [4]Genome and Systems Biology Degree Program, Academia Sinica and National Taiwan University, Taipei, Taiwan. [5]These authors contributed equally: Hannah K. C. Co, Chia-Chou Wu. ✉e-mail: shengchen@gate.sinica.edu.tw

death kinetics were absent after inhibition of GPX4 in some of the cell lines that we tested (Extended Data Fig. 1l,m). These results support the generality of the propagative nature of ferroptosis when induced by xCT suppression, but not by GPX4 inhibition. This difference may be attributable to the involvement of distinct regulatory pathways, depending on how ferroptosis is induced[20]. Nevertheless, our results, along with the large-scale ferroptosis observed under pathological conditions[3,21,22], prompted us to examine the possibility that large-scale cell death may be coordinated through ferroptosis.

## Ferroptosis propagates as trigger waves

Despite previous observations of the wave-like spread of ferroptosis, the challenge of obtaining time-resolved, long-distance ferroptosis quantifications has precluded investigations of its regulatory principles. For example, an assay that monitors cell death propagation over distances of millimetres would enable definitive distinction between simple diffusion-driven ferroptotic spread, which would slow down in space, and ferroptotic trigger waves, which would maintain a constant speed. To develop such an assay, we used RPE cells, which are sensitive to blue-light irradiation[23] and exhibit excessive ferroptosis during age-related retinal degeneration[24]. After erastin treatment, ferroptosis initiated and propagated across RPE cells (Extended Data Fig. 1d). However, annihilation of the ferroptotic waves from multiple cell-death-initiating sites impeded their long-distance measurement (Extended Data Fig. 2a). We found initiation of ferroptosis to be a random process, with the frequency and time interval between consecutive initiation events following Poisson and geometric distributions, respectively (Methods and Extended Data Fig. 2b–d). Consistently, cellular ROS and iron levels adopted a continuous probability distribution (Extended Data Fig. 2e,f), probably owing to the intrinsic stochasticity of the redox system. Thus, to reduce the number of ferroptotic initiations, we modulated the cell growth medium (Methods and Extended Data Fig. 2g,h) and then used RPE cell photosensitivity to control the timing and location of ferroptosis initiations through blue-light irradiation (432 nm) to introduce exogenous ROS (Extended Data Fig. 3a). Blue-light exposure elevated cellular ROS and prompted subsequent cell death of erastin-treated RPE cells (Extended Data Fig. 3b), similar to the effect of adding exogenous $H_2O_2$ (Extended Data Fig. 3c). Importantly, this light-induced cell death proved ferroptosis specific, as ferroptosis inhibitors (Fer-1, DFO and Lip-1) suppressed it, whereas inhibitors of apoptosis and necroptosis did not (Extended Data Fig. 3d).

In response to blue-light irradiation of erastin-treated cells, ferroptotic cell death initiated from the light-exposed area and propagated at a constant speed over a distance of several millimetres (in this experiment, at 5.48 µm min⁻¹ across 5 mm; Fig. 1a–d and Supplementary Video 2). Three independent experiments revealed that ferroptosis propagated at a constant speed of $5.52 \pm 0.09$ µm min⁻¹ (mean ± s.d.) over distances of ≥5 mm. Compared with the diffusive spread of either calcium[25] or a small protein[26], ferroptosis propagated linearly over time (Fig. 1e), supporting the idea that ferroptosis propagates through trigger waves and not through simple diffusion. Notably, the speed of ferroptotic propagation did not change according to photoinduction light intensity (Extended Data Fig. 3e–j), indicating that the cellular state of the population, but not the intensity of the initiating signal, is the determinant of wave speed. Again, this propagating cell death was ferroptosis specific as only ferrostatin-1 halted its propagation, whereas inhibitors of apoptosis and necroptosis did not (Extended Data Fig. 3k,l).

One hallmark of ferroptosis is the elevation of lipid peroxidation[22]. Using a lipid peroxidation probe, we observed wavefronts of lipid peroxidation that preceded cell death (Fig. 1f and 1g (magnified image) and Supplementary Video 3). These lipid peroxidation wavefronts propagated at a speed similar to ferroptotic cell death (Fig. 1h,i; in

this experiment, at 6.05 µm min⁻¹ ($5.83 \pm 0.40$ µm min⁻¹, mean ± s.d.) for three biological repeats). Similarly, cellular ROS—detected using a general ROS probe for hydroxyl radicals (·OH), superoxides ($O_2^-$) and hydrogen peroxides ($H_2O_2$)—were also observed as wavefronts (Extended Data Fig. 4a–d and Supplementary Video 4), again propagating at a speed similar to that of ferroptotic waves (in this experiment, at 5.29 µm min⁻¹ ($5.46 \pm 0.25$ µm min⁻¹, mean ± s.d.) for three biological repeats). Addition of various ROS scavengers halted the propagating cell death (Extended Data Fig. 4e), suggesting that multiple ROS can collectively contribute to the signalling wavefronts that drive ferroptosis propagation.

## Diffusion for spatial coupling

The formation of self-regenerating ROS wavefronts requires two critical components: (1) a spatial coupling mechanism for intercellular ROS transmission; and (2) an intracellular ROS amplification mechanism (for example, ROS bistability). One possible spatial coupling mechanism is the diffusion of ROS or an ROS-inducing molecule that is independent of direct cell–cell contact. To test that possibility, we created physical gaps (35–380 µm) between a wave-initiated region and a non-initiated region. Ferroptosis propagated across small gap sizes of 2–8 cells (35–118 µm in width) with no sign of stalling (Fig. 2), supporting the idea that direct cell–cell contact is not essential for ferroptosis propagation. When the gap width was large (>200 µm), ferroptosis propagation consistently ceased (Fig. 2). For medium-width gaps (120–200 µm), ferroptosis spread across these gaps in 50% of cases (that is, 5 out of 10 waves halted; Fig. 2b). In the cases in which waves halted, the cells at the edge of the non-initiated region exhibited a gradual increase in ROS (magnified views of the traces for the 156 µm and 224 µm gaps are shown in Fig. 2a). This ROS accumulation is probably caused by a diffusive molecule from ferroptotic cells of the wave-initiating region, as creating gaps alone does not increase ROS levels (Extended Data Fig. 5a). To characterize this diffusive molecule, we collected the erastin-free conditioned medium containing the secreted molecules from ferroptotic cells (Methods). Cells exposed to this conditioned medium underwent ferroptosis, indicative of the presence of the diffusive molecule secreted by ferroptotic cells (Extended Data Fig. 5b). However, when the conditioned medium was pretreated with antioxidants to scavenge lipid peroxides (Trolox, Fer-1), hydrogen peroxide, superoxide and hydroxyl radicals (TEMPO, Tiron), the cell death caused by the conditioned medium was greatly suppressed, indicating that this diffusive molecule may be a type of ROS whose activity is suppressed by ROS scavengers. One possible ROS candidate is $H_2O_2$, which is known to be long lived and freely diffusive. To test the involvement of $H_2O_2$, we subjected the conditioned medium to centrifugal filtration. In contrast to the eluate of culture medium containing exogenously added $H_2O_2$, which caused cell death, the eluate of the centrifuged conditioned medium did not induce cell death, implying that the diffusive molecule is unlikely to be $H_2O_2$ (Extended Data Fig. 5c). It is therefore possible that the spatial coupling mechanism underlying ferroptosis propagation is the diffusion of ROS molecules, such as peroxidized lipids[27] and/or their byproducts. These lipid peroxidation byproducts, such as lipid-derived electrophiles with a peroxide group, can be released from ferroptotic cells[28], and they are sufficiently stable chemically to freely diffuse[29]. Moreover, other signalling molecules (such as proteins, extracellular vesicles and/or their cargoes)[30,31] with oxidative capabilities may also mediate the spread of ferroptosis. Further studies are required to determine the identity of these diffusive ROS molecules.

## ROS amplification through feedback loops

For ferroptosis to propagate as trigger waves, intercellular diffusive ROS need to activate a ROS-amplification mechanism in neighbouring cells, in turn leading to their further production of diffusive ROS

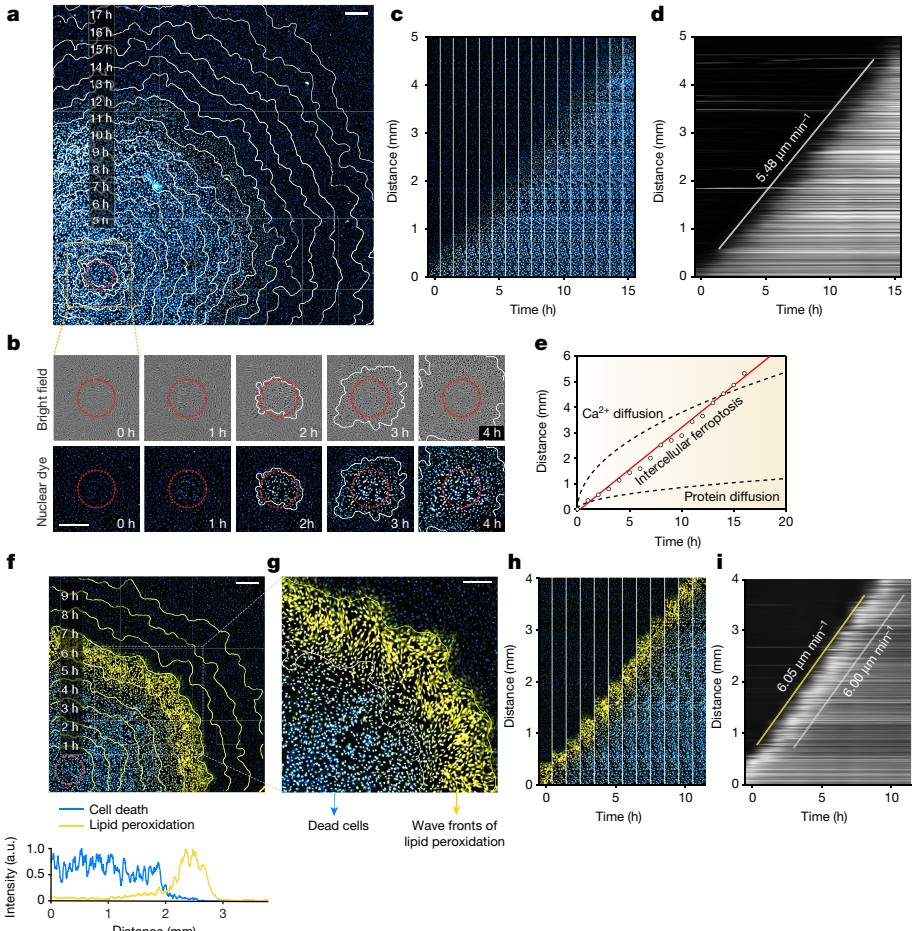

**Fig. 1 | Ferroptosis propagates across RPE-1 cells at a constant speed through lipid peroxidation wavefronts. a,b,** Ferroptosis initiates from the photoinduction area (red circle) and propagates across erastin-treated cells over 5 mm for 18 h. The contours (white outline) represent the border of cell death at specific timepoints. **a,** Nuclear dye fluorescence image (11 h after photoinduction) overlaid with contours 2–18 h after photoinduction. **b,** Time-lapse images of cell death after photoinduction, showing magnified views of the orange box in **a**. Cell death is indicated by cell rupture (bright field) and increased nuclear dye fluorescence (cyan to white). **c,d,** Data derived from **a**. **c,** Time-lapse image array of ferroptosis propagation. **d,** Kymograph for ferroptosis propagation. The white line is a linear least-squares fit to the nuclear dye fluorescent fronts, where its slope represents the speed of propagation. **e,** Comparison of travelling distances over time among ferroptosis propagation and diffusive molecules (Methods). **f,g,** Merged images of lipid peroxidation (yellow) and nuclear dye fluorescence (cyan). Lipid peroxidation was monitored using C11-BODIPY[581/591]. The yellow contours represent the border of lipid peroxidation at specific timepoints. **f,** Image (7 h after photoinduction) overlaid with contours 1–10 h after photoinduction (top). Bottom, fluorescence intensities of cell death and lipid peroxidation quantified across the bottom region of the image. **g,** Magnified view of the box in **f**. **h,** Time-lapse image array of lipid peroxidation and cell death in **f**. **i,** Kymograph for lipid peroxidation propagation in **f**. The slopes of the yellow and white lines represent the speeds of lipid peroxidation and cell death propagation, respectively. Data shown are representative of three biological repeats. Scale bars, 400 μm (**a, b** and **f**) and 250 μm (**g**).

and ferroptotic cell death. At least three ROS feedback loops could potentially operate to amplify ROS in the ferroptosis network: a glutathione (GSH)-mediated double-negative-feedback loop and the two positive-feedback loops of the Fenton reaction[32] and NADPH oxidase (NOX) signalling[33–35] (Fig. 3a). To examine the roles of these feedback loops in regulating ferroptosis propagation, we modulated their strengths by chemical perturbation (Fig. 3b).

The Fenton reaction is driven by cellular labile iron. It converts $H_2O_2$ to ·OH—a highly reactive ROS that elicits autocatalytic lipid peroxidation[32]. To examine the role of iron in ferroptosis propagation, we perturbed its level using the iron chelator deferoxamine (DFO) and the iron supplement ferric citrate (FC). After DFO treatment (80 μM), we observed that the speed of ferroptosis propagation diminished from 5.28 μm min⁻¹ to 2.33 μm min⁻¹ (Fig. 3c,d and Supplementary Video 6). This deceleration of ferroptosis propagation proved to be dose dependent, with propagation being completely halted at a DFO concentration of 160 μM (Fig. 3j). By contrast, supplying free iron

through FC accelerated ferroptosis propagation from 5.04 μm min⁻¹ to 9.40 μm min⁻¹ (Fig. 3e,f and Supplementary Video 7). Similarly, we observed dose dependency for ferroptotic wave speed (Fig. 3k). Using a labile-iron probe, we observed elevated iron levels at the ferroptosis wavefronts, supporting a critical role for iron in regulating ferroptosis propagation (Extended Data Fig. 6a,b).

NOXs are a major class of enzymes that generate cellular ROS ($O_2^-$ and $H_2O_2$). NOX-generated ROS inhibit tyrosine kinase phosphatases, resulting in activation of tyrosine kinases[36]. In turn, active tyrosine kinases and their downstream effectors (such as PI3K) can activate NOXs to further promote ROS production[33–35], thereby forming a positive-feedback loop (Fig. 3a). As *NOX4* is the predominant isoform expressed in RPE cells (Extended Data Fig. 7a), it is likely that this positive-feedback loop depends primarily on NOX4. Accordingly, we applied inhibitors of NOX1/NOX4 (GKT137831), PI3K (LY294002) and tyrosine kinases (dasatinib) to RPE cells (Fig. 3b). These small-molecule inhibitors are target specific and do not exhibit antioxidant activity (Extended Data Fig. 7b).

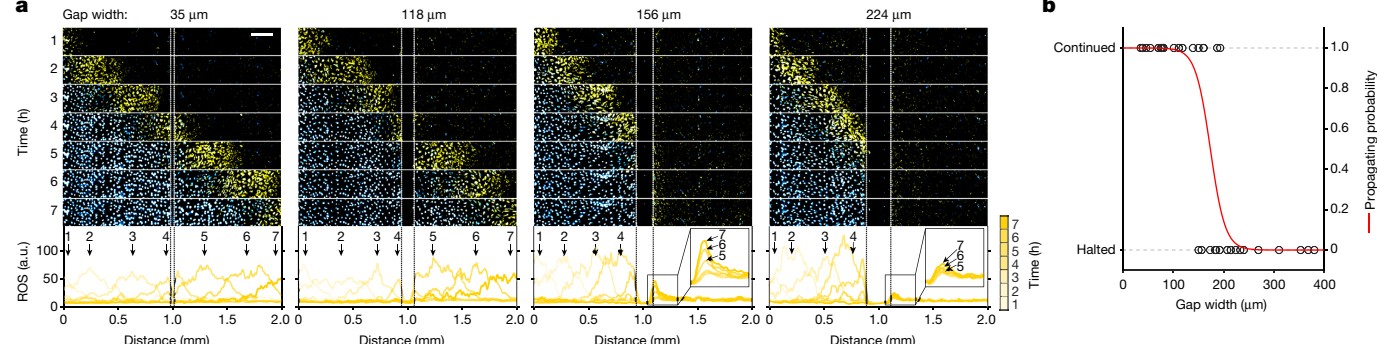

**Fig. 2 | Diffusion of ROS signal as the coupling mechanism for ferroptotic trigger waves. a**, Gaps were created between a wave-initiated region (left) and a non-initiated region (right). Time-lapse image sequence of cell death (cyan) and ROS (yellow) propagation across a gap. ROS was monitored in erastin-treated cells using CellROX dye. The mean fluorescence intensity of ROS was quantified along the 2 mm distance at specific timepoints. Scale bar, 200 μm. **b**, The probability of a wave passing through different gap widths (35–380 μm).

The propagating probability was obtained by fitting the data (from 38 wells) to a logistic model, $p(x) = \frac{1}{1 + e^{-(\beta_0 + \beta_1 x)}}$, where $p$ is the propagating probability and $x$ is the gap width. Gaps were created by scratching the bottom of the plate with needles of different tip sizes (20–400 μm) after wave initiation. The time-lapse video for this experiment is shown in Supplementary Video 5. Data are representative of three biological repeats.

All three inhibitors slowed down ferroptosis propagation (Fig. 3g–i) and presented dose dependency (Fig. 3l–n).

In addition to chemical perturbations, we genetically modulated the strength of NOX signalling by overexpressing *ERK2*, encoding an intermediate signalling protein in the RTK–ERK–NOX cascade[37] (Extended Data Fig. 8a). RPE cells overexpressing *ERK2* presented greater ERK2 activity and NOX activity (Extended Data Fig. 8b,c). Accordingly, ferroptotic waves propagated at a higher speed in *ERK2*-overexpressing cells (Extended Data Fig. 8d,e and Supplementary Video 8), supporting a primary role for the NOX-mediated feedback loop in wave propagation.

Activation of these ROS feedback loops not only enhanced ferroptosis propagation, but also induced it in cell populations that were otherwise resistant to ferroptosis. For example, one factor that causes ferroptosis resistance is high confluency, which can downregulate iron uptake[38], thereby probably suppressing the Fenton-mediated ROS feedback loop. Using lung carcinoma epithelial A549 cells and bone osteosarcoma U-2 OS cells as examples, we found that high cell densities resulted in ferroptosis resistance. By supplementing iron in combination with erastin treatment, we found that ferroptosis initiated and propagated across both these cell populations (Extended Data Fig. 6c,d), further highlighting the critical role of activating ROS feedback loops in ferroptosis propagation. Moreover, in addition to the regulators involved in amplifying ROS, other well-known ferroptosis regulators (such as ferroptosis suppressor protein 1; FSP1)[39] also contributed to ferroptosis propagation, with ferroptotic wave speed increasing after inhibition of FSP1 (Extended Data Fig. 9). Together, these results indicate that molecular regulators (amplifiers or suppressors) of ROS can govern the occurrence and progressivity of ferroptotic waves.

## ROS bistability on ferroptotic stress

At the core of the ferroptosis network is the GSH–ROS double-negative-feedback loop (Fig. 3a). To maintain ROS at physiological levels, cells depend heavily on de novo GSH synthesis through nutrient catabolism (for example, of glucose and cystine)[22,40]. Our previous study revealed that a glucose-deprivation-induced decline in GSH results in ROS bistable switches[41]. Similarly, suppression of cystine uptake may cause ROS bistability, allowing ferroptosis to propagate as trigger waves. To gain a quantitative understanding of this process, we built a mathematical model for ROS trigger waves incorporating (1) cell-intrinsic ROS feedback loops in the ferroptosis network; and (2) simple diffusion as the coupling mechanism between cells (Extended Data Fig. 10a).

Our simulations show that an increased erastin concentration can lead to a bifurcation in the ROS steady state, that is, switching from a monostable (<0.39 μM erastin) to a bistable regime (0.39–11.7 μM erastin) in which two ROS stable steady states (lower and upper) coexist (Fig. 4a). As expected from a bistable system[6], an activation threshold (that is, unstable steady state (USS)) separates the two steady states such that ROS levels above this threshold are amplified to the upper steady state. ROS usually propagate as trigger waves within this bistable regime (Extended Data Fig. 10b–d). When ROS levels are elevated sufficiently to surpass the activation threshold (Fig. 4a (blue to yellow)), either by exposure to blue light or by diffusive ROS from neighbouring ferroptotic cells, ROS feedback loops are activated for its switch to the upper steady state (Fig. 4a (yellow to red)). This process iterates as a series of ROS activation–diffusion events, emerging as self-regenerating trigger waves of ROS. Notably, our model predicts that increasing the erastin concentration (from 0 to 11.7 μM) exerts a relatively mild effect on the low-ROS steady state (Fig. 4a), but it apparently accelerates ferroptotic propagation and enhances the intensity of ROS wavefronts (Extended Data Fig. 10d,e). Thus, our simulations provide two testable predictions when the erastin concentration is increased: (1) the transition from monostable to bistable ROS; and (2) the dose-dependent response of ferroptosis propagation speed.

To experimentally test erastin-induced ROS bistability, we used photoinduction to modulate cellular ROS for the activation of its bistable switch. Within the predicted bistable regime, we anticipated that photoinduction (Fig. 4a (blue arrow)) would activate the ROS bistable switch by elevating ROS above the activation threshold. Consistently, when we increased the erastin concentration above 1.25 μM, the cellular ROS steady state switched to the upper level after photoinduction (Fig. 4b,c), similar to an ROS bistable switch. By contrast, at an erastin concentration of lower than 0.63 μM, cells maintained low ROS levels, resembling the monostable ROS regime. In particular, cells without erastin treatment did not exhibit ROS elevation regardless of the photoinduction intensity (Extended Data Fig. 11a), supporting their redox monostability. At the threshold level of 0.63 μM erastin, cells displayed a bimodal distribution of ROS levels, indicative of its proximity to the monostable/bistable bifurcation point. This monostable-to-bistable transition represents a cellular priming process during which individual cells acquire ROS bistability and the cell population becomes a bistable medium over which ROS propagate as trigger waves. Thus, erastin-treated cells exhibit properties of the ROS bistable medium predicted by our model, with features of an activation threshold for the ROS bistable switch, and a monostable-to-bistable bifurcation point.

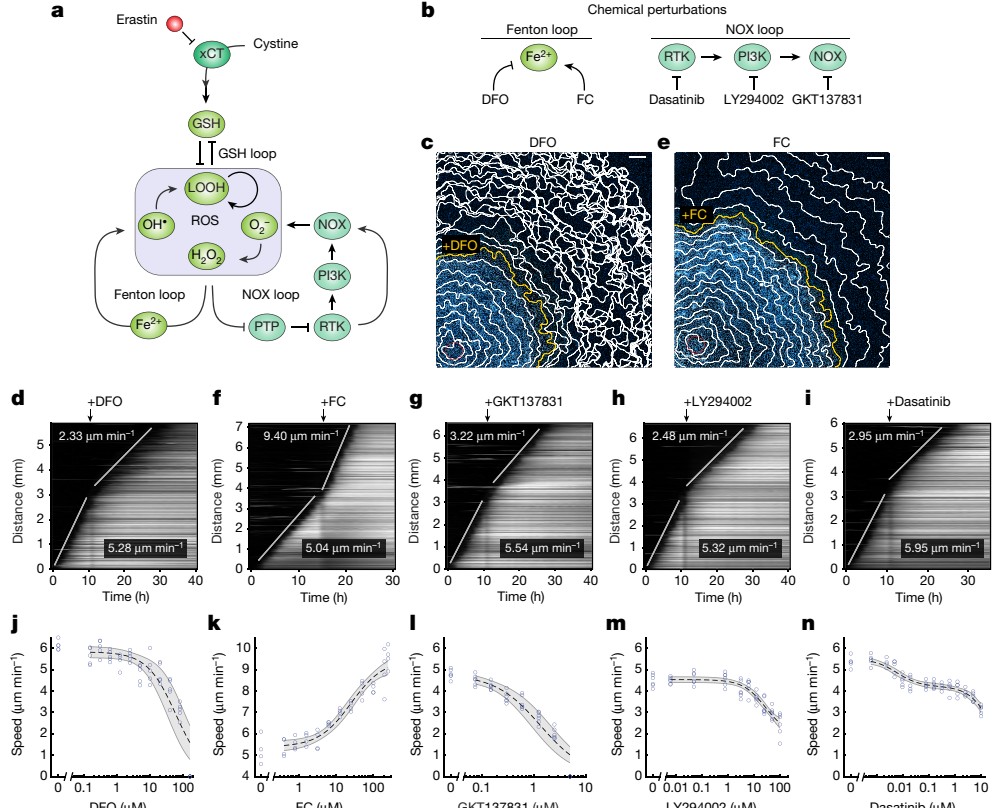

**Fig. 3 | ROS feedback loops modulate the speed of ferroptotic trigger waves.** **a**, A ferroptosis network includes three ROS feedback loops. **b**, Chemical perturbations applied to modulate the strength of Fenton and NOX feedback loops in erastin-treated cells. **c**,**d**, The speed of ferroptotic trigger waves declines after iron chelation. **c**, Nuclear dye fluorescence image (11 h after photoinduction) overlaid with cell death contours. The yellow contour indicates the timepoint at which DFO (80 μM) was added. **d**, Kymograph for the experiment in **c**. **e**,**f**, The speed of ferroptotic trigger waves increases after iron addition. **e**, Nuclear dye fluorescence image (15 h after photoinduction) overlaid with cell death contours. The yellow contour indicates the timepoint at which FC (250 μM) was added. **f**, Kymograph for the experiment in **e**. **g**–**i**, Addition of GKT137831 (1.25 μM) (**g**), LY294002 (100 μM) (**h**) or dasatinib (0.6 μM) (**i**) slows down ferroptotic trigger waves. **j**–**n**, Wave speed as a function of DFO (**j**), FC (**k**), GKT137831 (**l**), dasatinib (**m**) and LY294002 (**n**) concentrations. The dose–response curves (dashed line) were obtained by fitting the data to a Michaelian inhibition function for DFO, LY294002 and GKT137831; to a Michaelian activation function for FC; and to a biphasic inhibition function[49] for dasatinib (Methods). The shaded area bounded by grey lines represents the 95% confidence interval of model prediction. Data are derived from five (DFO, 0–0.6 μM and 2.5–160 μM), four (DFO, 1.25 μM), four (FC) and six (GKT137831, LY294002, dasatinib) technical replicates. Fitted parameters are shown in Extended Data Table 1. For **c**–**n**, data are representative of three biological repeats. Scale bars, 500 μm (**c** and **e**).

We next examined how erastin-mediated cellular priming may alter the states of ROS feedback loops. Increasing the concentration of erastin elicited a hyperbolic decrease in cellular GSH (Extended Data Fig. 11b). In contrast to the bistable nonlinear switch of ROS (Fig. 4c), a linear decrease in GSH was observed when the erastin concentration was increased from 0.31 μM to 1.25 μM. Thus, the bistable ROS switch is probably not a direct result of suppressing de novo GSH synthesis alone but an emergent property of ROS feedback-loop activation. Consistently, we found that labile iron and NOX activity, respective markers for the strength of the Fenton and NOX-signalling loops, increased with the erastin concentration (Extended Data Fig. 11c,d and 11f (bottom)), indicating interactions among these ROS feedback loops and their concerted activation during the cellular priming process. Concurrently, we detected a mild but consistent increase in ROS with increasing erastin concentration (Extended Data Fig. 11e and 11f (top)).

We also examined how the cellular priming status might quantitatively influence the behaviour of ferroptotic trigger waves. Consistent with our model prediction, the majority of the bistable regime enabled ferroptosis propagation, with wave speed increasing (from 3.08 μm min⁻¹ to 5.38 μm min⁻¹) in accordance with erastin concentration (Fig. 4d,e and Supplementary Video 9). Moreover, a faster propagating wave exhibited larger ROS wavefronts of higher amplitudes (Fig. 4f,g). Thus, ferroptosis stress primes cell populations, thereby endowing them with different degrees of progressivity for ferroptotic waves.

## Ferroptosis in the developing limb

Our cell-based assays show how ferroptosis propagates across a large population of cells as trigger waves. During embryonic development of complex organisms such as vertebrates, large-scale cell death is a recurrent mechanism that is responsible for shaping tissue structures and removing temporary embryonic-stage-specific tissues[1,2]. If ferroptotic waves act as a tissue-sculpting mechanism, it is conceivable that developmental signals (for example, morphogens) may encode spatial information that primes cells in different regions to exhibit distinct redox states. Regions in which cells are redox bistable would allow propagation of ferroptosis to eliminate unwanted structures. By contrast, cellular regions of redox monostability could act as boundaries to restrict the spread of ferroptosis, thereby facilitating region-specific tissue sculpting.

We investigated this possibility in the developing avian limb, in which large-scale cell death has been found to be a critical mechanism for muscle remodelling[42]. During this remodelling process, muscle masses are sculpted by cell death to form the anatomically distinct muscle bellies of the foot, shank and thigh[42,43]. At day 6.5 of development (stage HH30),

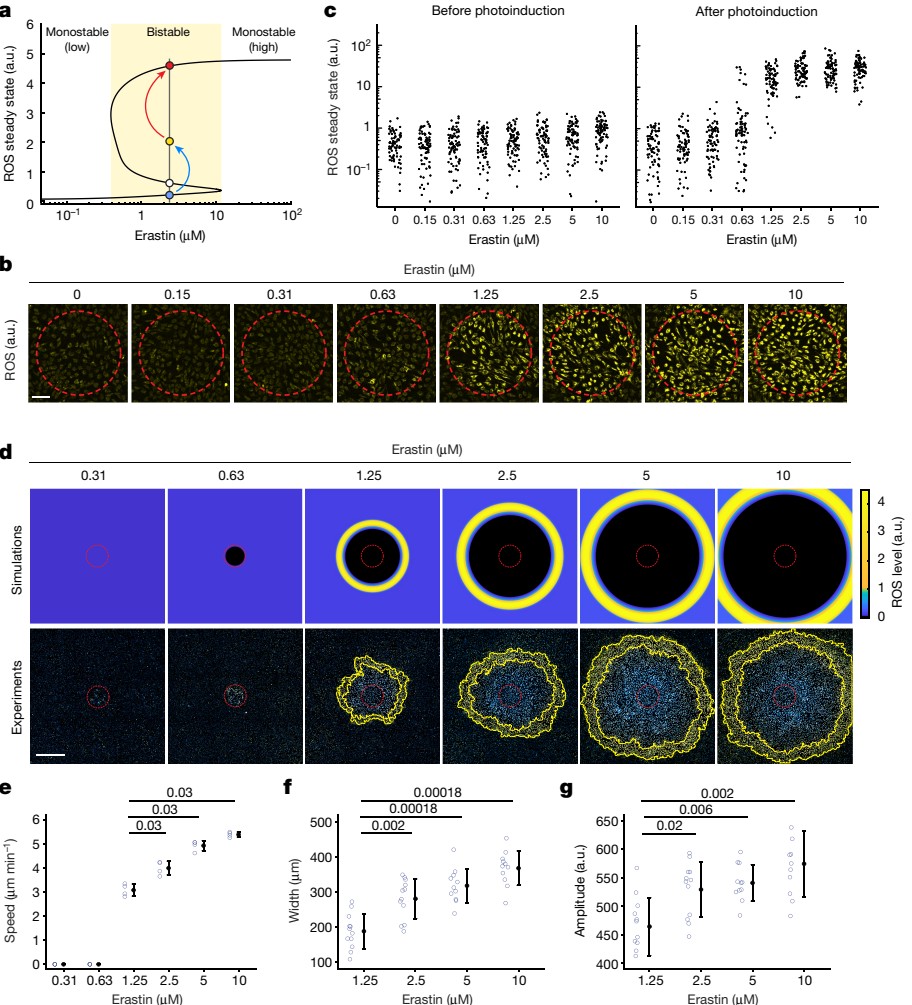

**Fig. 4 | Ferroptosis stress primes cells for ROS bistability and promotes the propagation of ferroptotic trigger waves. a**, In silico simulations showing ROS steady state as a function of erastin concentration. ROS steady state bifurcates from a monostable (low) to a bistable regime (yellow area) with increased erastin concentration. Stable low and high ROS steady states and the USS are denoted by blue, red and white circles, respectively. ROS elevation by photoinduction (blue arrow, elevation from blue to yellow circle) allows cells to surpass the USS, above which ROS is amplified (red arrow) to the high steady state. **b**, Images of ROS fluorescence (yellow) 20 min after photoinduction at different erastin concentrations. **c**, Single-cell ROS steady states were measured before and after photoinduction at different erastin concentrations in **b**.

For each erastin level, 80 cells are shown. **d**, Increasing erastin concentrations promote ROS wavefront propagation. Simulations (top) and experiments (bottom) of ROS propagation in cell populations treated with different erastin concentrations 6 h after photoinduction. **e**–**g**, Wave speed (**e**), wavefront width (**f**) and amplitude (**g**) at different erastin concentrations. Data are mean ± s.d. For wave speed, four technical replicates; wavefront width and amplitude were measured from twelve directions. Statistical analysis was performed using two-sided Wilcoxon rank-sum tests with false-discovery rate adjustment; *P* values are shown at the top. For **b**–**g**, data are representative of three biological repeats. Scale bars, 100 μm (**b**) and 800 μm (**d**).

foot muscles are continuous with the shank muscles, presenting longitudinal alignment (HH30; Extended Data Fig. 12a). From day 7 onwards, the foot muscles are progressively separated from the shank (7.5 day limb, stage HH32; Extended Data Fig. 12a), and this process is facilitated by massive cell death occurring along the central proximodistal axis of the zeugopod, as detected by TUNEL staining (muscular ventral death zone; Extended Data Fig. 12b). Concurrently, the foot and shank muscle masses segregate into individual muscle bellies through cell death, as indicated by myosin-positive cells with a rounded appearance (Extended Data Fig. 12d,e (top)). Notably, we detected colocalization of 4-hydroxynonenal (4-HNE), a lipid peroxidation marker and ferroptosis indicator, with dead cells in the muscular ventral death zone (Extended Data Fig. 12c) and in degenerating muscles (Extended Data Fig. 12d,e), indicating that muscle cells undergo ferroptosis during the process of muscle remodelling.

We next examined the occurrence of ferroptosis in the developing limb more broadly. We found that, in addition to the muscle layer, the dorsal and ventral ectodermal layers of the limb also exhibited abundant 4-HNE signals (Fig. 5a). Moreover, large-scale lipid peroxidation was detected at the ventral ectodermal layer, extending across the proximodistal axis of the limb zeugopod over a distance of greater than 2 mm (Fig. 5b). Consistently, whole-mount staining of 4-HNE indicated the occurrence of lipid peroxidation at the central region of the limb zeugopod (Fig. 5c), which coincided with the cell death observed in the same region detected by TUNEL staining (Fig. 5d). Quantification of 4-HNE and TUNEL signals in multiple limbs revealed stronger signals at the central region relative to the lateral regions of the zeugopod (Extended Data Fig. 12f–h), indicative of region-specific large-scale ferroptosis at the central ectodermal layer of the developing limb.

## Ferroptotic waves in the limb zeugopod

As ferroptosis has the potential to propagate over large areas, we examined whether the above-described large-scale cell death could

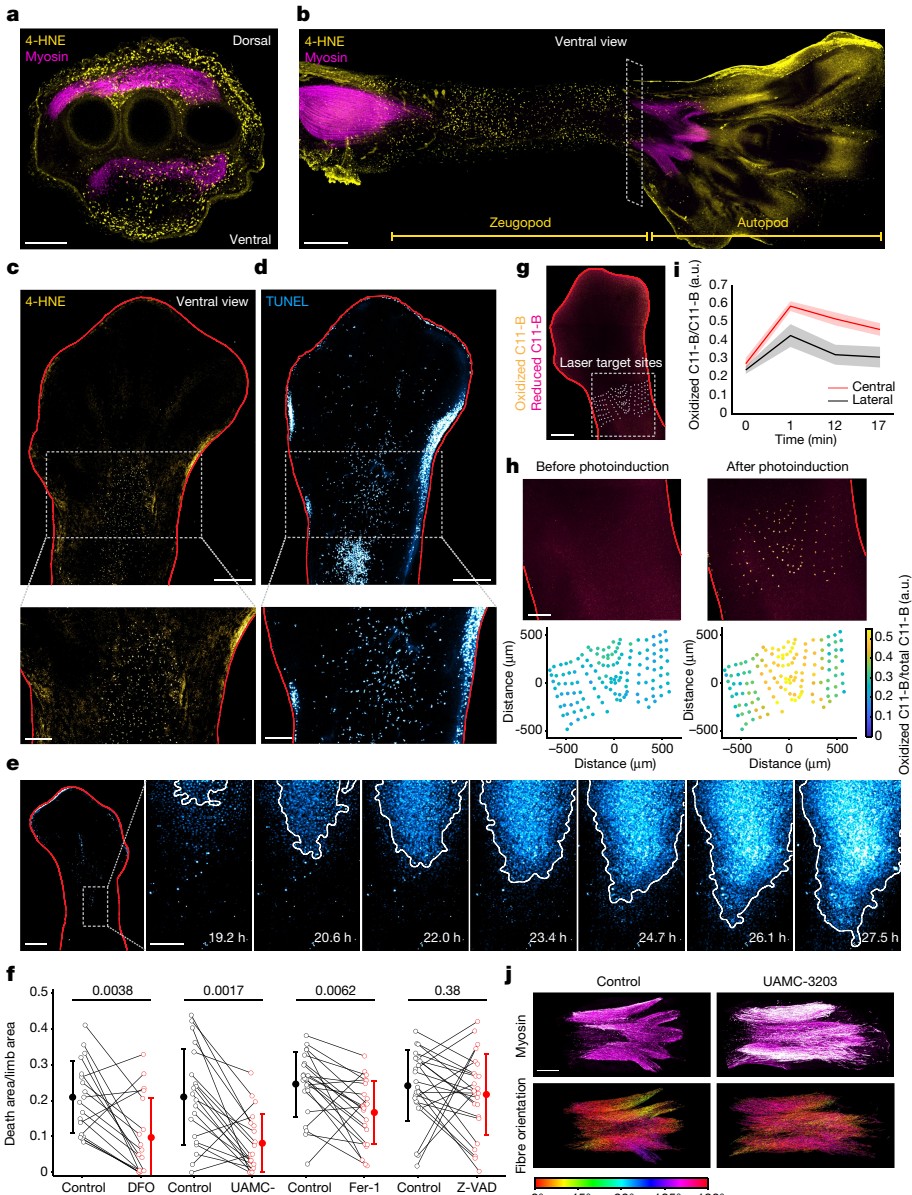

**Fig. 5 | Ferroptosis and its propagation facilitate muscle remodelling in the avian limb. a,b,** Co-immunostaining of lipid peroxidation marker (4-HNE, yellow) and myosin heavy chain (myosin, magenta) in transverse (**a**) and longitudinal (**b**) sections (HH33 limbs). The transverse section (**a**) is located at the zeugopod (box in **b**). The longitudinal section (**b**) features the ectodermal layer. **c,d,** Whole-mount immunostaining of 4-HNE (**c**) and TUNEL staining (**d**) in HH32 limbs outlined in red (magnified views are shown at the bottom). **e,** Nuclear dye fluorescence image (HH31 limb) (left). Right, time-lapse images of cell death (magnified views of the box are shown on the left). **f,** Cell death area with or without DFO (10 mM, $n = 18$), UAMC-3203 (1 μM, $n = 20$), Fer-1 (10 μM, $n = 22$) and Z-VAD-FMK (200 μM, $n = 24$). Data are mean ± s.d. Statistical analysis was performed using two-sided Wilcoxon rank-sum tests; $P$ values are shown at the top. **g,h,** PALP assay of C11-BODIPY[581/591] (C11-B)-stained limbs (merged images

of oxidized and reduced C11-BODIPY[581/591]). **g,** C11-BODIPY[581/591]-stained limb overlaid with the laser target sites. **h,** Magnified views of the box in **g** (top). Bottom, lipid peroxidation levels at target sites in the top images. **i,** Lipid peroxidation levels in the central (within 100 μm of the centre axis) and lateral (≥350 μm away from the centre axis) regions. Data are mean ± s.d. of 28 (central) and 61 (lateral) target sites. **j,** Immunostaining of myosin from UAMC-3203-treated and vehicle-control-treated embryos (HH33) (top). Bottom, muscle fibres colour coded according to their orientations. For **a**–**e** and **g**–**j**, data are representative of three biological repeats. Scale bars, 200 μm (**a**, **c** (bottom), **d** (bottom), **h** and **e** (right seven images)), 400 μm (**b**, **c** (top) and **d** (top)), 500 μm (**g**), 600 μm (**e** (left)) and 300 μm (**j**). Ventral views are shown unless otherwise stated.

be caused by ferroptosis waves. To this end, we used an ex vivo limb culture system (Methods) to monitor cell death occurrence in real time. As shown in Fig. 5e and Supplementary Video 10, cell death spread across the limb from distal to proximal regions at a speed of 1.15 μm min⁻¹, that is, the same order of magnitude relative to the wave speeds measured in our cell-based assay. Mean cell death propagation speed across three embryonic limbs was 1.30 ± 0.13 μm min⁻¹ (mean ± s.d.). Furthermore, the cell death area across the limb was significantly reduced by

treatment with various ferroptosis inhibitors (UAMC-3203, DFO and Fer-1), but not by the apoptosis inhibitor Z-VAD-FMK (Fig. 5f), indicating natural occurrence of ferroptosis in the ectodermal layer of the limb zeugopod.

Although it can extend over a large area, ferroptosis propagation is evidently confined to the central region of the proximodistal axis of the developing limb, with cells located in lateral regions remaining viable. As ferroptosis is involved in executing this cell death event,

it is possible that spatial patterning of polyunsaturated fatty acids (PUFAs) could act as a natural boundary to confine its propagation. To measure relative PUFA levels from the lateral to central regions along the proximodistal limb axis, we used photochemical activation of membrane lipid peroxidation (PALP)—an assay designed to measure relative PUFA levels using targeted laser pulses that photochemically induce lipid peroxidation[44] (Fig. 5g). As shown in Fig. 5h, PALP-induced lipid peroxidation was much greater in the central regions in which cell death propagates than in the lateral regions in which cells remain viable. The increase in lipid peroxidation in the lateral regions was transient, with lipid peroxidation reverting to basal levels soon after the application of laser pulses, indicative of their redox monostability (Fig. 5i). By contrast, the levels of lipid peroxidation at the central regions remained elevated, indicative of redox bistability. This result supports the hypothesis that different regions of the embryonic limb may exhibit distinct redox states to confine ferroptosis propagation.

## Limb muscle remodelling by ferroptosis

To examine a possible functional role of ferroptosis during muscle remodelling, we examined the consequence of suppressing ferroptosis on muscle development through in ovo injection of the ferroptosis inhibitor UAMC-3203. As shown in Fig. 5j, untreated embryos exhibited individualized muscle bellies at the anterior–posterior axis, each with a compact unit of muscle fibres well aligned with one another along a distinct orientation plane, represented by a coherent cell orientation angle. By contrast, suppressing ferroptosis resulted in excessive muscle fibres (quantification is shown in Extended Data Fig. 12i) and compromised the proper anterior–posterior segregation of the muscle masses. This phenotype was also accompanied by fibre disorganization, as revealed by the higher entropy of fibre orientations (quantification is shown in Extended Data Fig. 12j). These observations support the involvement of ferroptosis in regulating the number of muscle fibres and the subsequent individualization of the muscle mass during the development of avian limbs.

Ferroptotic cell death propagation in cell culture and in embryonic limbs exhibits several common properties. For example, their dependency on lipid peroxidation and cellular iron implies activation of shared ROS feedback loops that give rise to redox bistability. Supporting this notion, cell death propagates with speeds in the same order of magnitude in both systems. Moreover, similar to the multiple spontaneous death initiation events detected in cell culture, we also uncovered variation in the number of death initiation events in embryonic limbs (Supplementary Video 11), which may also be attributable to the stochastic nature of the cellular redox system (Extended Data Fig. 2). Nevertheless, the ferroptosis priming signals of both systems are distinct. Whereas ferroptosis propagation in cell culture results from erastin-mediated cellular priming, the wave-like propagation of ferroptosis in the embryonic limb arises from as-yet-uncharacterized developmental signals. Another distinct aspect of the two systems is the spatial area over which cell death waves extend. In contrast to the complete elimination of an entire cell population in culture, ferroptosis waves are confined to a specific region of the embryonic limb. One type of spatially encoded information from the priming signal is the level of PUFAs (Fig. 5g–i), the distribution of which constrains ferroptosis waves in the embryonic limb. Whether other ferroptosis regulators (for example, GPX4)[45] are modulated by developmental signals to exhibit region-specific activities remains to be examined.

## Discussion

Large-scale cell death during embryogenesis was initially observed in the nineteenth century[1]. These observations inspired decades of study to dissect the molecular mechanisms through which cell death is regulated, but few have pursued the fundamental question of how cell death is coordinated in space to permit the elimination of large populations of cells. These developmental cell death events were previously attributed to apoptosis. However, this conventional view has been challenged by several studies[46,47], suggesting the involvement of an as-yet-unclear mechanism that could act collaboratively and/or synergistically with apoptosis to coordinate developmental cell death events (such as ROS-mediated cell death[45,48]). Apart from during development, excessive cell death is also widely observed as occurring contiguously across tissues in disease pathologies, for example, during ischaemia–reperfusion injuries and degenerative diseases[3–5,9]. Our study provides one possible explanation for the occurrence of such massive cell death events. We show that ROS, as a ferroptotic death signal, can spread as trigger waves across cells over long distances without diminishing in speed. We have demonstrated that ferroptotic stress primes cell populations to become redox bistable by stimulating ROS feedback loops. Consequently, these redox bistable cell populations permit ferroptosis to propagate through a series of ROS amplification–diffusion events by means of short-range ROS diffusion as a spatial coupling mechanism. This mechanism of ferroptotic trigger waves overcomes the spatial limitations of simple diffusion to coordinate ferroptosis across cells, allowing the emergence of tissue-scale cell death. Finally, our study reveals ferroptosis propagation as a mechanism for eliminating temporary cell populations during muscle remodelling in developing limbs, substantiating its use as a tissue-sculpting strategy for shaping tissues and organs into appropriate forms during embryonic development. Our findings support the emerging notion that other modalities of cell death (for example, ferroptosis) alongside apoptosis are involved in organismal development, opening up future avenues for investigating the interplay between developmental signals and ferroptosis.

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

# Methods

## Reagents

Erastin (Cayman, 17754), RLS-3 (MedChemExpress, HY-100218A), ferrostatin-1 (Cayman, 17729), necrostatin-1 (Cayman, 11658), Z-VAD-FMK (Cayman, 14463), ionomycin (Sigma-Aldrich, I9657), staurosporine (MedChemExpress, HY-15141), GKT137831 (MedChemExpress, HY-12298), LY294002 (LC laboratories, L-7962), dasatinib (Cayman, 11498), FSEN1 (Cayman, 38025) and Trolox (MedChemExpress HY-101445) were all dissolved in DMSO and stored at −80 °C before usage. L-Buthionine sulfoximine (BSO) (Sigma-Aldrich, B2515), DFO mesylate salt powder (Sigma-Aldrich, D9533), Tiron (Sigma-Aldrich, 172553), TEMPO (Sigma-Aldrich, 176141), N-acetyl-L-cysteine (Sigma-Aldrich, A9165) and catalase (Sigma-Aldrich, C1345) were freshly prepared before each experiment by dissolving them in distilled $H_2O$. FC (Sigma-Aldrich, 3522-50-7) was prepared in distilled $H_2O$ and stored at −20 °C. Sytox Green nucleic acid stain (Invitrogen, S7020), siR-DNA probe (Spirochrome, SC007), siR700-DNA probe (Spirochrome, SC015), CellROX deep red (Invitrogen, C10422), C11 BODIPY$^{581/591}$ (Invitrogen, D3861), FeRhoNox-1 iron dye (Sigma-Aldrich, SCT030), DAPI (AAT Bioquest, 17513) and lucigenin (Cayman, 14872) were stored at −20 °C. NADPH (Sigma-Aldrich, N7505) was reconstituted in 10 mM Tris-HCl (pH 8) and stored at −80 °C. Phosphatase inhibitor cocktail (524629) and protease inhibitor cocktail (539134) were sourced from Calbiochem. Human holo-transferrin (R&D Systems, 2914-HT) was reconstituted in distilled $H_2O$ and stored at −80 °C. Phenol-red-free RPMI medium was made according to RPMI 1640 (Gibco, 11875) medium formulation using cell-culture-grade inorganic salts, amino acids and vitamins purchased from Sigma-Aldrich.

## Cell culture

hTERT RPE-1 (ATCC, CRL-4000), 786-O (ATCC, CRL-1932), G-402 (ATCC, CRL-1440), HOS (ATCC, CRL-1543), LN-18 (ATCC, CRL-2610), U-118 MG (ATCC, HTB-15), PANC1 (ATCC, CRL-1469), MDA-MB-231 (ATCC, HTB-26), HT-1080 (ATCC, CCL-121), NCI-H1650 (ATCC, CRL-5883), A549 (ATCC, CRM-CCL-185) and U-2 OS (ATCC, HTB-96) cells were cultured in RPMI 1640 medium (Gibco, 11875) containing 5% fetal bovine serum (FBS) (Sigma-Aldrich, TMS-013-BKR). HuH-7 (JCRB Cell Bank, JCRB0403), A-172 (ATCC, CRL-1620), Hs 895.T (ATCC, CRL-7637), HeLa (ATCC, CCL-2) and SH-SY5Y (ATCC, CRL-2266) cells were cultured in DMEM (Gibco, 11965) with 5% FBS. All cell lines were grown at 37 °C under 5% $CO_2$. Cells were passaged routinely to maintain around 80% confluency. All cell lines were tested and found to be free of mycoplasma.

## RPE-1 cells with *ERK2* overexpression

RPE-1 cell lines stably expressing *ERK2* and mCherry (control) were generated by lentivirus infection according to a standard protocol. In brief, lentivirus production was performed by HEK-293T transfection with the target construct (ERK2-P2A-eGFP or mCherry), packaging vector (pVSV-G) and envelope plasmid (psPAX2) using FuGENE HD transfection reagent (Promega, E2311), followed by lentivirus infection in RPE-1 cells. To generate the ERK2-P2A-eGFP lentiviral construct, the *ERK2* sequence (Addgene, 116760) was subcloned into a lentiviral backbone fused to a P2A-eGFP sequence using a Gateway recombination system (Invitrogen).

## Live-cell fluorescence microscopy

Time-lapse imaging experiments were performed using a Zeiss Axio Observer 7 inverted microscope equipped with a X-Cite Xylis (Excelitas Technologies, XT720S) LED illumination system, Definite Focus 2 system and a Prime BSI Scientific CMOS camera (Photometrics), under controlled temperature (37 °C), atmosphere (5% $CO_2$) and humidity (90–100%) in an environmental chamber. Images were taken with 2 × 2 binning using either a Zeiss ×10/0.45 M27 or ×20/0.8 M27 Plan-Apochromat objective. The fluorescence filter sets used for imaging experiments were as follows: Cy5 (Semrock, LED-Cy5-A-000), iRFP (Semrock, Cy5.5-C-000), FITC (Semrock, LED-FITC-A-000) and mCherry (Semrock, LED-mCherry-A-000).

## Enhanced nuclear dye fluorescence as a cell death reporter

To simplify imaging experiments, cell death was monitored according to the increased nuclear dye (siR-DNA or siR700-DNA dye) fluorescence that co-occurs with cell rupture (Extended Data Fig. 1a). To further validate this approach, we compared the increase in nuclear dye fluorescence with that of a cell death indicator, Sytox Green (3 nM), during ferroptosis. The increase in nuclear fluorescence consistently occurred 1–2 h earlier than the increase in Sytox Green fluorescence in all dead cells (Extended Data Fig. 1a,b). Sytox Green, siR-DNA and siR700-DNA were imaged using the FITC, Cy5 and iRFP filter sets, respectively.

## Comparison of cell death kinetics of different death types in RPE-1 cells

Cells ($5.4 \times 10^4$ cells per $cm^2$) were seeded on a 96-well μ-plate (Ibidi, 82406) in phenol-red-free RPMI medium with 5% FBS and siR-DNA (1 μM) 1 day before treatment with different death inducers (Extended Data Fig. 1d–j). For the cystine-starvation experiments, cells were washed three times and treated with cystine-free RPMI medium with 5% dialysed FBS (Gibco, 26400-044).

## Ferroptosis death kinetics in different cell lines

To obtain the death kinetics of erastin-induced (10 μM) ferroptosis in 16 different cell lines, and RSL3-induced (0.15 μM) ferroptosis in U-2 OS and A549 cells, the cells were seeded at densities of 25% confluency, followed by the indicated treatments after 2 days.

## Cell-based assay for ferroptotic trigger waves in RPE-1 cells

Two days before time-lapse imaging, $5.4 \times 10^4$ cells per $cm^2$ were seeded on a 24-well μ-plate (IBIDI, 82406) or a 96-well μ-plate coated with Matrigel matrix (Corning, 356231). Cells were grown in DMEM/F12 (Gibco, 21041025) supplemented with physiological levels of holo-transferrin (0.8–1.3 mg $ml^{-1}$). One day after seeding, the growth medium was replaced with homemade phenol-red-free RPMI with 0.15% FBS and the nuclear dye siR-DNA (1 μM). Cells were treated with erastin (10 μM) the next day, and imaged by fluorescence microscopy. To initiate ferroptotic death propagation, blue-light photoinduction (see the next section for details) was performed 8 h after erastin treatment. Generally, tiled (3 × 3 or 5 × 5) images were collected with a ×10 objective at a 1 h time interval.

## Photoinduction

For our standard photoinduction experiments, cells were exposed to blue light (Semrock, FF01-432/36-25) using a ×20 objective (60 mW for 10 s), unless otherwise stated. The size of the exposed area (~0.2 $mm^2$) was adjusted with an aperture diaphragm slider. To experimentally test ROS responses after light irradiation at different wavelengths (Extended Data Fig. 3e), we applied different filters (Semrock, FF01-378/52-25, FF01-432/36-25, FF01-474/27-25, FF01-509/22-25, FF01-554/23-25, FF01-578/21-25, FF01-635/18-25) to irradiate cells. A Gigahertz Optik Radiometer (PT-9610) was used to measure light power.

## Chemical perturbation of ferroptotic trigger waves

To quantify changes in trigger wave speed after chemical perturbations, we applied chemical inhibitors 11–15 h after photoinduction in 24-well μ-plates. To obtain the dose–response curves of wave speed in response to chemical inhibitors, experiments were performed in 96-well μ-plates into which chemical inhibitors had been added 4 h after photoinduction. To prevent potential toxicity, the final DMSO concentration was kept below 0.1%.

## Imaging cellular ROS and lipid peroxidation

To image cellular ROS and lipid peroxidation during ferroptosis propagation, cells were stained with CellROX (0.6 µM) or C11-BODIPY[581/591] (1.25 µM), respectively, in phenol-red-free RPMI medium with 0.15% FBS for 30 min. After staining, cells were washed twice with the growth medium before imaging. Filter sets were Cy5 for CellROX, and FITC and mCherry for C11-BODIPY[581/591].

## Intercellular gap creation

Intercellular gaps were created by scratching the bottom of the plate with needles of different tip sizes (20 µm to 400 µm) after wave initiation.

## Characterization of conditioned medium from ferroptotic cells

The cell death induced by conditioned medium from ferroptotic cells was assessed in a similar manner to a protocol described previously[27] with some modifications. Specifically, the recipient and donor RPE-1 cells were seeded at $4.5 \times 10^3$ cells per cm$^2$ and $5.4 \times 10^4$ cells per cm$^2$, respectively, according to the assay described in the 'Cell-based assay for ferroptotic trigger waves in RPE-1 cells' section above. Two days after seeding, the donor RPE-1 cells were treated with erastin (10 µM). After 12 h of treatment, when around 50% of cells were dead, erastin was washed out and replaced with erastin-free medium. After 4 h, these conditioned media were collected and pretreated with different ROS scavengers (Trolox, Fer-1, TEMPO and Tiron) (Extended Data Fig. 5b). In a separate experiment (Extended Data Fig. 5c), the conditioned media and $H_2O_2$-containing (100 µM) media underwent centrifugal filtration with a molecular mass cut-off of 30 kDa (Amicon ultracentrifugal filter). These ROS-scavenger-treated or filtered conditioned media were transferred to the recipient cells, before undergoing time-lapse imaging. As an erastin-wash control, we used a culture dish without cells to prepare the conditioned medium.

## GSH measurement

Total GSH level was measured using the GSH/GSSG-Glo Assay kit (Promega) according to the manufacturer's protocol. In brief, cells were seeded for 2 days in white-walled 96-well plates (Thermo Fisher Scientific, 136101). Before GSH measurement, cells were treated with erastin for 8 h. Luminescence signals were measured in relative light units using a SpectraMax Paradigm (Molecular Devices) microplate reader at 1 s integration time per well. GSH concentrations were interpolated from the linear range of the standard curve ($R^2 = 0.99$).

## Measurement of cellular labile iron

Intracellular labile iron ($Fe^{2+}$) levels were measured in RPE-1 cells 8 h after erastin treatment (Extended Data Fig. 11c and 11f (bottom)). Cells were stained with FeRhoNox-1 (5 µM) in phenol-red- and FBS-free RPMI medium. After incubation for 1 h, the cells were washed twice with the growth medium, then fixed (4% paraformaldehyde) and imaged. For image analysis, DAPI staining (30 nM) was applied to stain nuclei. FeRhoNox-1 and nuclear DAPI fluorescence signals were acquired using mCherry and DAPI filter sets, respectively. Similarly, to measure iron levels during cell death propagation, cells were stained as described above, and then photoinduced and processed for time-lapse imaging (Extended Data Fig. 6a,b).

## Measurement of NOX activity

NOX activities were measured using a lucigenin-derived chemiluminescence assay, as described previously[50] with some modifications. After 8 h of erastin treatment, whole-cell homogenates were collected with modified HEPES buffer (140 mM NaCl, 5 mM KCl, 0.8 mM MgCl$_2$, 1.8 mM CaCl$_2$, 1 mM Na$_2$HPO$_4$, 25 mM HEPES, 1% glucose, pH 7) supplemented with phosphatase inhibitor cocktail and protease inhibitor cocktail. After three freeze ($-80$ °C)–thaw cycles, lucigenin was added

to the homogenates to a final concentration of 5 µM and incubated at 37 °C for 30 min in the dark. The homogenates were then added into 96-well plates (equivalent to $2 \times 10^5$ cells per well). Immediately before luminescence measurement, 400 µM NADPH was added to each well. Luminescence was measured every 3 min using the EnSpire Multilabel Plate Reader with an integration time of 1 s per well and the temperature was maintained at 37 °C. Data are presented as relative luminescence units after 30 min of recording.

## DPPH assay

The antioxidant potential of small molecule inhibitors (GKT137831, LY294002, dasatinib) was determined using a 2,2-diphenyl-1-picrylhydrazyl (DPPH) assay kit (Dojindo) according to the manufacturer's protocol. The absorbance at 517 nm was measured using the EnSpire Multilabel Plate Reader.

## Western blot analysis

Cells were lysed in RIPA buffer supplemented with phosphatase and protease inhibitor cocktails, and then allowed to homogenize on ice for 30 min. Protein lysates were collected from the supernatants after centrifugation at 14,000$g$ for 10 min. After protein quantification using the BCA assay kit (Thermo Fisher Scientific, 23227), protein lysates (15 µg per lane) were mixed with sample buffer and then incubated at 70 °C for 10 min before loading. Proteins were separated by 10% SDS–PAGE and blotted onto PVDF membranes (Millipore, IPFL85R). After blocking with 5% bovine serum albumin at room temperature for 30 min, the membranes were incubated with primary antibodies against ERK (1:2,000, Cell Signaling Technology, 9102) and phosphorylated-ERK (1:1,000, Cell Signaling Technology, 9106S) at 4 °C overnight, followed by incubation with secondary antibodies (IRDye 680RD goat anti-mouse IgG, LiCOR, 926-68070; IRDye 800CW donkey anti-rabbit IgG, LiCOR, 925-32213) at room temperature for 1 h. The bands were visualized using the Typhoon laser scanner. Finally, the membranes were stained with Ponceau S to validate the transfer efficiency and to quantify protein loading.

## RT–qPCR analysis

Total RNA was extracted with TRIzol reagent (Invitrogen, 15596018) and reverse transcribed using the iScript cDNA synthesis kit (Bio-Rad, 1708891). Quantitative PCR with reverse transcription (RT–qPCR) was performed with iTaq Universal SYBR Green (Bio-Rad, 1708882) and monitored with a BioRadCFX96 system equipped with CFX Maestro software 2.3 (v.5.3.022.1030). The primers for qPCR were as follows: *NOX1*, fw: 5′-GTCTGCTCTCTGCTTGAAT-3′, rv: 5′-ATGAGATAGGCTGGAGAG-3′; *NOX2*, fw: 5′-CCCTTTGGTACAGCCAGTGAAGAT-3′, rv: 5′-CAATCCCGGCTCCCACTAACATCA-3′; *NOX3*, fw: 5′-ATGAACACCTCTGGGGTCAGCTGA-3′, rv: 5′-GGATCGGAGTCACTCCCTTCGCTG-3′; *NOX4*, fw: 5′-CAGAAGGTTCCAAGCAGGAG-3′, rv: 5′-GTTGAGGGCATTCACCAGAT-3′.

## TUNEL staining of chicken limbs

No ethical approval was required for experiments on chicken embryos at the desired Hamburger–Hamilton (HH) stage[51] 30–33. Fertilized Leghorn chicken eggs were incubated at 37.5 °C in a humidified incubator. Chicken hindlimbs were dissected and fixed for 12 h with 4% paraformaldehyde. The terminal deoxynucleotidyl-transferase-mediated dUTP-TRIC nick-end labelling (TUNEL) assay was performed using the In Situ Cell Death Detection Kit (Roche) according to the manufacturer's protocol. In brief, limbs and tissue sections were permeabilized at 37 °C with 2% Triton X-100 in PBS for 12 h. Antigen retrieval was performed by incubating in 0.1 M sodium citrate with 0.1% Triton X-100 in PBS at 70 °C for 30 min. TUNEL staining was conducted at 37 °C for 1 h, followed by imaging under the Zeiss LSM980 confocal microscope equipped with Zen Blue software (v.3.8). For tissue sections (Extended Data Fig. 12b,c), limbs were sliced to a thickness of 200 µm using a

vibratome. Immunostaining was applied to the tissue sections (see the 'Immunostaining of chicken limbs' section below), followed by double-labelling with TUNEL.

## Immunostaining of chicken limbs
Chicken limbs were fixed for 12 h with 4% paraformaldehyde at the indicated embryonic stages. For immunostaining of tissue sections, limbs were sliced to a thickness of 200 μm using a vibratome. Limbs and tissue sections were permeabilized with 2% Triton X-100 in PBS for 12 h. After permeabilization, blocking was conducted for 8 h with 10% FBS in PBS, followed by incubation with primary antibodies for 2 days, and with secondary antibody for 12 h in 2% Triton X-100 in PBS with 20% DMSO, unless otherwise specified. Whole-mount immunostaining of 4-HNE (Fig. 5c) was done in 0.5% Triton X-100 in PBS. All of the incubation steps were performed at room temperature. Imaging was conducted under the Zeiss LSM980 confocal microscope.

The following primary antibodies were used: anti-4-HNE (1:250, Abcam, ab46545; and 1:250, Abcam, ab48506) and anti-myosin heavy chain (2 μg ml$^{-1}$, Developmental Studies Hybridoma Bank, MF20). The following secondary antibodies were used: goat anti-rabbit IgG Alexa Fluor Plus 488, goat anti-mouse IgG Alexa Fluor Plus 647 and goat anti-rabbit IgG Alexa Fluor 568 (1:500, Invitrogen, A32731, A32728, A11036). The results of 4-HNE staining using anti-4-HNE (Abcam, ab46545) antibody was further validated with an additional anti-4-HNE antibody (Abcam, ab48506) in 2% Triton X-100 in PBS with 1% FBS.

## Time-lapse imaging of cell death in chicken limb ex vivo
Chicken limbs were dissected at stage HH31 and cultured in DMEM/F12 with 5% FBS and siR-DNA (5 μM). For imaging purposes, limbs were anchored to the bottom of the glass-bottomed plate (Mattek, P24G-1.0-13-F) by applying silicon glue (Picodent Twinsil, 13001000) to the farthest proximal region of the limb. To further stabilize the limb, we added a cover glass above the distal region of the limb. Time-lapse imaging experiments were performed using a Zeiss LSM980 confocal microscope or a Zeiss Axio Observer 7 inverted microscope under controlled temperature (35 °C), atmosphere (5% CO$_2$) and humidity (90–100%). Cell death was detected as increased fluorescence signal of siR-DNA dye, as described above. Images were taken every 1.5 h with a 639 nm laser (LSM980) and Cy5 filter set (Axio Observer 7) using the Zeiss ×10/0.45 M27 Plan-Apochromat objective.

## In ovo injection of ferroptosis inhibitor UAMC-3203
In ovo injection into the amniotic sac was conducted on stage HH30 embryos. Egg candling was performed to locate the target amniotic sac, and a hole was created using a 26 G 1/2 needle. Solutions of UAMC-3203 (4 mM) and DMSO (vehicle control) were prepared in saline (0.9% sodium chloride), and 120 μl of either solution was injected into the amniotic sac using a 26 G 1/2 needle. The success rate (>90%) of in ovo injection into the amniotic sac was determined by trial injections of a food dye solution. Chicken limbs were dissected at stage HH33 and fixed with 4% paraformaldehyde, before undergoing immunostaining.

## PALP assay
PALP assays[44] were performed using the Zeiss LSM980 confocal microscope equipped with Chameleon Ultra (Coherent) for two-photon laser excitation. Specific circular areas (98 μm$^2$) were stimulated by an 800 nm two-photon laser at 5% power output with eight iterations (scan speed = 7 fps; pixel time = 0.51 μs). To acquire images for C11-BODIPY fluorescence, 488 nm and 561 nm lasers were used.

## Computation of the travelling distances of diffusive molecules
The travelling distances for diffusive molecules in Fig. 1e were calculated using the diffusion equation $d = \sqrt{2Dt}$, where $d$ is the distance, $D$ is the diffusion coefficient and $t$ is time ($D$ of calcium = 200 μm$^2$ s$^{-1}$ and $D$ of a small globular protein = 10 μm$^2$ s$^{-1}$).

## Summary of image processing and data analysis
The image preprocessing procedures—including flatfield correction, ratiometric image calculation, image alignment and stitching of tiled images—were implemented using ImageJ (v.1.54 f). For image presentation, raw images were median-filtered with a circular area of 20 μm$^2$.

Wave outlines, representing the boundaries of wave propagation, were generated by identifying the top 1% of the image's fluorescence intensity by image binarization, followed by pixel dilation and mean filtering to visualize wavefronts of cell death (Fig. 1b,c), lipid peroxidation (Fig. 1g,h) and ROS (Fig. 2a and Extended Data Fig. 4a,b). The wavefronts of lipid peroxidation and ROS represent the increase in their signals, obtained by the fluorescence intensity difference between two consecutive images along time. Size filters were applied to exclude debris from analysis.

Fluorescence intensities of cell death, lipid peroxidation (Fig. 1f (bottom)) and cellular ROS (Extended Data Fig. 4a (bottom)) across space were obtained by calculating the mean signal intensities along the indicated distance with width (250 μm), and normalizing to their minimum and maximum intensities. The same procedure was done for quantifying ROS across space in Fig. 2a, but without signal normalization. The mean fluorescence intensities were smoothed with a window size of 50 μm.

To analyse the spatial and temporal patterns of cell death events, vector fields (Extended Data Fig. 1c) were constructed from the cell death outlines described above. The vectors, which indicate the directionality between death events, were generated based on two consecutive cell death outlines using the gradient function in MATLAB (v.R2023b). As an index to quantify spatial and temporal associations between death events, we computed the entropy of the vectors' angle distribution along different directions from an initial cell death event. Specifically, a distribution of vectors along a specific direction was obtained from the initial death event to the next successive death events across 3 h within its neighbourhood (typically 100 μm × 350 μm). The entropy of the vectors was calculated as $H = -\sum_{i=1}^{30} p_i \log_2 p_i$, where $p_i$ is the frequency that the angle of vector falls in the $i$th bin (30 bins in 360°).

Kymographs were generated using an array of cropped images of the fluorescent signals (cell death, lipid peroxidation, ROS or iron) along the direction of wave propagation ($y$ axis). For each cropped image, the value for each pixel on the $y$ axis represents the maximum signal along its width (250 μm). This operation was repeated on each cropped image for all timepoints, and the intensity lines were stacked along the $x$ axis. For speed measurement, we first marked the earliest time for each position on the $y$ axis that the threshold of fluorescence signals was reached (defined as the top 10% of the signal intensity for the kymograph). This yielded a distribution of spatial locations representing the wavefront for each timepoint. The top 11 locations closest to the mode of the distribution at each timepoint were used to estimate the wave speed using the polyfit function in MATLAB.

To characterize the ferroptosis initiation sites (Extended Data Fig. 2), a threshold-based segmentation was initially applied to binarized bright-field images to identify dead cells. We define a ferroptosis initiation site as the area where: (1) >5 dead cells are initially found within a circle of 40 μm radius, and (2) the cell death area increases (less than 20-fold) over time. All automatically identified initiation sites were manually inspected to eliminate false-positive sites (such as debris). In total, 761 ferroptosis initiation sites across 756 positions (area of 1.26 × 1.26 mm$^2$) were identified over a 5 h period. The distribution of the number of ferroptosis initiation sites was then compared with a Poisson distribution with the same mean (1.01 initiation events per 5 h) with $P = 1$ using a two-sample Kolmogorov–Smirnov test. To examine the distribution of time interval between two consecutive initiation events, we randomly combined the 756-time series of ferroptosis initiation events to calculate the time interval distribution.

When compared with a geometric distribution with the same mean (5.16 h), we determined a $P = 0.23$ using a two-sample Kolmogorov–Smirnov test.

The dose–response curves in Fig. 3j–n were obtained by fitting the data to a Michaelian inhibition function, $y = y_0 + (y_M - y_0)\frac{K}{K+x}$, for DFO, LY294002 and GKT137831; to a Michaelian activation function, $y = y_0 + (y_M - y_0)\frac{x}{K+x}$, for FC; and to a biphasic inhibition function, $y = y_0 + y_M\left(1 - f_1\frac{x}{K_1+x} - (1-f_1)\frac{x}{K_2+x}\right)$, for dasatinib[49], where $y$ is the trigger wave speed, $x$ is the drug concentration, and all of the other parameters are determined by model fitting.

In Fig. 4c, the ROS levels in individual cells were quantified 1 h before and after photoinduction using a semi-automatic cell tracking program, as described previously[41]. The slope of increased ROS was computed as an estimate for the ROS steady state.

The mean intensities of ROS and iron dye fluorescence (Extended Data Fig. 11c,e) were quantified in whole cells by nuclear segmentation using the ImageJ StarDist plugin, followed by nuclear dilation 7 pixels from the nuclear border. Size filtering was applied to exclude incorrect segmentation and dividing cells.

The widths and amplitudes of a ROS wavefront (Fig. 4f,g) were measured as the distances and the maximal intensities of ROS signal between two consecutive ROS outlines in multiple directions.

Maximum intensity projections of confocal (Fig. 5a–d,g,h,j and Extended Data Fig. 12) and epifluorescence (Fig. 5e) image stacks were obtained using the ImageJ Stack Focuser plugin. The margins of the limbs were outlined by threshold-based segmentation.

To quantify the co-localization of degenerating muscles (myosin heavy chain-positive and rounded cells) with 4-HNE, the images were initially binarized on the basis of the thresholds of their respective backgrounds, which enabled identification of signal-positive regions. The binary images of degenerating muscles were then used to construct 3D objects using the regionprops3 function in MATLAB, followed by dilation (3 μm from the object). The percentage of 4-HNE colocalization with the degenerating muscles was calculated as the ratio of the number of 4-HNE-positive degenerating muscles to the total number of degenerating muscles.

For the images in Fig. 5j, 3D median filter and background subtraction was applied using ImageJ. To further remove the background signals, threshold-based 3D surface segmentation was performed in Imaris (v.10.0.1). To quantify the numbers of muscle fibres and their orientations (Extended Data Fig. 12i,j), we better visualized the fibre structures by applying a tubeness filter (ImageJ), followed by ridge detection (ImageJ) to identify individual muscle fibres. The muscle fibre count (Extended Data Fig. 12i) represents the total number of detected fibres across all $z$ stacks encompassing the ventral foot muscles. The muscle fibre orientations were determined using the ImageJ OrientationJ plugin and are colour coded in Fig. 5j. To quantify the entropy of the fibre orientations, fibres within a 900 μm$^2$ area were considered, and the entropy of the fibre orientations was calculated as $H = -\sum_{i=1}^{90} p_i \log_2 p_i$, where $p_i$ is the frequency with which the fibre orientation falls in the $i$th bin (90 bins from 0° to 180°).

In Fig. 5h,i, lipid peroxidation was quantified using the following formula: $\frac{\text{Oxidized C11} - B}{\text{Oxidized C11} - B + \text{reduced C11} - B}$. The mean intensities of lipid peroxidation were quantified at the laser target sites before and after photoinduction.

The wave speed in Fig. 5e was measured by generating kymographs as described above. To quantify the area of cell death in Fig. 5f, a threshold was applied to identify the area of cell death, which was normalized to the total limb area. For image presentation in Fig. 5e, the debris outside the limb were threshold-filtered followed by manual removal.

## Computational modelling of ROS trigger waves

We developed a reaction–diffusion model to model propagation of ROS trigger waves. Two types of ROS feedback loops were considered,

that is, positive-feedback loops (through NOX and Fenton loops) and a double-negative feedback loop (through GSH). For simplicity, we modelled the two positive-feedback loops as a single term. The resulting partial differential equation is:

$$\frac{\partial \text{ROS}}{\partial t} = D\frac{\partial^2 \text{ROS}}{\partial x^2} + k_{\text{positive-fb}}\overbrace{\frac{\text{ROS}^{n_{\text{positive-fb}}}}{EC50_{\text{positive-fb}}^{n_{\text{positive-fb}}} + \text{ROS}^{n_{\text{positive-fb}}}}^{\text{positive-feedback loop}}$$

$$- k_{\text{d-negative-fb}}\left(c_{\text{GSH}} + \frac{EC50_{\text{erastin}}}{EC50_{\text{erastin}} + E}\right)\overbrace{\frac{EC50_{\text{d-negative-fb}}^{n_{\text{d-negative-fb}}}}{EC50_{\text{d-negative-fb}}^{n_{\text{d-negative-fb}}} + \text{ROS}^{n_{\text{d-negative-fb}}}}^{\text{double-negative-feedback loop}}\text{ROS}$$

$$- k_{\text{deg}}\text{ROS} + k_{\text{synth}}$$

where ROS denotes the cellular ROS level and $E$ denotes the erastin concentration. The parameters were as follows: $D = 178\ \mu\text{m}^2\ \text{min}^{-1}$, $k_{\text{positive-fb}} = 1.2\ \mu\text{M}\ \text{min}^{-1}$, $EC50_{\text{positive-fb}} = 1\ \mu\text{M}$, $n_{\text{positive-fb}} = 3$, $k_{\text{d-negative-fb}} = 1.5\ \text{min}^{-1}$, $c_{\text{GSH}} = 0.1$, $EC50_{\text{erastin}} = 0.27\ \mu\text{M}$, $EC50_{\text{d-negative-fb}} = 2\ \mu\text{M}$, $n_{\text{d-negative-fb}} = 3$, $k_{\text{deg}} = 0.26\ \text{min}^{-1}$ and $k_{\text{synth}} = 0.1\ \mu\text{M}\ \text{min}^{-1}$.

The ROS positive and double-negative feedback loops were modelled as hyperbolic functions with Hill coefficients ($n_{\text{positive-fb}}$ and $n_{\text{d-negative-fb}}$) equal to 3. The impact of erastin on the ROS double-negative feedback loop follows a hyperbolic response, with $EC50_{\text{erastin}} = 0.27\ \mu\text{M}$. $c_{\text{GSH}}$ represents the basal production of GSH independently of cystine import.

In our simulations, the diameter of a cell was assumed to be 16 μm with an intercellular distance of 5 μm. To simulate the initiation of ROS trigger waves, we allowed all cells to first reach their lower ROS steady states as a function of erastin concentration. A local ROS elevation (photoinduction area = 0.2 mm$^2$) was simulated to surpass the USS threshold, mimicking blue light irradiation in the initiating cells (Fig. 4a (blue arrow)). When the USS threshold was surpassed, ROS underwent a bistable switch to its higher steady state (Fig. 4a (red arrow)). We defined the ROS threshold for cell death to be 90% of the higher steady state. After reaching this ROS threshold for 30 min, ROS production was stopped by setting $k_{\text{positive-fb}}$, $k_{\text{d-negative-fb}}$ and $k_{\text{synth}}$ to be zero, representing cell death.

## Reproducibility and statistical analysis

All experiments were independently performed at least three times with similar results. Technical repeats were performed in independent wells, and all the data from technical repeats are consistent across different biological replicates. The number of biological replicates is indicated in the figure legend as $n$. Details of statistical testing can be found in the corresponding figure legends. Wilcoxon rank-sum tests and the two-sample Kolmogorov–Smirnov tests were conducted in MATLAB. The wave propagation probability (Fig. 2b) was obtained by fitting the data to a logistic model using the fitglm function in MATLAB. The dose–response curves (Fig. 3j–n) were obtained by fitting the data to hyperbolic functions (indicated in the Methods above) using the nlinfit function in MATLAB.

## Data reporting

No statistical method was used to predetermine sample size. For the animal experiment in Fig. 5f, the left and right limbs of an individual animal were randomly allocated in control and experimental groups. For Fig. 5j and Extended Data Fig. 12h,i, animals were randomly allocated in control and experimental groups. Investigators were not blinded during data collection. However, data quantification was performed automatically using computational algorithms as described.

## Reporting summary

Further information on research design is available in the Nature Portfolio Reporting Summary linked to this article.

## Data availability

All data are available in the Article and its Supplementary Information. Microscopy data have been deposited at Figshare[52] (https://doi.org/10.6084/m9.figshare.25762806). Additional supporting microscopy data are available from the corresponding author on request, without any restrictions. Source data are provided with this paper.

## Code availability

Custom codes used for mathematical simulations and imaging analyses are available at GitHub (https://github.com/imb-lcd/ftw2024).

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

**Acknowledgements** We thank the members of our laboratory, Y.-C. Hsu, M.-R. Wu and J.-Y. Leu for discussions; J.-A. Chen, E. S. Liau, J.-H. Wei and Y.-C. Liu for their assistance with embryo experiments; and S.-P. Lee for her technical support with imaging. We acknowledge the funding support from Academia Sinica (AS-GCS-111-L04 and AS-GCP-113-L02) to S.-h.C., and the National Center for Theoretical Sciences, Physics Division, for the Postdoctoral Fellowship to C.-C.W.

**Author contributions** S.-h.C. and H.K.C.C. conceptualized the project. S.-h.C. and H.K.C.C. designed the experiments. S.-h.C., H.K.C.C. and Y.-C.L. conducted the experiments and analysed the data. C.-C.W. developed computational tools, and performed data analysis and simulations with intellectual inputs from S.-h.C. and H.K.C.C.; S.-h.C. and H.K.C.C. wrote the paper with input from all of the authors. S.-h.C. supervised and acquired funding for the research project.

**Competing interests** The authors declare no competing interests.

**Additional information**
**Correspondence and requests for materials** should be addressed to Sheng-hong Chen.

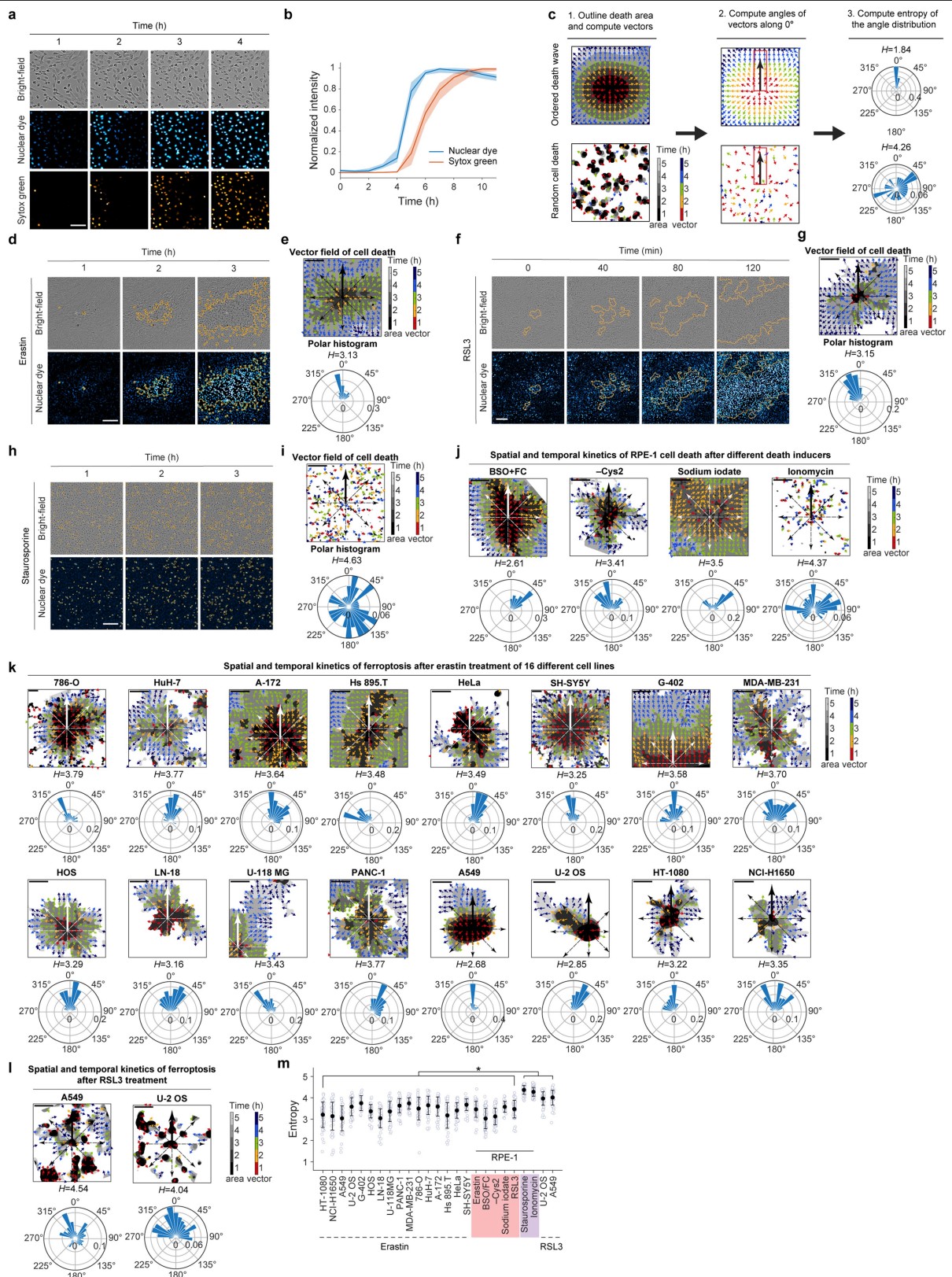

**Extended Data Fig. 1 |** See next page for caption.

**Extended Data Fig. 1 | Ferroptosis propagates across different cell types.**
**a**, Time-lapse images showing cell rupture (bright-field) co-occurring with an increase in nuclear dye fluorescence (cyan to white), followed by an increase in sytox green (orange) after erastin (10 μM) treatment. **b**, Single-cell quantification of nuclear fluorescence and sytox green signal upon ferroptosis. Data represent median and interquartile range for 20 cells. **c**, Vector field of cell death representing directionality of death events was generated from cell death contours (1). Each vector and each area bounded by contours are colour-coded for a specific time-point. The angles of the vectors along a specific direction (i.e., 0°) from the initial death event are computed (2). The distribution of vector angles is shown in the polar histogram (3). Entropy (*H*) is calculated as an index for the randomness of the vector angles (Methods). **d-i**, Spatial and temporal analysis of cell death events in RPE-1 cells after treatment with erastin (**d**, **e**), RSL3 (**f**, **g**), or staurosporine (**h**, **i**). **d**, **f**, **h**, Time-lapse images of cell death in RPE-1 cells treated with erastin (10 μM) (**d**), RSL3 (0.15 μM) (**f**) or staurosporine (0.15 μM) (**h**). Shown are bright-field and nuclear dye fluorescence images overlaid with cell death contours (orange outlines). **e**, **g**, **i**, Upper panel: vector

field of cell death. Lower panel: polar histogram with its entropy (*H*). **j**, Spatial and temporal analysis of cell death in RPE-1 cells induced by different chemicals: from left to right, ferroptosis inducers BSO (8 mM) + FC (1 mM), cystine starvation (- Cys2), sodium iodate (12.5 mM); and the intracellular calcium inducer ionomycin (2.5 μM). Upper panels: vector fields of cell death. Lower panels: example polar histogram of cell death vectors. **k**, **l**, Spatial and temporal analysis of erastin (10 μM)-induced ferroptosis in 16 different cell lines (**k**), and RSL3 (0.15 μM)-induced ferroptosis in U-2 OS and A549 cells (**l**). Upper panels: vector fields of cell death. Lower panels: example polar histogram of cell death vectors. **m**, Entropy calculation of cell death vector fields in (**e-l**). Data represent mean ± s.d. (18 angles from three cell death areas for each condition). The entropies of staurosporine- and ionomycin-induced cell death in RPE-1 cells, and RSL3-induced cell death in U-2 OS and A549 cells are significantly different from other conditions. Significance was tested by two-sided Wilcoxon rank-sum tests (* FDR-adjusted $P < 1 \times 10^{-5}$). Representative time-lapse movies for this experiment are shown in Supplementary Video 1. Scale bars, 100 (**a**, A549 and U-2 OS in **k**), and 200 (**d-l**) μm.

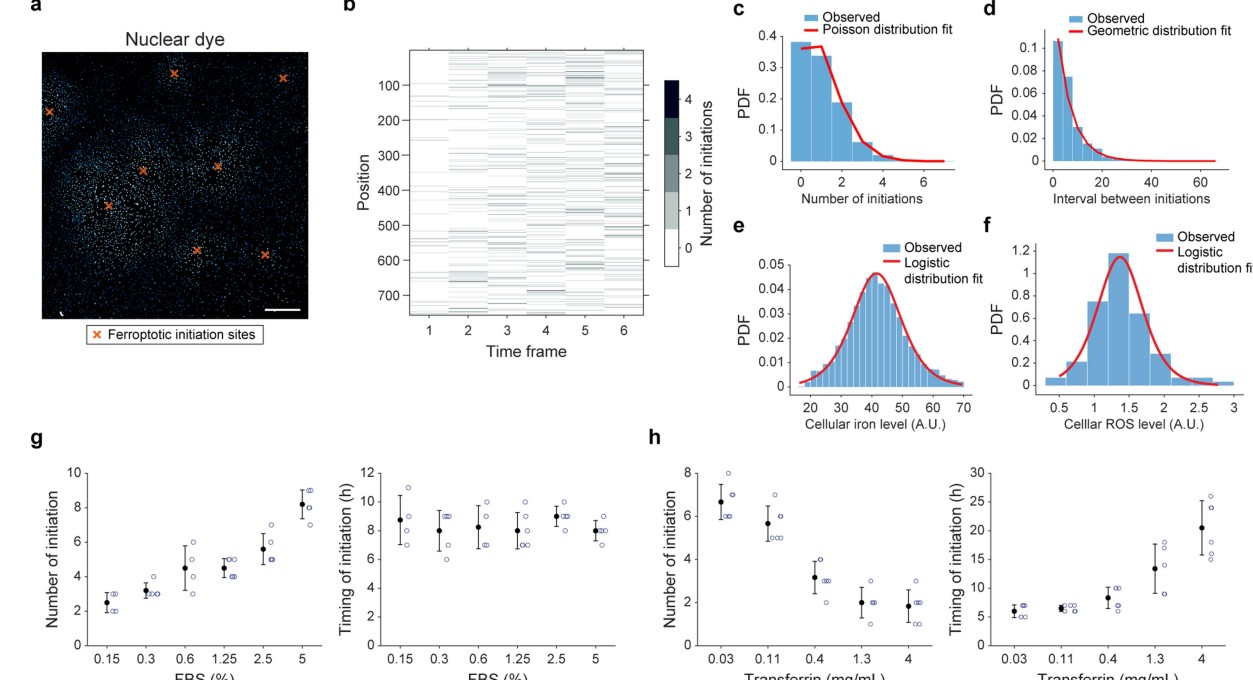

**Extended Data Fig. 2 | Ferroptosis initiation is a random process.**
**a**, A representative image of RPE-1 cells cultured in RPMI media + 5% FBS, followed by erastin (10 μM) treatment for 7 h. Orange crosses indicate sites of ferroptosis initiation events. Scale bar, 500 μm. **b**, Time series of ferroptosis initiation events from 756 positions. Time interval between frames is 50 min. **c**, Distribution of the number of initiation events identified over 5 h with its fit to a Poisson distribution (mean = 1.01 initiation events in 5 h, two-sample Kolmogorov-Smirnov test $P$ = 1, indicating no significant difference from the Poisson distribution). **d**, Distribution of the time interval between two consecutive initiation events with its fit to a geometric distribution (mean = 5.16 h, two-sample Kolmogorov-Smirnov test $P$ = 0.23, indicating no significant difference from the geometric distribution). **e**, **f**, Distributions of single-cell iron (**e**) and ROS (**f**)

levels in RPE cells after 8 h of erastin treatment with their fit to the logistic distribution. The two-sample Kolmogorov-Smirnov test $P$ are 0.35 (**e**) and 0.51 (**f**). **g**, **h**, Adjusting the concentrations of transferrin and FBS reduces the number of ferroptotic initiation sites. Number of initiation sites (left panel) and timing of initiation (right panel) after erastin (10 μM) treatment as a function of FBS (**g**) and transferrin (**h**) concentrations. Cells were seeded in RPMI media for experiments in (**g**). For experiments in (**h**), cells were seeded in DMEM/F12 supplemented with different transferrin concentrations without FBS, followed by media-change to RPMI with 0.15% FBS one day after seeding. Data represent mean ± s.d. from at least four technical repeats. All experiments were independently repeated three times with similar results.

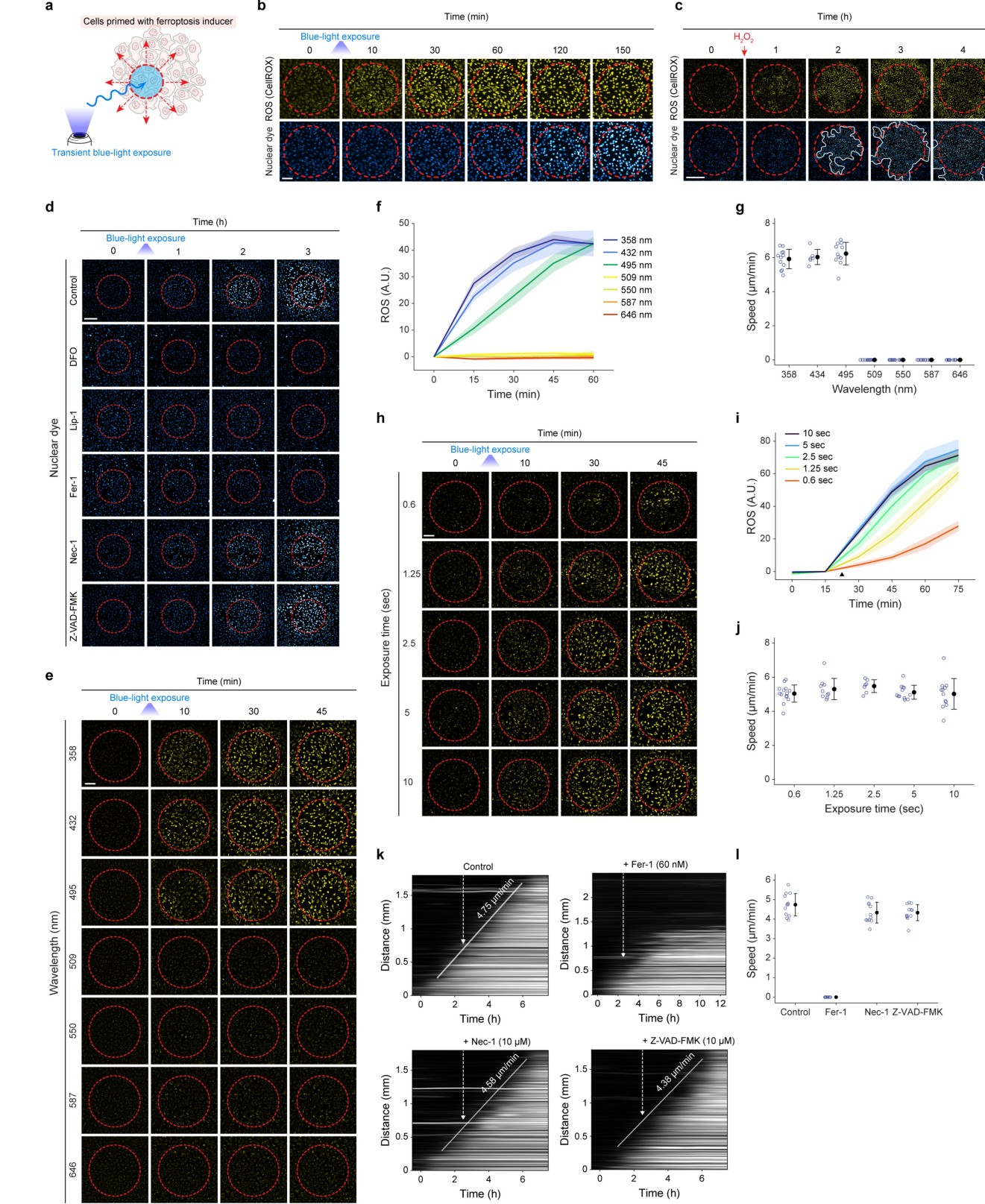

**Extended Data Fig. 3** | See next page for caption.

**Extended Data Fig. 3 | Blue light irradiation elevates cellular ROS levels, and causes ferroptosis cell death and its propagation in erastin-treated cells.**
**a**, A cartoon illustrating a cell-based assay for long-distance measurement of ferroptosis propagation (Methods). RPE-1 cells were treated with erastin prior to blue light (432 nm) irradiation. A local area (the photoinduction area marked with a red circle, ~0.2 mm²) was irradiated with blue light to initiate ferroptosis at a desired time and location. **b-d**, The area of the red circles in (**b**) and (**d**) was irradiated with blue light (432 nm, 60 mW) 8 h after erastin (10 μM) treatment. **b, c**, Time-lapse images of ROS (yellow) and cell death (nuclear dye, cyan) after photoinduction in (**b**), and after local addition of $H_2O_2$ (80 μM, 2 μL) (approximately within the area indicated by the red circles) in (**c**). **d**, Chemical inhibitors of ferroptosis (DFO, 200 μM; Lip-1, 30 nM; Fer-1, 60 nM), necroptosis (Nec-1, 10 μM), and apoptosis (Z-VAD-FMK, 10 μM) were added before photoinduction. Cell death was monitored for 3 h after photoinduction using nuclear dye. **e**, Time-lapse image sequences of ROS (yellow) in erastin-treated cells before and after photoinduction with different wavelengths of light (358 nm, 432 nm, 495 nm, 509 nm, 550 nm, 587 nm, 646 nm). **f**, Quantification of ROS at the photoinduction area in (**e**). Data represent mean ± s.d. of four wells. **g**, Speeds of ferroptosis propagation initiated by different wavelengths of light. Data represent mean ± s.d. from four wells with more than two directions calculated for each. **h**, Time-lapse image sequences of ROS (yellow) in erastin-treated cells before and after photoinduction with 432 nm light of different durations (0.6, 1.25, 2.5, 5, 10 sec). **i**, Quantification of ROS at the photoinduction area in (**h**). Data represent mean ± s.d. of four wells. **j**, Speeds of ferroptosis propagation initiated by different durations of light exposure (432 nm). Data represent mean ± s.d. from three wells with more than two directions calculated for each. **k**, Kymographs of cell death propagation in erastin-treated cells with or without addition of cell death inhibitors. Chemical inhibitors of ferroptosis (Fer-1, 60 nM), necroptosis (Nec-1, 10 μM), and apoptosis (Z-VAD-FMK, 10 μM) were added 2.5 h after photoinduction (white arrow). **l**, Speed measurements for experiments in (**k**). Data represent mean ± s.d. from four wells with three directions calculated for each. All experiments were independently repeated three times with similar results. Scale bars, 100 (**b**, **h**, **e**), 500 (**c**), and 200 (**d**) μm.

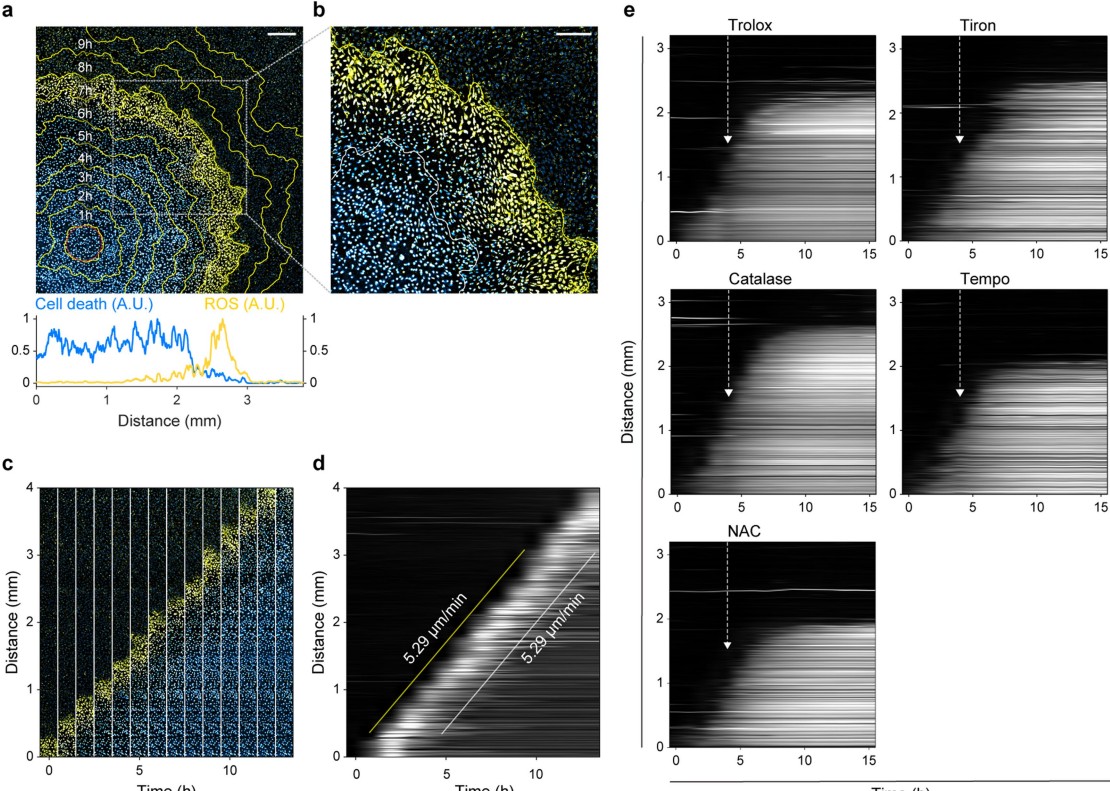

**Extended Data Fig. 4 | Cellular ROS (·OH, O₂⁻ and H₂O₂) wave fronts precede ferroptosis propagation. a**, **b**, Cellular ROS was monitored in erastin-treated cells using a general ROS dye (CellROX) that detects ·OH, O₂⁻ and H₂O₂. Images are derived from merging ROS (yellow) and nuclear dye fluorescence (cyan). Each yellow contour represents the border of the ROS wave front at a specific time-point. **a**, Upper panel: Image (8 h after photoinduction) overlaid with ROS contours 1-11 h after photoinduction. Lower panel: Fluorescence intensities of cell death and ROS signals were quantified across the bottom region of the image. **b**, Zoomed-in view of the box in (**a**). **c**, Time-lapse image array of ROS (yellow) and cell death (cyan) over 14 h. The image sequence was cropped from the same experiment in (**a**). **d**, Kymograph for ROS propagation in (**a**). The slopes of the yellow and white lines represent the speeds of ROS wave fronts and cell death propagation, respectively. The time-lapse movie for this experiment is shown in Supplementary Video 4. **e**, Kymographs of cell death propagation in erastin-treated cells after addition of ROS scavengers (Trolox, 6 μM; Tiron, 2 mM; catalase, 2000 U/mL; TEMPO, 125 μM; NAC, 15 μM) 4 h after photoinduction (white arrow). Data shown are representative of three biological repeats. Scale bars, 400 (**a**), and 250 (**b**) μm.

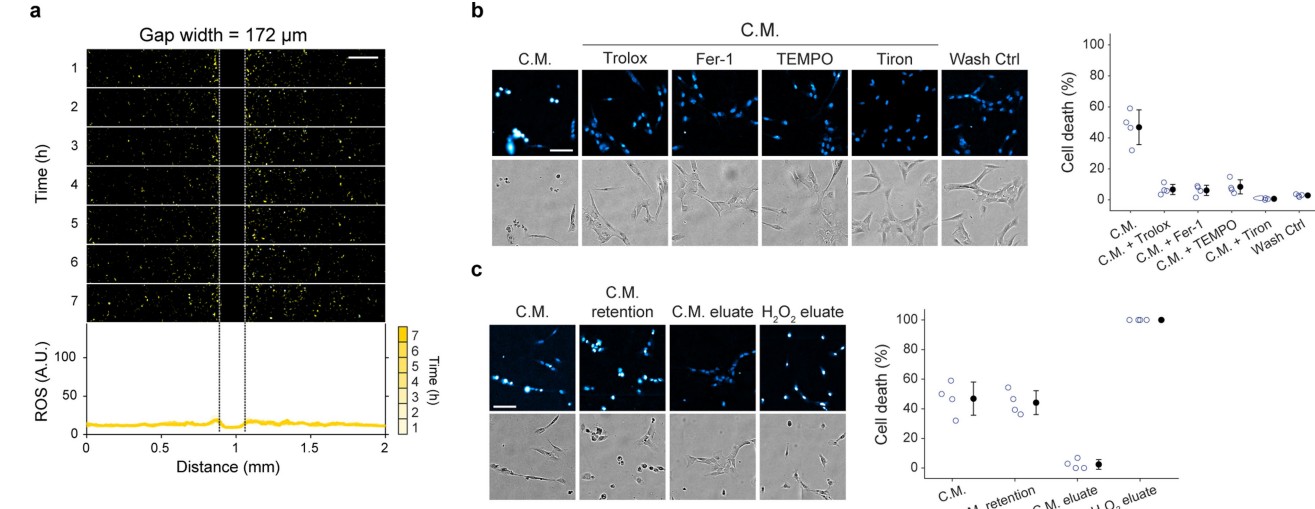

**Extended Data Fig. 5 | The spatial coupling mechanism involves the diffusion of a type of ROS. a**, Time-lapse image sequence of ROS (yellow) in erastin-treated cells across a gap (width = 172 μm). Lower panel: the mean intensity of ROS was calculated along the 2-mm distance at specific time-points. **b**, Left panel: Nuclear fluorescence and bright field images 20 h after incubation with erastin-free conditioned media (C.M.), ROS scavenger-treated C.M. (Fer-1, 60 nM; Trolox, 20 μM; TEMPO, 125 μM; Tiron, 2 mM), and media after washing out erastin-containing media in a culture dish without cells (Wash Ctrl). **c**, Nuclear fluorescence and bright field images 12 h after incubation with C.M., the eluate and retention fractions of the centrifugally filtered C.M., and eluate of the centrifugally filtered $H_2O_2$-containing media. Right panels in (**b**, **c**): Cell death (%) quantified from left panel. Data represent mean ± s.d. of four wells. All experiments were independently repeated three times with similar results. Scale bars, 200 (**a**), and 100 (**b**, **c**) μm.

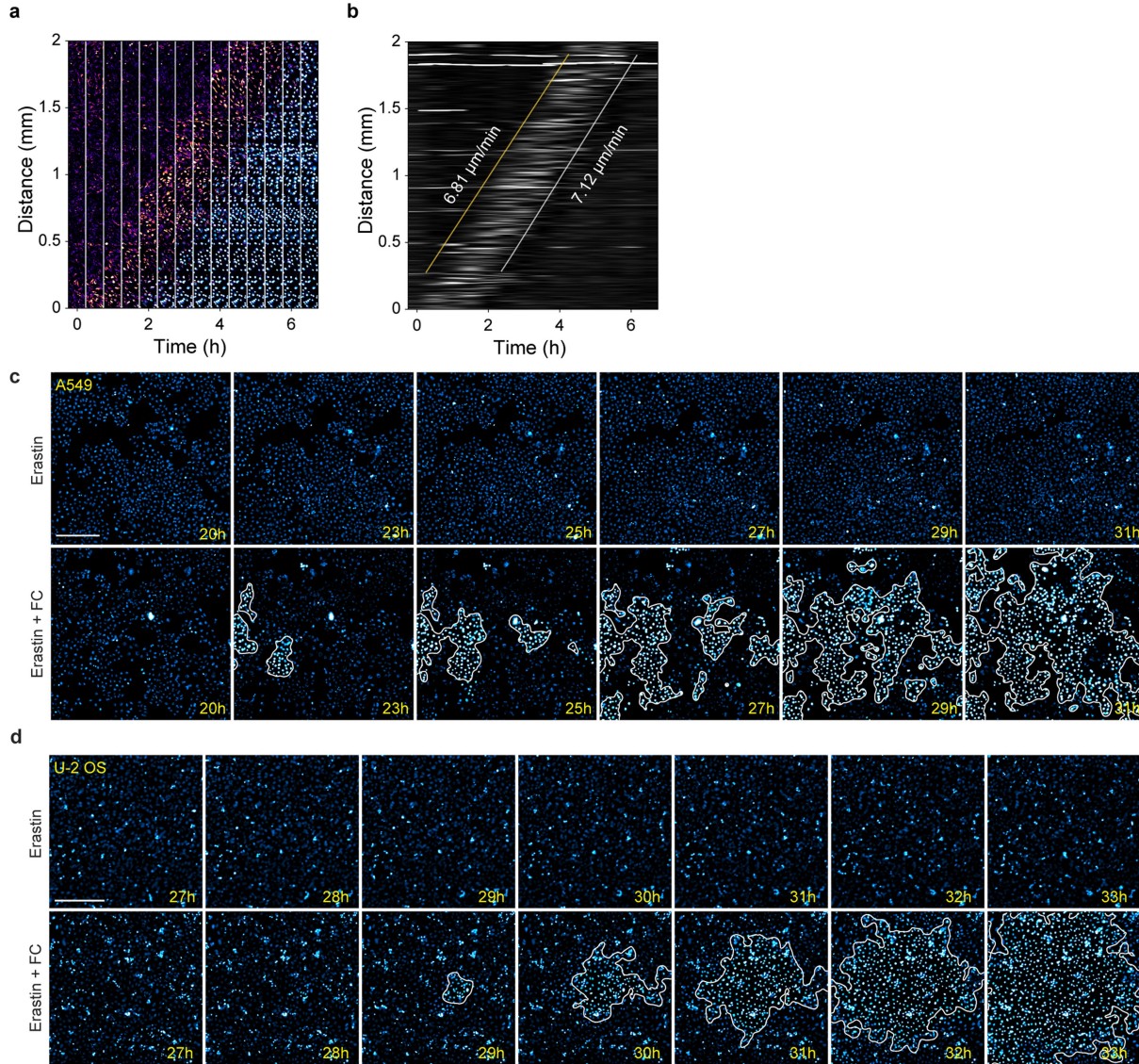

**Extended Data Fig. 6 | Fenton-mediated ROS feedback loop is critical for ferroptosis occurrence and propagation. a**, Time-lapse image array of cellular labile iron in pseudocolor (magenta) and cell death (cyan) propagation in erastin-treated cells. Cellular labile iron was monitored using an iron dye (FeRhoNox-1). **b**, Kymograph for cellular labile iron propagation from the image sequence shown in (**a**). The slopes of the yellow and white lines represent the speed of labile iron at the wave fronts and cell death propagation, respectively.

**c**, **d**, Enhancing Fenton-mediated ROS feedback loop induces wave propagation in cells gaining ferroptosis resistance at high confluency. Time-lapse images of A549 (**c**) and U-2 OS (**d**) cells at the indicated time points after erastin treatment (10 μM) with or without ferric citrate (FC, 125 μM). Cell death is indicated by increased nuclear dye fluorescence signal. The white outlines represent the boundaries of cell death areas. Experiments were independently repeated three times with similar results. Scale bar, 300 μm (**c**, **d**).

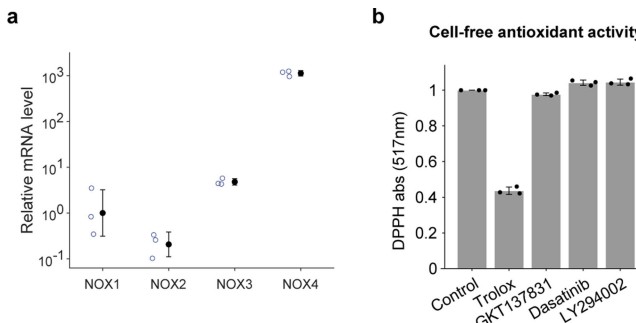

**a**

**b** Cell-free antioxidant activity

**Extended Data Fig. 7 | Targeting the NOX feedback loop by chemical inhibitors. a**, NOX4 is the dominant NOX isoform in RPE-1 cells. RT-qPCR was performed to quantify the relative mRNA levels of *NOX1-4* in RPE-1 cells. Data represent mean ± s.d. of three technical replicates. Data shown is a representative of three biological repeats. **b**, Small-molecule inhibitors targeting the NOX loop do not exhibit antioxidant potential. Cell-free antioxidant potentials of Trolox (32 µM), GKT137831 (5 µM), LY294002 (100 µM), and dasatinib (10 µM) were measured as the DPPH absorbance at 517 nm relative to DMSO (vehicle control). Data represent mean ± s.d. from three wells.

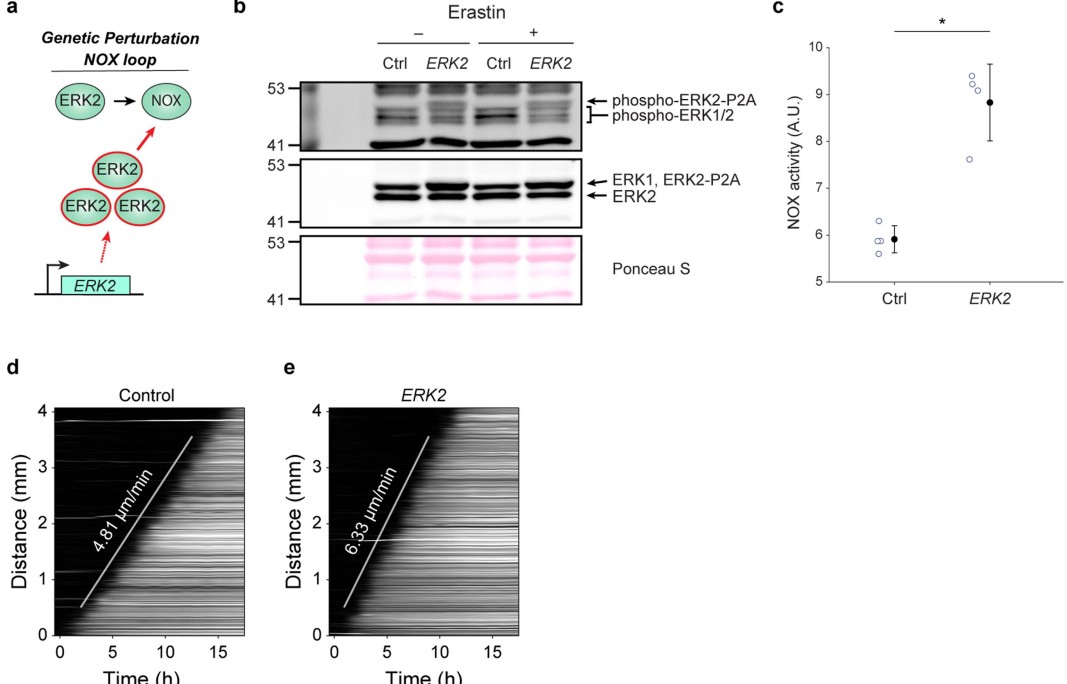

**Extended Data Fig. 8 | *ERK2* overexpression in RPE-1 cells increases phosphorylated ERK2, NOX activity, and the speed of ferroptotic trigger waves. a**, Genetic modulation of NOX signalling by *ERK*2 overexpression. **b**, Western blot analysis of overexpressed ERK2 (ERK2-P2A) and its phosphorylated form (phospho-ERK2-P2A) with or without erastin treatment (10 µM) in *ERK2*-overexpressing and control RPE-1 cells. The size of ERK2-P2A is bigger than wild type ERK2 and co-migrates with ERK1 due to its fusion to P2A peptide. For blot source data, see Supplementary Fig. 1. **c**, Relative NOX activity measured in *ERK2*-overexpressing and control RPE-1 cells. Data represent mean ± s.d. with four technical repeats. NOX activity of *ERK2*-overexpressing cells is significantly different from that of control cells (* two-sided Wilcoxon rank-sum test *P* = 0.0286). **d**, **e**, *ERK2* overexpression increases the speed of ferroptotic trigger waves. Kymographs representing ferroptosis propagation in control cells (**d**) and *ERK*-overexpressing cells (**e**). Data shown (**b**-**e**) are representative of three biological repeats.

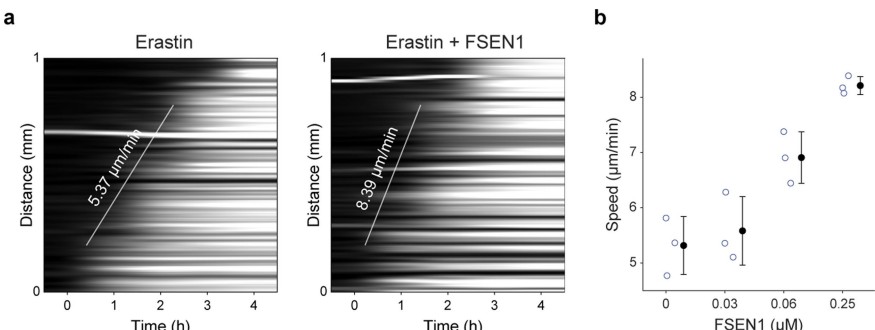

**Extended Data Fig. 9 | FSP1 inhibition increases ferroptosis wave speed.** **a**, Kymographs of cell death propagation in RPE-1 cells treated with erastin (left panel) or erastin + FSEN1 (0.25 µM) (right panel). **b**, Wave speed as a function of FSEN1 concentration. Data represent mean ± s.d. with three technical repeats. Experiments were repeated three times with similar results.

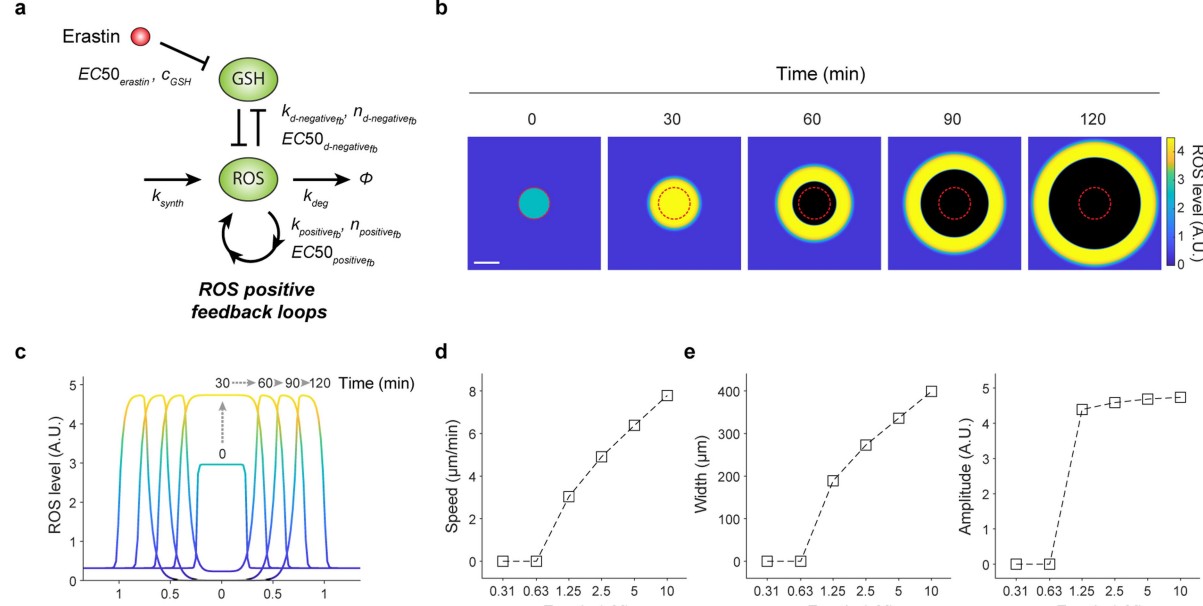

**Extended Data Fig. 10 | In silico simulations of ROS trigger waves. a**, A diagram of the ROS feedback loops and parameters used to build the mathematical model. **b**, **c**, Time-course simulations of ROS levels in a 200 × 200 population of cells treated with 10 μM erastin. Photoinduction was simulated by elevating ROS levels above the unstable steady state (USS) within the area of the red circle. Scale bar, 400 μm (**b**). **c**, Cross-section of ROS kinetics along the midline in (**b**). **d**, Speed of ROS propagation as a function of erastin concentration. **e**, Width (left panel) and amplitude (right panel) of the ROS wave front as a function of erastin concentration.

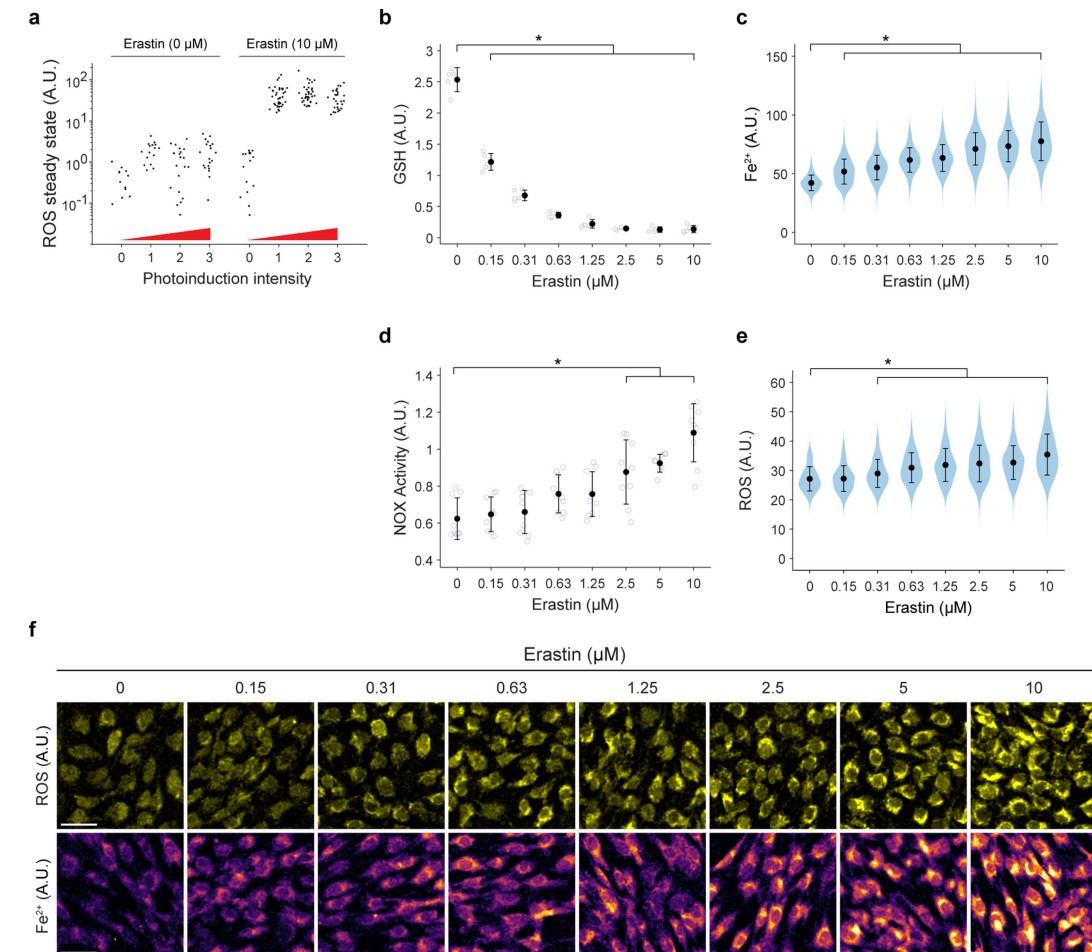

**Extended Data Fig. 11 | Erastin quantitatively modulates ROS feedback loops. a**, ROS steady state remains low regardless of photoinduction intensity in the absence of erastin treatment. Single-cell ROS steady states were measured in cells (40 cells) treated with or without erastin (10 µM) after photoinduction with different light intensities (1: 60 mW for 10 s; 2: 240 mW for 10 s; 3: 240 mW for 40 s). **b-e**, Cellular levels of GSH (**b**), labile iron (**c**), NOX activity (**d**), and ROS (**e**) after treatment with different erastin concentrations. Data represent mean ± s.d. (GSH: four technical replicates; labile iron: ≥ 257 cells; NOX activity:

three biological repeats, with three technical repeats each; and ROS: ≥ 249 cells). Measurements are significantly different from those of untreated (0 µM) cells (two-sided Wilcoxon rank-sum tests, *FDR-adjusted $P < 0.05$ (**b**), $<2 × 10^{-30}$ (**c**), $<4 × 10^{-3}$ (**d**), $<2 × 10^{-5}$ (**e**)). (**f**) Representative images of ROS (upper panels) and labile iron (lower panels) 8 h after different erastin treatments. Scale bar, 50 µm. All experiments were independently repeated three times with similar trends.

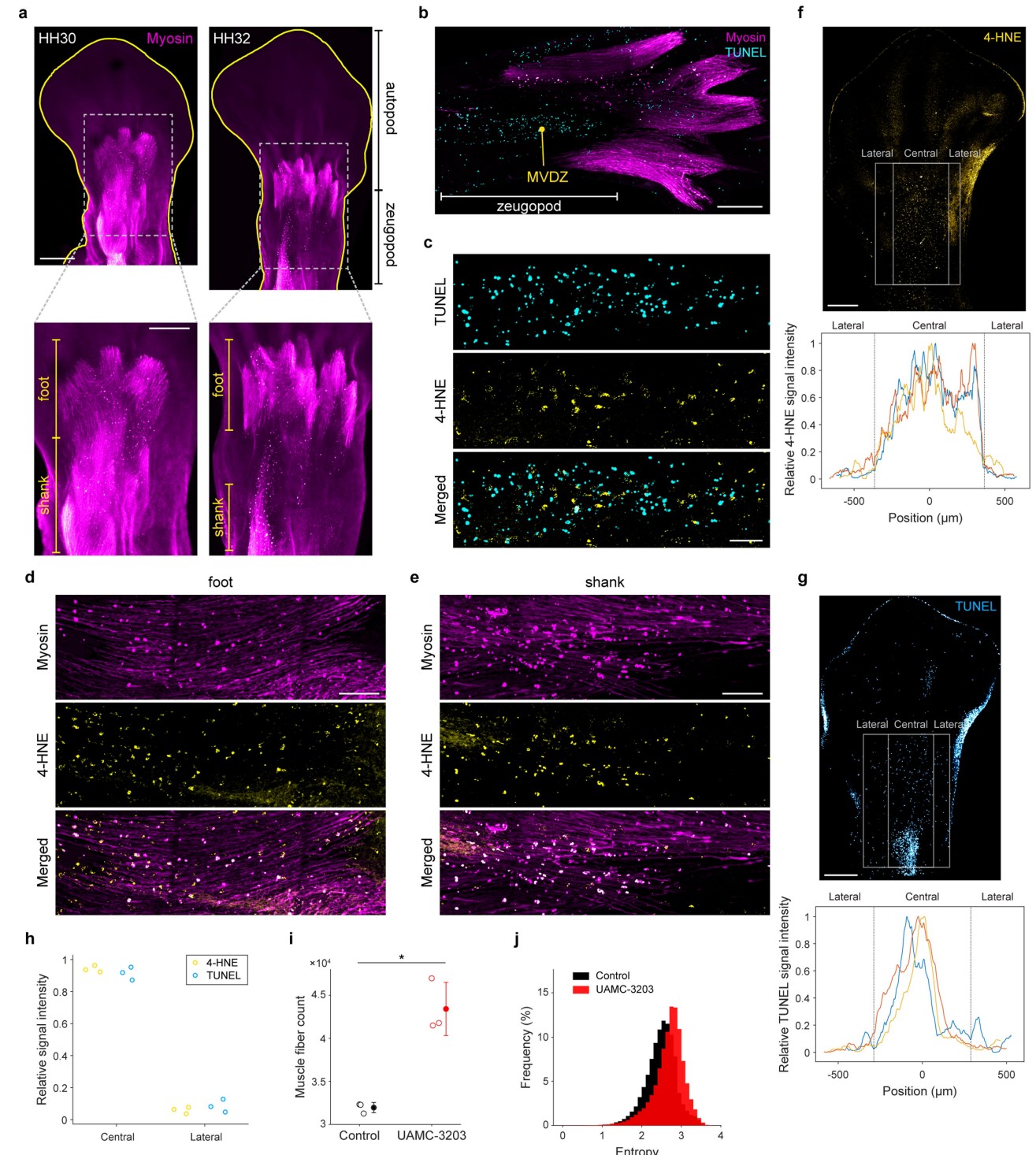

**Extended Data Fig. 12 | Ferroptosis is involved in muscle remodelling during avian limb development. a**, Whole-mount immunostaining of myosin heavy chain (myosin) in avian hindlimb at stages HH30 and HH32 of embryonic development. Ventral views of the limbs are shown, with limb margins outlined in yellow. Lower panels: Zoomed-in views of the boxes in the upper panels. The foot and shank muscles are labelled. **b-e**, Longitudinal sections of stage HH33 limbs. **b**, Co-staining of myosin and TUNEL, showing the muscular ventral death zone (MVDZ) at the zeugopod area. **c**, Co-staining of TUNEL and 4-HNE at the MVDZ in (**b**). **d**, **e**, Co-immunostaining of myosin and 4-HNE in the foot (**d**) and shank (**e**) regions. Degenerating muscles display a rounded and beaded appearance. Data shown (**a-e**) are representative of three biological repeats. **f**, **g**, 4-HNE and TUNEL signals are abundant at the central region relative to the lateral regions of the limb zeugopod. Upper panels: Immunostaining of 4-HNE (**f**) and TUNEL staining (**g**) in stage HH32 limbs. Lower panels: The relative mean values of 4-HNE (**f**) and TUNEL (**g**) signal intensities were calculated along the indicated region (lateral and central, grey box 1600 × 1000 μm²) for three biological repeats (yellow, blue, orange curves). **h**, The 4-HNE and TUNEL signal intensities at the central and lateral regions normalized to the total signal intensities. **i**, **j**, *In ovo* ferroptosis suppression impairs muscle remodelling in avian embryonic limb. Muscle fibre count (**i**) and distribution of entropy of the muscle fibre orientation (**j**) in embryonic limbs dissected from UAMC-3203-treated and vehicle control (DMSO)-treated embryos. Data represent mean ± s.d. of three biological repeats. The muscle fibre count and entropy of the muscle fibre orientations for limbs dissected from UAMC-3203-treated embryos are higher than those from the control embryos (* two-sided Wilcoxon rank-sum tests, *P* = 0.0495 and 0, respectively). Scale bars, 500 (upper panels in **a**), 300 (lower panels in **a**), 200 (**b**), 50 (**c**), 100 (**d**, **e**), and 400 (**f**, **g**) μm.

**Extended Data Table 1 | The parameters derived from model fitting in Fig. 3j–n**

| Chemical | Equation | $y_0$ | $y_M$ | $K$ | $f_1$ | $K_2$ |
|---|---|---|---|---|---|---|
| DFO | | 0 ± 1.69 | 5.83 ± 1.95 | 59.91 ± 42.6 | | |
| GKT137831 | $y = y_0 + K(y_M - y_0)/(K+x)$ | 0 ± 0.84 | 4.77 ± 1.1 | 1.35 ± 0.69 | | |
| LY294002 | | 1.92 ± 0.58 | 2.61 ± 0.72 | 28.48 ± 17.31 | | |
| FC | $y = y_0 + x(y_M - y_0)/(K+x)$ | 5.39 ± 0.26 | 4.06 ± 0.73 | 24.94 ± 12.08 | | |
| Dasatinib | $y = y_0 + (y_M - y_0)(1 - f_1 x/(K+x) - (1 - f_1)x/(K_2+x))$ | 0 ± 13.09 | 2.2 ± 13.28 | 0.006 ± 0.004 | 0.23 ± 0.54 | 31.47 ± 126.75 |

Parameters were obtained by fitting the data points to the equations indicated in Methods ("Summary of image processing and data analysis").

# Reporting Summary

Please do not complete any field with "not applicable" or n/a.  Refer to the help text for what text to use if an item is not relevant to your study.
For final submission: please carefully check your responses for accuracy; you will not be able to make changes later.

## Statistics

For all statistical analyses, confirm that the following items are present in the figure legend, table legend, main text, or Methods section.

| n/a | Confirmed | |
|---|---|---|
| ☐ | ☒ | The exact sample size ($n$) for each experimental group/condition, given as a discrete number and unit of measurement |
| ☐ | ☒ | A statement on whether measurements were taken from distinct samples or whether the same sample was measured repeatedly |
| ☐ | ☒ | The statistical test(s) used AND whether they are one- or two-sided<br>*Only common tests should be described solely by name; describe more complex techniques in the Methods section.* |
| ☒ | ☐ | A description of all covariates tested |
| ☐ | ☒ | A description of any assumptions or corrections, such as tests of normality and adjustment for multiple comparisons |
| ☐ | ☒ | A full description of the statistical parameters including central tendency (e.g. means) or other basic estimates (e.g. regression coefficient) AND variation (e.g. standard deviation) or associated estimates of uncertainty (e.g. confidence intervals) |
| ☐ | ☒ | For null hypothesis testing, the test statistic (e.g. $F$, $t$, $r$) with confidence intervals, effect sizes, degrees of freedom and $P$ value noted<br>*Give P values as exact values whenever suitable.* |
| ☒ | ☐ | For Bayesian analysis, information on the choice of priors and Markov chain Monte Carlo settings |
| ☒ | ☐ | For hierarchical and complex designs, identification of the appropriate level for tests and full reporting of outcomes |
| ☒ | ☐ | Estimates of effect sizes (e.g. Cohen's $d$, Pearson's $r$), indicating how they were calculated |

*Our web collection on statistics for biologists contains articles on many of the points above.*

## Software and code

Policy information about availability of computer code

| Data collection | Imaging data were collected with Zeiss LSM980 confocal microscope using Zen Blue software (version 3.8), and Zeiss Axio Observer 7 inverted microscope using Zen Blue software (version 3.1). Quantitative PCR was performed on a Bio-Rad CFX 96 with CFX Maestro software 2.3 (version 5.3.022.1030). Mathematical simulations were performed with MATLAB (version R2023b). Custom codes are deposited in GitHub: https://github.com/imb-lcd/ftw2024 |
|---|---|
| Data analysis | Image processing and data analyses were performed in MATLAB (version R2023b), ImageJ (version 1.54f), and Bitplane Imaris (version 10.0.1). |

For manuscripts utilizing custom algorithms or software that are central to the research but not yet described in published literature, software must be made available to editors and reviewers. We strongly encourage code deposition in a community repository (e.g. GitHub). See the Nature Portfolio guidelines for submitting code & software for further information.

## Data

Policy information about availability of data

All manuscripts must include a data availability statement. This statement should provide the following information, where applicable:
- Accession codes, unique identifiers, or web links for publicly available datasets
- A description of any restrictions on data availability
- For clinical datasets or third party data, please ensure that the statement adheres to our policy

All data are available within the Article and Supplementary Information. Microscopy data are deposited in Figshare (https://doi.org/10.6084/

m9.figshare.25762806). Additional supporting microscopy data are available from the corresponding author upon request, without any restrictions. Source data are provided with this paper.

# Research involving human participants, their data, or biological material

Policy information about studies with human participants or human data. See also policy information about sex, gender (identity/presentation), and sexual orientation and race, ethnicity and racism.

| | |
|---|---|
| Reporting on sex and gender | This study did not involve human participants. |
| Reporting on race, ethnicity, or other socially relevant groupings | This study did not involve human participants. |
| Population characteristics | This study did not involve human participants. |
| Recruitment | This study did not involve human participants. |
| Ethics oversight | This study did not involve human participants. |

Note that full information on the approval of the study protocol must also be provided in the manuscript.

# Field-specific reporting

Please select the one below that is the best fit for your research. If you are not sure, read the appropriate sections before making your selection.

☒ Life sciences       ☐ Behavioural & social sciences       ☐ Ecological, evolutionary & environmental sciences

For a reference copy of the document with all sections, see nature.com/documents/nr-reporting-summary-flat.pdf

# Life sciences study design

All studies must disclose on these points even when the disclosure is negative.

| | |
|---|---|
| Sample size | No statistical method was used to predetermine sample size. Samples sizes were determined based on previous studies with similar methodologies: Chang, J. & Ferrell Jr, J., Nature, 2013; Cheng, X. & Ferrell Jr, J., Science, 2018; Huang, J. et al., Molecular Systems Biology, 2021; Cordeiro, I. et al., Developmental Cell, 2019. |
| Data exclusions | No data exclusions. |
| Replication | All experimental results were validated by at least three independent experiments. All experimental findings are reproducible. |
| Randomization | For cell culture experiments, samples were randomized when possible. For the animal experiment in Fig. 5f, the left and right limbs of an individual animal were randomly allocated in control and experimental groups. For Fig. 5j and Extended Data Fig. 12h and 12i, animals were randomly allocated in control and experimental groups. |
| Blinding | Investigators were not blinded during data collection. However, data quantification was performed automatically using computational algorithms as described in Methods. |

# Reporting for specific materials, systems and methods

We require information from authors about some types of materials, experimental systems and methods used in many studies. Here, indicate whether each material, system or method listed is relevant to your study. If you are not sure if a list item applies to your research, read the appropriate section before selecting a response.

## Materials & experimental systems

| n/a | Involved in the study |
|---|---|
| ☐ | ☒ Antibodies |
| ☐ | ☒ Eukaryotic cell lines |
| ☒ | ☐ Palaeontology and archaeology |
| ☐ | ☒ Animals and other organisms |
| ☒ | ☐ Clinical data |
| ☒ | ☐ Dual use research of concern |
| ☒ | ☐ Plants |

## Methods

| n/a | Involved in the study |
|---|---|
| ☒ | ☐ ChIP-seq |
| ☒ | ☐ Flow cytometry |
| ☒ | ☐ MRI-based neuroimaging |

# Antibodies

| | |
|---|---|
| Antibodies used | Primary antibodies used were Erk (1:2000 for WB, Cell Signaling Technology, #9102), phospho-Erk (1:1000 for WB, Cell Signaling Technology, #9106S), 4-HNE (1:250 for IF, Abcam, ab46545 and 1:250 for IF, Abcam, ab48506), and myosin heavy chain (2 μg/mL for IF, Developmental Studies Hybridoma Bank, MF20). Secondary antibodies used were Goat anti-Rabbit IgG Alexa Fluor Plus 488 (1:500, Invitrogen, A32731), Goat anti-Mouse IgG Alexa Fluor Plus 647 (1:500, Invitrogen, A32728), and Goat anti-Rabbit IgG Alexa Fluor 568 (1:500, Invitrogen, A11036). |
| Validation | Erk antibody (Cell Signaling Technology, #9102) and phospho-Erk antibody (Cell Signaling Technology, #9106) were validated for WB in human cells on manufacturer's website (https://www.cellsignal.com/products/primary-antibodies/p44-42-mapk-erk1-2-antibody/9102 & https://www.cellsignal.com/products/primary-antibodies/phospho-p44-42-mapk-erk1-2-thr202-tyr204-e10-mouse-mab/9106, respectively). <br> 4-HNE antibody (Abcam, ab46545) is species independent and was validated for IF in a previous publication (Feng, H. et al., Cell Reports, 2020). <br> 4-HNE antibody (Abcam, ab48506) is species independent and was validated for IF on manufacturer's website (https://www.abcam.com/products/primary-antibodies/4-hydroxynonenal-antibody-hnej-2-ab48506.html). <br> Myosin heavy chain antibody (Developmental Studies Hybridoma Bank, MF20) was used in avian tissues and validated for IF in a previous publication (Bader, D. et al., Journal of Cell Biology, 1982). <br> Secondary antibodies Goat anti-Rabbit IgG Alexa Fluor Plus 488 (1:500, Invitrogen, A32731), Goat anti-Mouse IgG Alexa Fluor Plus 647 (1:500, Invitrogen, A32728), and Goat anti-Rabbit IgG Alexa Fluor 568 (1:500, Invitrogen, A11036) were validated for IF based on manufacturer's website (https://www.thermofisher.com/antibody/product/Goat-anti-Rabbit-IgG-H-L-Highly-Cross-Adsorbed-Secondary-Antibody-Polyclonal/A32731; https://www.thermofisher.com/antibody/product/Goat-anti-Mouse-IgG-H-L-Highly-Cross-Adsorbed-Secondary-Antibody-Polyclonal/A32728; https://www.thermofisher.com/antibody/product/Goat-anti-Rabbit-IgG-H-L-Highly-Cross-Adsorbed-Secondary-Antibody-Polyclonal/A-11036). |

# Eukaryotic cell lines

Policy information about cell lines and Sex and Gender in Research

| | |
|---|---|
| Cell line source(s) | hTERT RPE-1 (CRL-4000), 786-O (CRL-1932), G-402 (CRL-1440), HOS (CRL-1543), LN-18 (CRL-2610), U-118 MG (HTB-15), PANC1 (CRL-1469), MDA-MB-231 (HTB-26), HT-1080 (CCL-121), NCI-H1650 (CRL-5883), A549 (CRM-CCL-185), U-2 OS (HTB-96), A-172 (CRL-1620), Hs 895.T (CRL-7637), HeLa (CCL-2), and SH-SY5Y (CRL-2266) were obtained from ATCC. HuH-7 (JCRB0403) was obtained from JCRB Cell Bank. |
| Authentication | None of the cell lines used were authenticated. |
| Mycoplasma contamination | All cell lines were tested negative for mycoplasma contamination. |
| Commonly misidentified lines (See ICLAC register) | No commonly misidentified cell lines were used. |

# Animals and other research organisms

Policy information about studies involving animals; ARRIVE guidelines recommended for reporting animal research, and Sex and Gender in Research

| | |
|---|---|
| Laboratory animals | Fertilized Leghorn chicken eggs (stage HH30-33) were used in this study. |
| Wild animals | This study did not involve wild animals. |
| Reporting on sex | Sex-based analysis was not performed. |
| Field-collected samples | This study did not involve samples collected from the field. |
| Ethics oversight | No ethical approval was required for experiments using chick embryos at embryonic stages HH30-HH33. |

Note that full information on the approval of the study protocol must also be provided in the manuscript.

## Plants

| | |
|---|---|
| Seed stocks | *Report on the source of all seed stocks or other plant material used. If applicable, state the seed stock centre and catalogue number. If plant specimens were collected from the field, describe the collection location, date and sampling procedures.* |
| Novel plant genotypes | *Describe the methods by which all novel plant genotypes were produced. This includes those generated by transgenic approaches, gene editing, chemical/radiation-based mutagenesis and hybridization. For transgenic lines, describe the transformation method, the number of independent lines analyzed and the generation upon which experiments were performed. For gene-edited lines, describe the editor used, the endogenous sequence targeted for editing, the targeting guide RNA sequence (if applicable) and how the editor was applied.* |
| Authentication | *Describe any authentication procedures for each seed stock used or novel genotype generated. Describe any experiments used to assess the effect of a mutation and, where applicable, how potential secondary effects (e.g. second site T-DNA insertions, mosiacism, off-target gene editing) were examined.* |

