## [Peer Review file · Nature]

Manuscript Title: Emergence of large-scale cell death via trigger waves of ferroptosis

Reviewer Comments & Author Rebuttals

Reviewer Reports on the Initial Version:

Referees' comments:

Referee #1 (Remarks to the Author):

population. The focus of this study is on ferroptosis, a non-apoptotic form of cell death that is known from published work by Linkermann and colleagues and Overholtzer and colleagues to 'spread', in some manner, between cell in the form of a wave. Trigger waves are proposed to synchronize ferroptosis between cells within a population. The concept that trigger waves modulate cell death is also not entirely new, having been observed within cells during the execution of apoptosis, for example (Cheng & Ferrell, Science, 2018). The authors suggest a biochemical mechanism for the propagation of ferroptotic cell death in space and suggest that this might be important for sculpting of tissues during development, using as an example the bird limb.

The concept that trigger waves may allow for the propagation of a ferroptotic signal is intriguing, and certainly consistent with previous reports. The authors develop a clever experimental means of locally initiating ferroptosis in a population using blue light illumination. This allows for the process to be tracked more conveniently. This paper itself is clearly written. These are the strengths. However, several notable weaknesses, missing controls, and key unanswered questions are also apparent.

-Under bath treatment conditions (e.g., constant erastin application, extended data figure 1) what causes ferroptosis to initiate in some cells but not others? It would be useful to offer some insight into this question. Does the model eventually proposed by the authors allow them to predict in which cells in a population a trigger wave will initiate? This would be useful information and also validation of the modelling work.

-The authors present evidence in extended data Figure 1h that the wave-like ferroptosis mechanism can be observed in a variety of cells. It is interesting that HeLa cells exhibited this phenotype, as HeLa are generally not known to be especially sensitive to ferroptosis. What about HT-1080 cells, U-2 OS cells, or A549 cells – more classic cell models in the ferroptosis field that those reported in this study? Furthermore, were those lines reported in extended data Figure 1h all the cell lines that were tested, or only the cell lines that showed wave-like propagation amongst a larger group that were examined? Do any cell lines not show wave-like propagation? If not, why not (e.g., do they not engage some GSH/NOX molecular mechanism that is intrinsic to the proposed model?). How might that impact the interpretations in this paper? Our own incidental observations and those of others in the field is that only some cell lines exhibit wave-like ferroptosis propagation while others – which die very nicely by ferroptosis – do not exhibit this wave-like property at all.

-In establishing the main experimental model system, the concentration of ferrostatin-1 used in Extended data figure 3 (5 μM) is about 100x higher than the reported EC50 for this molecule. There is some concern this may be having unanticipated effects. It would be good to repeat this experiment and see whether lower doses of Fer-1 are protective or not, and also include additional canonical inhibitors at reasonable concentrations (liproxstatin-1, iron chelators) so as to ensure that the process being monitored conforms to the ferroptosis mechanism as classically understood and not some related but distinct mechanism of cell death induced by the blue light exposure.

-The authors invoke a ROS diffusion model to account for wave propagation. The only suitable small molecule "ROS" species with the chemical stability to diffuse longer distances might be hydrogen peroxide (H_2O_2). Does the addition of catalase to cell cultures prevent wave propagation across distance? If not, this suggests another species whose identity requires isolation and definition.

-In general, the nature of the diffusible species that mediates ROS wave propagation appears to be very ill-defined. How have the authors ruled out the possibility of a reactive lipid fragment diffusing between cells, for example? Or release of iron from dying cells that then can induce peroxidation in neighboring cells? These and many other mechanisms apart from the one proposed by the authors could be envisioned. All of these models need to be rigorously tested using chemical but also genetic manipulations.

-In figure 3a, it is hard to understand how the authors see the process of lipid peroxidation (LOOH) contributing to the formation of soluble hydrogen peroxide (blue box in 3a). Presumably, the authors envision that H_2O_2 diffusing between cells can initiate lipid peroxidation in a neighboring cell. How plausible is this? Under the conditions the authors employ, is direct application of H_2O_2 in a localized manner sufficient to trigger "ferroptotic waves", as would be predicted from the model?

-GPX4 inhibition are canonical inducers of ferroptosis. Does RSL3 or a similar covalent GPX4 inhibitor induce the same trigger wave phenomenon in RPE1 and other cells? If not, what might this imply?

-What is the effect of manipulating the expression of well-known ferroptosis effectors, like GPX4 or FSP1, on wave propagation? The authors focus on NOX enzymes but appear to ignore the role of more well-known and potentially universal regulators of ferroptosis.

-Relevant to Figure 4 and the modelled bistable switch, the concept is intriguing but additional controls are needed. Perhaps erastin, between two concentration ranges (e.g., 0.63 and 1.25 μM), simply has non-linear effects on target function, more effectively inhibiting cystine uptake and/or depleting intracellular thiol-containing metabolites. It would be important to know whether or not

-I am unclear whether the data reported in Figure 5h are from 'normal' physiological remodeling/death events, or PALP/laser-induced death in the isolated limb model. If the latter, it is unclear what relevance this has. One would like to test whether a reasonable concentration of a ferroptosis inhibitor (or genetic overexpression of a ferroptosis inhibitor like GPX4) is sufficient to prevent cell death in this model undergoing cell death without any external and artificial induction.

-Relevant to the wave-like cell death in developing avian limbs, what is the natural trigger for this process? What is known about this developmental event and how could it be related to the induction of ferroptosis? It would also be good to experimentally test whether inhibition of apoptosis or other forms of cell death in this context do or do not prevent cell death. The working assumption for most people would be that apoptosis accounts for most developmental cell death, so the concept that ferroptosis may underlie this massive developmentally regulated event may require some stronger confirmation and additional controls to rule out other possible mechanisms.

-Some very high-profile research suggests that ferroptosis is inhibited at high cell density (Wu et al, 2019, Nature). The authors should perhaps comments on how this may be related to their model and to the concept of ferroptosis wave-like propagation if at high cell density the process is inhibited. Have the authors examined wave propagation at various cell densities including especially high cell densities? In vivo, how can the authors account for wave propagation in (presumably) tightly packed cells?

Minor:

-It is unclear what cells are being examined in extended data figure 1a and b.

-Extended data Figure 2. What does a “technical repeat” mean in this context? Measuring the same image multiple times?

-It would be important to demonstrate that all small molecule inhibitors used here are not acting as RTAs except when that is the known/desired effect. E.g., is the PI3K or TK inhibitor an RTA at the concentrations employed, and could this explain the observed effects?

-Figure 5f the correct formula to use is oxidized / (reduced + oxidized) to normalize for probe uptake. These data need to be re-analyzed and, possibly, this will alter the interpretation.

-Quantification of data from multiple limbs/animals in Fig 5a,b seems warranted.

-The avian limb experiments are intriguing but preliminary. 50 μ M Fer-1 is a very high concentration of inhibitor to use.

Referee #2 (Remarks to the Author):

In this interesting paper, the authors have discovered that ferroptosis spreads from cell to cell in culture via trigger waves, which spread over substantial distances (millimeters) without slowing or diminishing in amplitude. This phenomenon can be seen in several different cell types and in response to a variety of ferroptotic induction strategies. The waves can propagate across a physical gap of up to about 150-200 μm , showing that some diffusible species is responsible for the local coupling of the cells, and inhibitor studies show the species is probably some ROS. The authors then characterize various perturbations that can make the wave speed faster (e.g. Fe^{3+} -citrate, ERK2 overexpression) or slower (NOX inhibitor, PI3K inhibitor, Src/Abl inhibitor), which implicates various known mediators of ferroptosis in the positive feedback that underpins the trigger waves. Finally, they show that ferroptotic cell death occurs in the developing avian limb bud, and that it appears to spread at speeds comparable to those seen in cell culture.

Figures 1-3 and the related videos and supplementary figures are terrific. They convincingly establish that ferroptosis spreads via trigger waves, show that a diffusible species is responsible for the local coupling, and implicate various regulators in wave generation. The work is original, convincing, thorough, and important. Figure 4 is a bit more problematic. I think there are other equally plausible explanations for how photoinduction changes the system to make it better able to generate/propagate ferroptotic waves. And Figure 5, which argues that the cell culture work is relevant to limb remodeling in chick development, may be important, but I do not know the field well enough to assess it adequately. I would gladly defer to a developmental biologist on this point.

Major criticisms:

1. Fig 4. I am not completely convinced of the authors' interpretation of this experiment. Isn't it possible that the system is bistable even prior to photoinduction, but the cells are just more stuck on the lower stable branch of the S-shaped response curve? Perhaps the photoinduction pulls the unstable branch down toward the stable lower branch rather than changing a single-valued response function to a multi-valued response function. This would be another way that ferroptotic stress could, in principle, prime the cells to propagate ferroptotic trigger waves.

2. Fig. 5 is beyond my textbook-level understanding of the roles of cell death in development. My impression had been that phenomena like limb remodeling are due to coordinated apoptosis, but perhaps, as the authors imply, in at least some instances it is due to ferroptosis instead. Certainly there is massive cell death in the limb, and it is accompanied by lipid peroxidation, as the authors have convincingly demonstrated.

But as to the issue of whether the limb remodeling presented here is spreading from cell to cell via trigger waves, I'm not certain. The authors make some estimates of speeds, and the speeds do fit reasonably well with what is seen in culture, but overall this does not look like such a clear-cut example of a constant-speed front of cell death. Maybe it is starting from multiple foci, making it hard to define wave fronts, or maybe flows as well as diffusion are involved in the local coupling. Perhaps the authors could sum up in what ways the process in the limb resembles the ferroptotic spread seen in culture, and in what ways it differs. I do think it is admirable and useful that the

authors have gone the “next step” from cell culture to in vivo development, but I am not sure how similar the in vivo and cell culture phenomena are to each other.

Minor criticisms:

3. Line 27: Please spell out what xCT stands for (cystine-glutamate transporter).

4. Line 49: I suggest replacing “had remained unknown” with “is unknown”.

5. Line 67: Please mention that you are looking at several different cell types in Supplementary Video 1—I think that is an important point to convey in the text.

6. Fig 2: Please mention in the text that the gaps in the sheets of cells were produced by scratching with glass needles. I know it’s in the methods section, but I think it’s helpful for the reader to know without having to search for the info.

7. Line 131: The estimate for the diffusion coefficient of the ferroptotic ROS signal across the cell gap assumes there is no degradation of the ROS species. Is this a reasonable assumption? If not, perhaps the authors should just skip this analysis, or repeat the analysis with a range of plausible degradation rates in a reaction/diffusion model.

8. Extended Data Figure 10. Please indicate what-is-what better in panel A. E.g., is the top blot probed with ERK1/2 antibodies? Is the second blot probed with pERK1/2 antibodies? Does ERK2-PA co-migrate with ERK1?

Referee #3 (Remarks to the Author):

In this work, the authors define mechanisms of large-scale, propagating waves of cell death triggered via ferroptosis. Several cell culture assays support the range of speed of these waves, their reliance on ROS for propagation, including across gaps in cell contact, and their experiments support a threshold at which ROS steady state levels bifurcate from monostable to bistable states.

This reviewer was specifically asked to comment on Figure 5 and the ferroptosis propagation in limb development. The data presented look convincing. A detail as a note to authors is that the Nile Blue pictures in Figure 5a appear to be viewed dorsally and the rest of the pictures show the opposite outline (ventral view?); it would be helpful to show the Nile Blue from the same aspect as the rest of the data as it is whole mount, and to define this.

The most well known spatially resolved region of cell death in the limb occurs in the interdigital mesenchyme, which is not examined here. As it is such a well studied and clear phenomenon where the entirety of the tissue disappears, the reviewer can't help but wonder if this was examined. It may be appropriate to at least comment on this. The authors show distal to proximal cell death that appears to begin at or near the central carpal region and extend proximally at least through the zeugopod according to the authors' picture. John Saunders' 1966 publication was not accessible to this reviewer during the review period (requiring archival access, not online) but other publications note four areas of cell death in chick limbs: the anterior necrotic zone, posterior necrotic zone, interdigital necrotic zone and the opaque patch (Fernandez-Teran, et al, Dev Dyn, 2006; Montero, et al, Dev Dyn, 2021). One presumes this may be the opaque patch? It will be important that the authors put this cell death area in the context of accessible literature and established limb developmental biology and further define this. Importantly also, this reviewer is unaware of anything in the central chick limb that 'disappears' via cell death. The interdigital mesenchyme, of course, leads to the complete disappearance of the tissue between the digits, but the data this reviewer found (only Fernandez-Teran showed scant data at HH31-32) did not suggest cell death across a continuous swath of tissue in the central limb. The radius and ulna initiate as separate cartilage condensations and the tissue between the radius and ulna does not disappear, leaving this reviewer wondering how the chick limb accommodates such a large swath of continuous cell death in the center of the limb, leaving no empty space. It is encouraged that the authors contextualize this cell death further for the readers.

**Author Rebuttals to Initial Comments:
Overall responses to all reviewers**

We thank each of the reviewers for their careful assessments and constructive suggestions that have helped improve the quality of our work. All three reviewers have noted our finding that **ferroptosis propagates across cell populations as trigger waves** as a central and novel aspect of our study. We further extended this discovery to answer a century-old question in development, i.e., how does large-scale cell death occur to eliminate temporary structures in shaping tissues and organs into their appropriate forms? Our study has uncovered ferroptosis waves in the developing limb and their functional role in muscle remodeling. We have now added substantial amounts of **new** data to the revised manuscript and altered the text to address the reviewers' concerns and comments.

Major changes include:

A. Data and clarifications related to ferroptosis occurrence and its functional role during limb development (Reviewers 1, 2, and 3). In the original manuscript, we described ferroptosis and its propagation at the ectodermal layer of the embryonic limb (new Fig. 5e). In our revised manuscript, we have further investigated this phenomenon in the context of muscle remodeling during limb development.

Large-scale cell death serves as a critical regulatory mechanism for muscle remodeling during limb development¹. During this remodeling process, muscle masses are sculpted by cell death to form the anatomically distinct muscle bellies of the foot, shank and thigh^{1,2}. At day 6.5, foot muscles are initially continuous with the shank muscles (6.5-day limb, Figure R1a, also new Extended Data Fig. 18a). From day 7 onwards, the foot muscles progressively separate from the shank (7.5-day limb, Figure R1a, also new Extended Data Fig. 18a). This separation is facilitated by massive cell death occurring at the muscular ventral death zone (MVDZ) along the proximo-distal axis of the limb zeugopod (Figure R1b, also new Extended Data Fig. 18b). We now provide **new** evidence supporting the functional role of ferroptosis and its propagation in this muscle remodeling process during limb development.

Figure R1. Ferroptosis is involved in muscle remodeling during avian limb development. **a**, Whole-mount immunostaining of myosin heavy chain in the avian limb at days 6.5 and 7.5 of development. Ventral views of the limbs are shown, with limb margins outlined in yellow. Lower panels: Zoomed-in views of the gray boxes in the upper panels. The foot and shank muscles are labeled. **b-e**, Longitudinal sections of limbs at day 7.8 of development. **b**, Co-staining of myosin heavy chain and TUNEL, showing the muscular ventral death zone (MVDZ) in the zeugopod area. **c**, Co-staining of TUNEL and 4-HNE at the MVDZ in **(b)**. **d**, **e**, Co-immunostaining of myosin heavy chain and 4-HNE at the foot **(d)** and shank **(e)** regions. Degenerating muscles exhibit a rounded and beaded appearance.

A.1 Cell death at MVDZ involves ferroptosis. Previous study has attributed massive cell death at the MVDZ to apoptosis (so, previously, it was termed the muscular ventral apoptosis zone, MVAZ) ¹ based on TUNEL staining data. However, TUNEL staining has recently been shown to also detect programmed necrotic cell death, including ferroptosis ^{3,4}. In fact, we have found that this massive cell death event co-localizes with the signal of a ferroptosis indicator, 4-Hydroxynonenal (4-HNE) (Fig. R1c, also new Extended Data Fig. 18c), supporting the involvement of ferroptosis in this large-scale muscle cell death process.

A.2 Degenerating muscle cells co-localize with the ferroptosis indicator 4-HNE. Our new data shows that degenerating muscle cells in the foot and shank muscle bellies largely (~80%) co-localize with 4-HNE signals (Fig. R1d, e, also new Extended Data Fig. 18d, e).

A.3 Ferroptosis encompasses both the muscular layer and the ectodermal layer of the limb zeugopod. We found that 4-HNE signals not only localize at the muscular layer, but also extend to the ectodermal layer of the zeugopod (transverse section of the embryonic limb in new Fig. 5a).

A.4 Large-scale ferroptosis occurs at the central ectodermal layer of the limb zeugopod. We found abundant 4-HNE signals extensively localized in the central ectodermal layer of the zeugopod (new Fig. 5b, c), representing the area where we also observed higher levels of oxidizable lipids (Fig. 5g-i) and large-scale cell death propagation (new Fig. 5e). We now further show that this cell death propagation is ferroptosis-specific, since the cell death is significantly suppressed by the ferroptosis inhibitors Fer-1, DFO and UAMC-3203, but not by the apoptosis inhibitor Z-VAD-FMK (new Fig. 5f).

A.5 Suppression of ferroptosis impairs the muscle remodeling process. *In ovo* injection of a ferroptosis inhibitor (UAMC-3203) resulted in muscle remodeling defects, manifesting as excessive muscle fibers, compromised muscle mass segregation, and fiber disorganization (new Fig. 5j, also new Extended Data Fig. 20). These observations support the involvement of ferroptosis in regulating the number of muscle fibers and the subsequent individualization of the muscle mass during limb development.

Together, these new results place our original findings in an important context, i.e., that ferroptosis and its propagation facilitates the emergence of large-scale cell death in the muscular and ectodermal layers of the limb during the process of muscle remodeling. Our findings raise an interesting possibility that ectodermal cell death, mediated by wave-like ferroptotic propagation, facilitates muscle differentiation and muscle mass segregation. We have now incorporated and discussed these results in the revised manuscript (pages 13-16, lines 300-374).

B. Clarifications related to the conventional view of apoptosis-mediated developmental cell death (Reviewers 1, 2, and 3). We have made changes to the manuscript text to clarify how our findings indeed challenge the conventional view of apoptosis being the predominant mode of developmental cell death. Nevertheless, we also wish to emphasize that our results align with several previous studies (see below) hypothesizing the existence of an **as-yet-unknown cell death mechanism** that can act collaboratively with apoptosis to mediate cell death during embryonic development. In fact, our study convincingly demonstrates this possibility by establishing a **functional role for ferroptosis** in muscle remodeling during limb development. These previous studies (elaborated in detail below) proved instrumental in providing a framework for our investigation of ferroptosis during embryonic development.

Using interdigit regression as a classical example, it has been proposed that apoptosis mediates interdigital cell death during limb development ⁷. However, several previous studies have also suggested the possibility that other forms of cell death, alongside apoptosis, may participate in interdigit regression. First, knockout of two major apoptosis regulatory genes (BAX and BAK) only partially reduced interdigital regression in mice and elicited mild syndactyly phenotypes ⁸. Second, using transmission electron microscopy, a previous study showed that dying cells in the interdigital tissue during embryonic development exhibit different types of cell death, as evidenced by the mixture of cells

displaying apoptotic and necrotic morphological characteristics⁹. More recently, it has been found that mice completely lacking apoptosis (BOK/BAX/BAK triple-knockout mice) can survive until adulthood¹⁰. Interdigit regression in these surviving mice is apparently present, and organs thought to rely on apoptosis for morphogenesis largely appear normal (Fig. 7 in the respective publication)¹⁰, without compensatory occurrence of necroptosis, pyroptosis, or autophagy pathways (Fig. S3A & S3B in the respective publication)¹⁰. These observations are indicative that an as-yet-unknown mechanism may act collaboratively and/or synergistically with apoptosis to mediate cell death for interdigit regression and organogenesis.

In our current study, we chose to focus on a different developmental event that involves large-scale cell death, i.e., muscle mass remodeling during limb development. In this specific developmental event, a mixture of different forms of cell death is also evident. Degenerating muscles appear to co-localize with either caspase-3 (17%) (Fig. R2) and/or 4-HNE (~80%) (Fig. R1d, e, also new Extended Data Fig. 18d, e), implying the involvement of both apoptosis and ferroptosis during muscle remodeling of the limb. Thus, contrary to the conventional view that apoptosis is the only prominent form of cell death during embryonic development, different types of cell death may co-exist with apoptosis to mediate developmental cell death events. Although our findings evidence the involvement of ferroptosis in muscle remodeling, we do not rule out the possibility that yet other types of cell death can co-occur with ferroptosis. Accordingly, our study opens up new avenues for future investigations of the roles of other forms of cell death and their possible co-occurrence during development. In our revised manuscript, we now mention the emerging notion that modalities of cell death other than apoptosis can also be involved in embryonic development for a broader readership (page 17, lines 397-401).

Figure R2. Apoptosis is also involved in muscle remodeling. a, Co-immunostaining of cleaved caspase-3 and myosin heavy chain at the foot region. A longitudinal section of the avian limb at day 7.8 of development.

C. Further characterization of the generality of ferroptosis trigger waves (Reviewer 1). Reviewer 1 suggested that we include more data to demonstrate the generality of ferroptosis waves, thereby addressing a controversial issue in the field of ferroptosis. Accordingly, we have now added the following new data:

C.1 Examination of additional cell lines for ferroptosis trigger waves. We have now tested additional cell lines and found that **all** 17 cell lines from 12 different tissues of origin, including those mentioned by the reviewer (HT-1080, U-2 OS, A549), exhibit wave-like ferroptosis propagation (new Extended Data Fig. 1j, k, also Supplementary Video 1). We now further elaborate on possible factors that may preclude observation of ferroptotic waves in some of the cell lines mentioned by the reviewer (see our detailed response to **Reviewer 1-P2**).

C.2 Assessment of an additional ferroptosis inducer and an effector for their effects on ferroptosis trigger waves. We have now characterized the cell death kinetics induced by erastin and RSL3 (new Extended Data Fig. 1c-f), and provided our rationale for the observed differences. We have also modulated the activity of a well-known ferroptosis effector, FSP-1, and observed its impact on the progressivity of ferroptotic trigger waves (new Extended Data Fig. 14).

D. Characterization of the diffusive molecule for ferroptosis propagation (Reviewer 1). It has been proposed previously that diffusive cytotoxic molecules, such as lipid peroxides and their byproducts (e.g., lipid electrophiles with a peroxide group), are released from ferroptotic cells^{11,12}. We have now

further characterized the diffusive molecule and added the following **new** data (see our detailed response to **Reviewer 1-P4**):

D.1 The diffusive molecule is a type of ROS. Addition of antioxidants to scavenge lipid peroxides (Trolox, Fer-1), superoxide radicals, hydroxyl radicals and hydrogen peroxide (Tiron, TEMPO, NAC) effectively halted ferroptosis propagation (new Extended Fig. 6e). Furthermore, we now show that conditioned media containing the diffusive molecule from ferroptotic cells elicits ferroptotic cell death in naïve untreated cells. Pre-treatment of the conditioned media with different ROS scavengers suppressed the resulting cell death, indicating that this diffusive molecule is a type of reactive oxygen species (new Extended Data Fig. 8a).

D.2 The diffusive molecule is not hydrogen peroxide. We rule out hydrogen peroxide (H₂O₂), a chemically stable small molecule ROS, as being the diffusive ROS molecule based on a centrifugal filtration experiment (new Extended Data Fig. 8b).

According to our current characterization and the results of previous studies ^{11,12}, we suspect that the diffusive ROS molecules underlying ferroptosis wave propagation are likely peroxidized lipids and/or their byproducts, representing a promising avenue for further detailed research.

E. Clarifications related to the interpretation of photoinduction experiments for detecting ROS bistability (Reviewer 2). Reviewer 2 presents an equally valid interpretation of our photoinduction experiment, i.e., in the absence of ferroptosis stress, the cellular redox state can be bistable, but with a greater threshold to overcome the switch to the upper ROS steady state. Our **new data** now shows that this alternative scenario is less likely. ROS remain at low steady states even after applying the maximum light intensity (new Extended Data Fig. 16) and high concentrations of exogenous H₂O₂ (Fig. R16).

F. Other key changes

F.1 We have added **new data** from experiments and mathematical modeling to show that the initiation of ferroptosis is a random process (new Extended Data Fig. 2c-f, also Fig. R3; **Reviewer 1-P1**).

F.2 We show that enhancing the Fenton-mediated ROS feedback loop sensitizes cells at high confluency to ferroptosis, facilitating population-wide ferroptosis waves (new Extended Data Fig. 13; **Reviewer 1-P2**).

F.3. We have significantly revised the manuscript text to incorporate the reviewers' comments and to enhance clarity and logical flow.

Below are our point-by-point responses to each of the reviewers' comments.

Referee #1 (Remarks to the Author):

This manuscript by Chen and colleagues examines how cell death propagates between cells within a population. The focus of this study is on ferroptosis, a non-apoptotic form of cell death that is known from published work by Linkermann and colleagues and Overholtzer and colleagues to 'spread', in some manner, between cell in the form of a wave. Trigger waves are proposed to synchronize ferroptosis between cells within a population. The concept that trigger waves modulate cell death is also not entirely new, having been observed within cells during the execution of apoptosis, for example (Cheng & Ferrell, Science, 2018). The authors suggest a biochemical mechanism for the propagation of ferroptotic cell death in space and suggest that this might be important for sculpting of tissues during development, using as an example the bird limb.

The concept that trigger waves may allow for the propagation of a ferroptotic signal is intriguing, and certainly consistent with previous reports. The authors develop a clever experimental means of locally initiating ferroptosis in a population using blue light illumination. This allows for the process to be tracked more conveniently. This paper itself is clearly written. These are the strengths. However, several notable weaknesses, missing controls, and key unanswered questions are also apparent.

We appreciate reviewer 1 for the insightful comments and suggestions that have helped to enhance the quality of our work. We have performed additional experiments and analyses to address the points raised. In particular, we now provide new results to address the dispute in the field regarding the generality of ferroptosis propagation in different cell types and ferroptosis inducers. We have also provided new data and demonstrate the utility of our mathematical model to: (1) explain the difference between the ferroptosis inducers erastin and RSL3, and (2) cell death initiation occurrence. Moreover, we provide new experimental evidence to support the role of ferroptosis in muscle remodeling during limb development. By adopting this reviewer's helpful suggestions, we think that our manuscript is now significantly improved in terms of depth and rigor. We also wish to clarify how our work builds upon, but differs from, previous studies.

Our finding that ferroptosis propagates across cells as trigger waves holds biological meaning and implications that are markedly distinct from a previous study on apoptosis trigger waves (Cheng & Ferrell, *Science*, 2018)¹³. In their study, apoptotic signal was shown to propagate in frog egg extract and single-cell frog eggs. They found that the trigger wave mechanism allows apoptotic signal to constantly spread **intra-cellularly in a single-cell frog egg**, the size of which is too large for apoptotic signal spread to be explained by simple diffusion. Although that study has indeed been instrumental in identifying the mechanism of trigger waves, whether cell death can **propagate inter-cellularly across cell populations** as trigger waves had remained unknown. Therefore, unlike that study, our work demonstrates ferroptotic cell death propagation **across a cell population of millions of cells**, indicating the possibility of catastrophic tissue- and organ-level cell death under oxidative stress. These findings may potentially explain the large-scale cell death events observed in various ferroptosis-related pathological conditions, with our study revealing a system-level mechanism for their occurrence.

Despite previous observations by Linkermann and colleagues¹⁴ and Overholtzer and colleagues¹⁵ of the wave-like behavior of ferroptosis, the regulatory principles that underpin this phenomenon have remained elusive. Linkermann and colleagues found that renal tubules can undergo synchronized cell death through ferroptosis upon ischemia. Overholtzer and colleagues reported the interesting phenomenon of ferroptosis-mediated pore formation, an event that precedes cell rupture and transmission of cell death¹⁵. Extending beyond these studies, **our work reveals a system-level mechanism underlying ferroptosis propagation, i.e., trigger waves** that allow signal propagation with constant speed and intensity over long distances. Due to the challenge of obtaining time-resolved, long-distance quantification of ferroptosis, previous studies have only examined ferroptosis propagation over short distances, making it impossible to distinguish between diffusion and trigger waves. Similarly, it has been shown that in other types of cell death, diffusive death signals such as cytotoxic molecules (e.g., calcium)¹⁶ or secreted signaling factors (e.g., TNF- α)¹⁷ can be transmitted from cell to cell in a process known as the bystander effect. Therefore, to definitively distinguish between the spread of ferroptosis and diffusive bystander effects requires dedicated experimental designs and quantitative measurements. We have overcome this challenge by developing a photo-inducible ferroptosis cell death assay for large-scale measurement of ferroptosis propagation allowing us to show quantitatively that ferroptosis propagates as trigger waves. Furthermore, we demonstrate the existence of a **cellular redox bistable switch**, representing an emergent property of the ROS feedback loops that drive ferroptosis trigger waves. Together these findings yield a **conceptual departure from the long-held model of diffusion-mediated transmission of cell death signals**.

Finally, our discovery of the involvement of ferroptosis and its propagation during limb development provides the first evidence of how ferroptosis trigger waves can be utilized to sculpt tissues/organs during embryonic development. Unlike apoptosis, which was initially identified as being responsible for cell trimming during development¹⁸, ferroptosis trigger waves can be harnessed to induce large-scale cell death during embryogenesis. We suspect that this type of tissue-sculpting process is especially relevant for complex organisms whose development relies on spatial coordination of developmental signals (e.g., long-distance spread of morphogens). Our findings open up new avenues for future investigations on the interplay between developmental signals and ferroptosis. Consequently, we strongly believe that the conceptual advances presented in our work extend significantly beyond earlier studies.

P1. Under bath treatment conditions (e.g., constant erastin application, extended data figure 1) what causes ferroptosis to initiate in some cells but not others? It would be useful to offer some insight into this question. Does the model eventually proposed by the authors allow them to predict in which cells in a population a trigger wave will initiate? This would be useful information and also validation of the modelling work.

We thank reviewer 1 for these questions that prompted us to investigate further the cause of ferroptosis initiation. Biochemical reactions are usually noisy due to the probabilistic nature of molecular interactions¹⁹. Accordingly, it is possible that the level of Fe^{2+} , as well as NOX activity, are subjected to biochemical noise. This possibility is consistent with our observation that Fe^{2+} and ROS levels in a cell population follow a continuous probability distribution (best fit with the logistic distribution, new Extended Data Fig. 2e, f). Considering that the redox state of a cell population follows a probability distribution, cells with higher ROS levels, i.e., bypassing the bistability threshold, will undergo a bistable switch to their upper steady states and ultimately initiate ferroptosis. If this scenario holds true, we would expect initiation of ferroptosis to be similar to a random process.

To assess if the occurrence of ferroptosis initiation is a random process, we carried out new experiments to quantify: (1) the number of ferroptosis initiation events over time; and (2) the time interval between two consecutive ferroptosis initiation events. If initiation of ferroptosis is a random process, we expected that the probability density functions of ferroptosis initiation and the time interval between consecutive initiation events would present a Poisson distribution and geometric distribution, respectively²⁰. Accordingly, we developed an imaging analysis algorithm to automatically identify ferroptosis initiation events over time (see Methods). Based on the identified time series of ferroptosis initiation events (a total of 761 ferroptosis initiation sites across 756 positions of 1.26 X 1.26 mm² in size, new Extended Data Fig. 2b), we calculated the distributions of ferroptosis initiation events and of the time interval between consecutive initiation events, which as anticipated follow a Poisson distribution and a geometric distribution, respectively (new Extended Data Fig. 2c, d), indicating that initiation of ferroptosis is a random process.

To further test this hypothesis using mathematical modeling, we simulated ferroptosis initiation events in a cell population where the cellular redox state follows a logistic distribution. Consistent with our experimental results, ferroptosis initiation and the time interval between consecutive initiation events follow a Poisson distribution and a geometric distribution, respectively (Fig. R3). In summary, both our experimental results and *in silico* simulations support the likelihood that ferroptosis initiation is a random process, likely due to the stochastic nature of biochemical reactions in the redox system. We now describe these results in page 4, line 90-95 of the revised manuscript.

Figure R3. *In silico* simulations of ferroptosis initiation. **a**, Simulated time series of ferroptosis initiation events from 756 positions with areas of 1.26 X 1.26 mm². **b**, Distribution of the number of ferroptosis initiation events over a time period of 5 h with its fit to a Poisson distribution (mean = 1.17 initiation events in 5 h, p-value of the two-sample Kolmogorov-Smirnov test = 1, indicating no significant difference from the Poisson distribution). **c**, Distribution of the time interval between two consecutive ferroptosis initiation events with its fit to a geometric distribution (mean = 4.28 h, p-value of the two-sample Kolmogorov-Smirnov test = 0.13, indicating no significant difference from the geometric distribution).

P2. The authors present evidence in extended data Figure 1h that the wave-like ferroptosis mechanism can be observed in a variety of cells. It is interesting that HeLa cells exhibited this phenotype, as HeLa are generally not known to be especially sensitive to ferroptosis. What about HT-1080 cells, U-2 OS cells, or A549 cells – more classic cell models in the ferroptosis field that those reported in this study? Furthermore, were those lines reported in extended data Figure 1h all the cell lines that were tested, or only the cell lines that showed wave-like propagation amongst a larger group that were examined? Do any cell lines not show wave-like propagation? If not, why not (e.g., do they not engage some GSH/NOX molecular mechanism that is intrinsic to the proposed model?). How might that impact the interpretations in this paper? Our own incidental observations and those of others in the field is that only some cell lines exhibit wave-like ferroptosis propagation while others – which die very nicely by ferroptosis – do not exhibit this wave-like property at all.

We appreciate these important questions raised by reviewer 1. We share the same view that the generality of ferroptosis propagation in different cell types remains a debated topic in the field. A previous study by Overholtzer and colleagues¹⁵ examined the occurrence of ferroptosis propagation in five cell lines. To address the reviewer's comment, we have investigated the prevalence of ferroptosis propagation by testing its occurrence in **17 different cell lines from 12 tissues of origin**, including those mentioned by the reviewer (i.e., HeLa, HT-1080, U-2 OS and A549). We identified two critical factors that dictate whether ferroptotic waves occur in those cell lines: A) cell density; and B) sensitivity to ferroptosis stress. By tuning these variables, we observed ferroptotic waves in **all** of the cell lines we examined. We have now added movies for the 17 different cell lines (Supplementary Video 1), and mention these results in page 3, lines 71-74. We further elaborate on these findings below:

Figure R4. Density-dependent ferroptosis waves in A172 cells. Representative time-lapse images at the indicated time points after erastin treatment (10 μ M) are shown. A172 cells were seeded at three different densities: 1.5×10^3 cells/cm² (low density, upper panel), 3×10^4 cells/cm² (medium density, middle panel), and 9×10^4 cells/cm² (high density, lower panel) two days before erastin treatment. Cell death is indicated by increased nuclear dye fluorescence signal. The white outlines represent the boundaries of cell death areas.

A. Cell density-dependent occurrence of ferroptosis waves

As mentioned by reviewer 1 (in **P11**), ferroptosis is suppressed at high cell densities (Wu et al., 2019, *Nature*)²¹. Consistently, we also observed a profound effect of cell density on the occurrence of ferroptosis waves. Using the glioblastoma cell line A172 as an example, we observed that ferroptosis displayed wave-like behavior at a medium cell density, eliminating the majority of the cell population (middle panel, Fig. R4). In contrast, at a low density, cell death occurred in a cell-autonomous and sporadic manner, likely due to there being insufficient adjacent cells for ferroptosis to propagate (upper panel, Fig. R4). As expected, ferroptosis was suppressed at a high cell density (lower panel, Fig. R4). These results demonstrate the important influence of cell density on the occurrence of ferroptosis waves. Below, we further elaborate on the effect of density in terms of the specific cell lines mentioned by the reviewer (i.e., HeLa, U-2 OS and A549).

1. HeLa cells

As mentioned by reviewer 1, HeLa cells are resistant to ferroptosis. Indeed, when we seeded HeLa cells at a high density they maintained their viability under erastin treatment (lower panel, Fig. R5). In contrast, cells seeded at a low density exhibited wave-like cell death propagation across cell populations (upper panel, Figure R5), indicating density-dependent occurrence of ferroptosis waves.

Figure R5. Density-dependent ferroptosis waves in HeLa cells. Representative time-lapse images at the indicated time points after erastin treatment (10 μ M) are shown. HeLa cells were seeded at two different densities: 1 $\times 10^4$ cells/cm² (low density, upper panel), and 2 $\times 10^4$ cells/cm² (high density, lower panel) two days before erastin treatment. Cell death is indicated by increased nuclear dye fluorescence signal. The white outlines represent the boundaries of cell death areas.

2. A549 and U-2 OS cells

The occurrence of ferroptosis waves in A549 cells also depends on cell density. Ferroptosis is suppressed at high cell densities ($>8 \times 10^3$ cells/cm²) (Fig. R6a). However, at lower cell densities, we observed colony-scale propagation of ferroptosis (Fig. R6b, also new Supplementary Video 1b). Similarly, for U-2 OS cells, we also observed ferroptosis propagation across individual colonies (Fig. R6c, also Supplementary Video 1b). Since this colony-scale propagation was only observed to occur in $\sim 30\%$ of the A549 colonies, and for $\sim 40\%$ of the U-2 OS colonies, a bigger field of view covering a larger number of colonies and a fine-scale time resolution (time interval ≤ 1 hour) were needed.

Figure R6. Colony-scale propagation of ferroptosis in A549 and U-2 OS cells.

a, Representative nuclear dye fluorescence images of A549 cells seeded at different densities (from left to right: two-fold increase from 1 $\times 10^3$ cells/cm² to 1.6 $\times 10^4$ cells/cm²) 57 h after erastin (10 μ M) treatment. **b-c**, Representative time-lapse images of cell death in A549 (**b**) and U-2 OS (**c**) cell colonies after erastin treatment.

B. Sensitivity to ferroptosis stress

Cell-autonomous sensitivity to ferroptosis stress affects the occurrence of ferroptosis waves. As detailed below, resistance to ferroptosis can preclude observation of ferroptosis waves in certain cell lines under some conditions (e.g., A549 and U-2 OS at high cell densities). However, the strong sensitivity of HT-1080 cells to ferroptosis renders it difficult to observe ferroptosis waves in this cell line.

1. A549 and U-2 OS cells

Despite the fact that ferroptosis waves can be observed in A549 and U-2 OS cells, cell death propagation is limited to some individual colonies and not the entire cell population (Fig. R6). It is possible that certain ferroptosis-resistant cells have relatively weaker ROS feedback loops (e.g., due to low cellular iron). Thus, enhancing the strength of the ROS feedback loop may induce population-wide propagation of ferroptosis, even at a high cell density. We explored this possibility by modulating the Fenton-mediated ROS feedback loop through iron supplementation (ferric citrate, FC) of A549 and U-2 OS cells in combination with erastin treatment. Whereas erastin treatment alone did not elicit cell death, FC supplementation (125 μM) plus erastin treatment resulted in population-wide propagation of ferroptosis in A549 and U-2 OS cells (new Extended Data Figure 13).

Echoing the hypothesis of the reviewer, these more ferroptosis-resistant cell lines, i.e., A549 and U-2 OS, possibly display lower-level ROS production from the Fenton reaction. Thus, enhancing their ROS feedback loop sensitizes the cells to ferroptosis, allowing the occurrence of ferroptosis waves. We have now incorporated these results and discussed this point in page 9, lines 208-216.

We suspect that there are many other ways to induce ferroptosis propagation in cell lines that are typically resistant to ferroptosis. Other ferroptosis regulators, including FSP1²² and GPX4²³, whose activities render cells resistant to ferroptosis, can also be downregulated to sensitize cells and induce ferroptosis propagation. We address this point further in our responses to **P6 & P7**.

2. HT-1080 cells

We observed ferroptosis spread in populations of HT-1080 cells at a medium density (upper panel, Fig. R7, also Supplementary Video 1a). At this density, multiple cell death initiation sites occur, which limits the time and distance for observing cell death propagation. Consequently, to observe ferroptosis waves in HT-1080 cells, an appropriate time resolution ($\leq 1\text{h}$) for imaging is required. The presence of multiple cell death initiation sites may be attributable to the inherently higher sensitivity of HT-1080 cells to ferroptosis²³. When HT-1080 cells were seeded at a low density (lower panel, Fig. R7), we observed sporadic cell death but not ferroptosis waves, i.e., similar to the low-density condition determined for A172 cells (upper panel, Fig. R4). The lack of ferroptosis waves is likely due to there being insufficient neighboring cells for cell death to propagate. This scenario can partly explain why ferroptosis spread has not been reported previously for HT-1080 cells since previous studies deployed relatively low cell densities (e.g., $\sim 5.5 \times 10^3$ cells/cm² and $\sim 1.3 \times 10^4$ cells/cm²)^{4, 24}.

Figure R7. Density-dependent ferroptosis waves in HT-1080 cells. Representative time-lapse images at the indicated time points after erastin treatment (10 μM) are shown. HT-1080 cells were seeded at two different densities: 2×10^4 cells/cm² (medium density, upper panel), and 4×10^3 cells/cm² (low density, lower panel) two days before erastin treatment. Cell death is indicated by increased nuclear dye fluorescence signal. The white outlines represent the boundaries of cell death areas.

In summary, cell density, ferroptosis sensitivity, and other technical factors (e.g. time resolution and field of view for imaging) are critical to observe wave-like ferroptosis propagation. Our findings indicate that ferroptosis propagation in erastin-treated cells is a general phenomenon in various types of human cells.

P3. In establishing the main experimental model system, the concentration of ferrostatin-1 used in Extended data figure 3 (5 μ M) is about 100x higher than the reported EC50 for this molecule. There is some concern this may be having unanticipated effects. It would be good to repeat this experiment and see whether lower doses of Fer-1 are protective or not, and also include additional canonical inhibitors at reasonable concentrations (liproxstatin-1, iron chelators) so as to ensure that the process being monitored conforms to the ferroptosis mechanism as classically understood and not some related but distinct mechanism of cell death induced by the blue light exposure.

We thank reviewer 1 for these suggestions. To further strengthen the evidence supporting that the blue light-induced cell death is ferroptosis, we have now repeated this experiment with a lower dose of ferrostatin-1 (60 nM). We have also included additional ferroptosis inhibitors in the experiment, i.e., liproxstatin-1 (30 nM) and DFO (200 μ M). All three inhibitors potently inhibited blue light-induced cell death, further confirming it to be ferroptosis. These results have been incorporated in the revised manuscript in new Extended Data Figure 3d.

P4. The authors invoke a ROS diffusion model to account for wave propagation. The only suitable small molecule “ROS” species with the chemical stability to diffuse longer distances might be hydrogen peroxide (H₂O₂). Does the addition of catalase to cell cultures prevent wave propagation across distance? If not, this suggests another species whose identity requires isolation and definition. In general, the nature of the diffusible species that mediates ROS wave propagation appears to be very ill-defined. How have the authors ruled out the possibility of a reactive lipid fragment diffusing between cells, for example? Or release of iron from dying cells that then can induce peroxidation in neighboring cells? These and many other mechanisms apart from the one proposed by the authors could be envisioned. All of these models need to be rigorously tested using chemical but also genetic manipulations.

We thank reviewer 1 for these insightful comments and suggestions. Identifying diffusive cytotoxic molecules in ferroptotic cells has already been a subject of considerable interest in the ferroptosis field. Nishizawa et al. (2021) showed that cells undergoing ferroptosis release cytotoxic molecules that can transmit cell death, proposing that lipid peroxides act as the secreted molecule ¹¹. Moreover, by-products of lipid peroxidation, such as lipid electrophiles containing a peroxide group, have also been proposed as other candidate diffusive molecules secreted from ferroptotic cells ¹². The reviewer also mentions H₂O₂ as potentially being the diffusive molecule due to its chemical stability. We have now further characterized the diffusive molecule for ferroptosis waves by carrying out the following new experiments:

1. Addition of antioxidants to scavenge lipid peroxides (Trolox, Fer-1), superoxide radicals, hydroxyl radicals and hydrogen peroxide (Tiron, TEMPO, NAC) effectively halted ferroptosis propagation (new Extended Fig. 6e), indicating that multiple ROS can collectively contribute to the signaling wave fronts that drive ferroptosis propagation.

2. Conditioned media containing the diffusive molecule from ferroptotic cells can induce naïve untreated RPE cells to die. Pre-treatment of the conditioned media with different ROS scavengers suppressed this cell death, supporting that the diffusive molecule is **a type of reactive oxygen species** (new Extended Data Fig. 8a), and unlikely to be iron released from dying cells as suggested by the reviewer.

3. **We have ruled out the small molecule ROS H_2O_2** as being the diffusive molecule based on the results of centrifugal filtration experiments (new Extended Data Fig. 8b). In contrast to eluates of media to which H_2O_2 had been added exogenously, eluates of the conditioned media did not induce cell death.

Collectively, these results show that the diffusive molecule is a type of ROS, the activity of which is greatly diminished by ROS scavengers. It is also sufficiently chemically stable to diffuse between cells. Having ruled out H_2O_2 as the diffusing molecule, we suspect that the actual diffusing ROS are peroxidized lipids and/or their by-products, such as lipid-derived electrophiles with a peroxide group, since they have been found to accumulate in cells undergoing ferroptosis^{12,25} and they are chemically stable allowing them to freely diffuse between cells (half-life ~20 mins)²⁶. Future investigations employing lipidomic, proteomic and dedicated biochemical analyses are critical to definitively determine the nature of the diffusive ROS molecule, which we feel are beyond the scope of the current work. We now discuss these new results in page 6-7, lines 144-161 of the revised manuscript.

P5. In figure 3a, it is hard to understand how the authors see the process of lipid peroxidation (LOOH) contributing to the formation of soluble hydrogen peroxide (blue box in 3a). Presumably, the authors envision that H_2O_2 diffusing between cells can initiate lipid peroxidation in a neighboring cell. How plausible is this? Under the conditions the authors employ, is direct application of H_2O_2 in a localized manner sufficient to trigger “ferroptotic waves”, as would be predicted from the model?

We thank reviewer 1 for this question. As depicted in Figure 3a, H_2O_2 can react with iron via the Fenton reaction to produce hydroxyl radicals that react with lipids to generate lipid peroxides (LOOH)^{27,28}. Therefore, as per the scenario suggested by reviewer 1, addition of exogenous H_2O_2 would be expected to elevate LOOH levels and activate the respective autocatalytic loop, eliciting ferroptotic cell death. We have now validated this model by adding exogenous H_2O_2 to erastin-treated cells. Consistent with our prediction, when we added H_2O_2 to a localized region of a cell population, we observed elevated levels of cellular ROS and initiation of ferroptosis at that site (new Extended Data Fig. 3c), followed by ferroptosis propagation across the entire cell population (Fig. R8). This result shows that H_2O_2 is capable of initiating ferroptosis trigger waves, likely by elevating lipid peroxidation, as illustrated in our model (Fig. 3a). We have now incorporated this result in the revised manuscript (page 4, line 100). Moreover, we wish to re-emphasize that the diffusive ROS is not H_2O_2 (as elaborated in our response to **P4**).

Figure R8. Localized H_2O_2 addition initiates ferroptosis trigger waves. Image derived from merging ROS (yellow) and nuclear dye fluorescence (cyan) signals 9 hours after wave initiation upon local addition of exogenous H_2O_2 (80 μ M, 2 μ L). Cellular ROS was monitored using a general ROS dye (CellROX). Each yellow contour represents the border of the ROS wave front at a specific time-point.

P6. [For clarity, we would like to address this point P6 before P7] What is the effect of manipulating the expression of well-known ferroptosis effectors, like GPX4 or FSP1, on wave propagation? The authors focus on NOX enzymes but appear to ignore the role of more well-known and potentially universal regulators of ferroptosis.

We thank the reviewer 1 for this suggestion. Indeed, our initial emphasis was on regulators involved in amplifying ROS through positive feedback loops. However, we have now investigated the impact of other well-known ferroptosis effectors, i.e., FSP1 and GPX4, on ferroptosis propagation:

1. Ferroptosis suppressor protein 1 (FSP1)

FSP1 is an oxidoreductase that regenerates the reduced form of coenzyme Q₁₀ (CoQ), which functions as a radical-trapping antioxidant to scavenge lipid peroxy radicals²². Treatment with a FSP1 inhibitor (FSEN1) alone did not effectively induce ferroptosis or its propagation, consistent with previous studies²⁹. We have further examined its effect on ferroptosis waves by combining erastin treatment with FSP1 inhibition. Our results show that FSP1 inhibition increases the speed of ferroptosis propagation (new Extended Data Fig. 14), indicating its regulatory role in modulating the progressivity of ferroptosis waves.

Similar to FSP1, other ferroptosis regulators, namely dihydroorotate dehydrogenase (DHODH)³⁰ and the biopterin-dihydrofolate reductase system³¹, also function to suppress autocatalytic lipid peroxidation. Using mathematical modeling, we examined the potential general effect of these lipid peroxidation suppressors on ferroptosis propagation. We considered the rate of ROS degradation (k_d) to be dependent on the activities of these ferroptosis suppressors in the presence of erastin. Our simulations show that reducing the activities of these ferroptosis suppressors can promote the speed of ferroptosis waves in a dose-dependent manner (Fig. R9), consistent with our experimental results. Together, our results show that, in addition to ROS amplifiers such as NOX enzymes, lipid peroxidation suppressors can also alter the progressivity of ferroptosis trigger waves. We now discuss these results in page 9, lines 216-221 of the revised manuscript.

Figure R9. The speed of ferroptosis trigger waves increases with lower ROS degradation rates. The simulated speed of ferroptosis trigger waves in erastin-treated cells is plotted with respect to varying ROS degradation rates (k_d).

2. Glutathione peroxidase 4 (GPX4)

In our original manuscript, we show that RSL3 induces wave-like propagation of ferroptosis in RPE cells, as indicated by the higher spatial and temporal orders (i.e., lower entropy) of cell death kinetics. We have now added time-lapse images of RPE cells treated with RSL3 to further demonstrate ferroptosis propagation upon GPX4 inhibition (new Extended Data Fig. 1e, f). Therefore, unlike ROS amplifiers and suppressors (e.g., FSP1) that only influence the speed of ferroptosis, **GPX4 directly impacts the occurrence of ferroptosis waves.**

P7. GPX4 inhibition are canonical inducers of ferroptosis. Does RSL3 or a similar covalent GPX4 inhibitor induce the same trigger wave phenomenon in RPE1 and other cells? If not, what might this imply?

[Text redacted]

[Figure redacted]

[Text redacted]

[Figure redacted]

[Text redacted]

[Figure redacted]

[Text redacted]

[Figure redacted]

[Text redacted]

P8. Relevant to Figure 4 and the modelled bistable switch, the concept is intriguing but additional controls are needed. Perhaps erastin, between two concentration ranges (e.g., 0.63 and 1.25 μM), simply has non-linear effects on target function, more effectively inhibiting cystine uptake and/or depleting intracellular thiol-containing metabolites. It would be important to know whether or not

We thank reviewer 1 for these questions that have provided us with the opportunity to further clarify our interpretation of the results. Erastin is a small molecule that inhibits uptake of cystine, a metabolic precursor for *de novo* GSH synthesis. When we compared GSH levels after treatments with different erastin concentrations, we detected an overall hyperbolic response, with GSH levels decreasing linearly with increasing erastin concentrations (from 0.31 μM to 1.25 μM) (new Extended Data Fig. 17a). Notably, there was no non-linear decrease in GSH levels when the erastin concentration was increased from 0.63 μM to 1.25 μM , indicating that erastin's immediate and secondary targets (cystine and GSH, respectively) change linearly with erastin concentrations.

The non-linear response of ROS with respect to erastin concentrations mentioned by the reviewer (Fig. 4c) can be attributed to the transition of the cellular redox system, i.e., from having a single low ROS steady state to having bistable states when two stable steady states co-exist (lower and upper ROS steady states) (Fig. 4a). This monostable-to-bistable transition confers the non-linear and discontinuous ROS steady state response with respect to erastin concentration. By introducing sufficient ROS-inducing stress (e.g., erastin for ferroptosis stress), ROS positive feedback loops are activated, prompting the emergence of ROS bistability³⁷. For instance, when the erastin concentration is less than 0.63 μM , the redox system is monostable, so photoinduction only causes a transient increase in ROS that immediately reverts back to its low steady state (Fig. 4c). However, the redox system is bistable when the erastin concentration is greater than 0.63 μM , so elevating ROS above the activation threshold (as in Fig. 4a, blue arrow; threshold: white dot) by means of photoinduction activates the ROS positive feedback loops that drive the ROS increase to the upper steady state. At an erastin concentration of 0.63 μM , cells display a bimodal distribution of ROS levels, indicative of its proximity to the monostable/bistable transition point. This transition from monostable to bistable, also known as a saddle-node bifurcation⁴⁰ in the language of dynamical systems theory, is the framework for many all-or-none irreversible cell fate decisions, including differentiation⁴⁰, cell division⁴¹, the epithelial-to-mesenchymal transition⁴² and, in our study, the cell death decision. Following the reviewer's suggestion, we now underscore the linear change in glutathione level in accordance with erastin concentration to strengthen our point in page 12, lines 270-275.

P9. I am unclear whether the data reported in Figure 5h are from 'normal' physiological remodeling/death events, or PALP/laser-induced death in the isolated limb model. If the latter, it is unclear what relevance this has. One would like to test whether a reasonable concentration of a ferroptosis inhibitor (or genetic overexpression of a ferroptosis inhibitor like GPX4) is sufficient to prevent cell death in this model undergoing cell death without any external and artificial induction.

We thank reviewer 1 for this question that allows us to further clarify our results. The data we report in original Figure 5h (new Fig. 5i in the revised manuscript) are death events during normal physiological remodeling, and no laser or any artificial death inducer was introduced. The large-scale cell death in the central ectodermal layer of the limb zeugopod was observed to occur *in vivo* (new Fig. 5d), and co-occurred with lipid peroxidation (Fig. 5c). We assessed if this large-scale cell death could be caused by ferroptosis waves by monitoring cell death occurrence in real-time. As shown in Fig. 5e, cell death propagates as a wave along the central region of the limb zeugopod.

According to the reviewer's suggestion, we have now tested if a lower concentration of Fer-1 (10 μM) could prevent this cell death event. As shown in new Figure 5i, 10 μM of Fer-1 significantly reduced the extent of cell death in the embryonic limb. Thus, compared to the effect of the higher dose of Fer-1 (50 μM) we presented in the original manuscript, 10 μM of Fer-1 decreased the cell death area to a lesser extent. We have strengthened the evidence supporting the involvement of ferroptosis in this cell death event by testing the impact of two additional ferroptosis inhibitors (UAMC-3203, 1 μM ; and DFO, 10 mM), as well as an apoptosis inhibitor (Z-VAD-FMK, 200 μM). These ferroptosis inhibitors, but not the apoptosis inhibitor, significantly suppressed cell death in the embryonic limb (new Fig. 5i). We anticipated that UAMC-3203 would display greater potency in terms of cell death inhibition than Fer-1, since previous study demonstrated its superior efficacy in inhibiting ferroptosis³. Additionally, we also performed *in ovo* injection of UAMC-3203 (5 mM), which resulted in profound muscle remodeling defects in the developing limb (as elaborated in "Overall responses to all reviewers", see point A.5 above). Together, these results substantiate the natural occurrence of ferroptosis in inducing large-scale cell death in the embryonic limb during the process of muscle remodeling.

We wish to emphasize that the drug doses we used are comparable to the doses used on *ex vivo* tissues in the published literature (Fer-1, 10 μM ⁴³; UAMC-3203, 10 μM ⁴⁴; DFO, 10 mM⁴⁵; Z-VAD-FMK, 100 μM ⁴⁶). The dose of UAMC-3203 used for *in ovo* injection was estimated based on the dosage difference for compounds used *ex vivo* and *in ovo*. For example, there is a 17000-fold difference in the

concentration of retinoic acid used in *ex vivo* culture (1 μM ⁴⁷) and *in ovo* (17 mM⁴⁸) to study the same function. Thus, the ~2500-fold difference in the UAMC-3203 dose for *ex vivo* (10 μM ⁴) and our *in ovo* (5 mM) experiments is reasonable.

P10a. [For clarity, we have broken this point down into two parts and answer them separately]. Relevant to the wave-like cell death in developing avian limbs, what is the natural trigger for this process? What is known about this developmental event and how could it be related to the induction of ferroptosis? It would also be good to experimentally test whether inhibition of apoptosis or other forms of cell death in this context do or do not prevent cell death.

We thank reviewer 1 for raising these questions and apologize for not giving sufficient background relevant to this developmental event in our original submission. We have now added substantial amounts of **new** data that place our original findings in an important context, i.e., ferroptosis and its propagation in the ectodermal layer of the limb zeugopod plays a functional role in muscle remodeling during limb development (see our detailed response in “Overall responses to all reviewers”, point A above). As per the suggestion of all the reviewers, we now discuss these results in the context of relevant literature on muscle remodeling in page 13-14, lines 300-325 of our revised manuscript.

Regarding the natural trigger for muscle mass remodeling via cell death, this intriguing subject has not yet been explored extensively. Rodriguez-Guzman et al. (2007)¹ proposed the involvement of retinoic acid signaling in inducing muscle cell death. Previous studies have also proposed a role for BMP signaling in muscle remodeling, with high BMP levels having been found to cause muscle precursor death⁴⁹. Moreover, BMPs were found to be prominently expressed in the limb ectoderm, creating a gradient of BMP signal that determines the spatial positioning of muscle cells⁵. However, it remains unknown how BMP signals regulate cell death during muscle mass segregation.

How these developmental signals prime cells to undergo ferroptosis awaits further investigation. Based on our current understanding of how retinoic acid regulates ferroptosis, it is likely context-dependent whether retinoic acid signals promote or suppress ferroptosis. Although retinoic acid has been shown to inhibit ferroptosis through its antioxidant activity⁵⁰, it has also been found to promote ferroptosis by inhibiting glutathione synthesis^{51,52}. The ability of retinoic acid to enhance free radical generation⁵³ may also impact on ferroptosis susceptibility. Moreover, since the interplay between BMP and ferroptosis has to our knowledge not been studied previously, we can only speculate that BMP might promote ferroptosis by stimulating ROS production by upregulating NADPH oxidase⁵⁴. These morphogens can create concentration gradients in space that may encode region-specific expression patterns of ferroptosis regulators, thereby priming the cell population to undergo ferroptosis during development, representing an intriguing hypothesis for future investigation. To do so, detailed mechanistic studies examining the potential involvement of retinoic acid and BMP in the ferroptosis pathway will be required. Though such a mechanistic investigation is beyond the scope of the current work, our findings illuminate the path for investigating how developmental signals regulate ferroptosis during tissue- and organ-level sculpting. We now raise this intriguing question in the revised manuscript (page 12-13, lines 293-298; page 16-17, lines 383-392).

P10b. The working assumption for most people would be that apoptosis accounts for most developmental cell death, so the concept that ferroptosis may underlie this massive developmentally regulated event may require some stronger confirmation and additional controls to rule out other possible mechanisms.

We appreciate reviewer 1 for raising this important aspect of our study. We agree with the reviewer's point of view that apoptosis has been well-recognized as the most prominent form of cell death during embryonic development. We have already provided our perspective on this sentiment shared by all reviewers (see our “Overall response to all reviewers” point B above). Contrary to the conventional view

of apoptosis being the predominant form of cell death, several studies have also indicated that an as yet unknown mechanism could act collaboratively and/or synergistically with apoptosis to coordinate developmental cell death events. In fact, our new data (outlined in “Overall responses to all reviewers” point A above) directly challenges this conventional view by demonstrating a functional role for ferroptosis in muscle remodeling during limb development.

P11. Some very high-profile research suggests that ferroptosis is inhibited at high cell density (Wu et al, 2019, Nature). The authors should perhaps comments on how this may be related to their model and to the concept of ferroptosis wave-like propagation if at high cell density the process is inhibited. Have the authors examined wave propagation at various cell densities including especially high cell densities? In vivo, how can the authors account for wave propagation in (presumably) tightly packed cells?

We thank reviewer 1 for raising this important point. As mentioned by the reviewer, a high cell density has been shown to suppress ferroptosis by inhibiting the Hippo-YAP pathway, resulting in downregulation of pro-ferroptosis genes, such as ACSL4, TFRC (Wu, et al., 2019, *Nature*)²¹ and NOX4⁵⁵. Downregulating these pro-ferroptosis genes involved in iron and NOX signaling can weaken or inactivate the Fenton- and NOX-mediated ROS positive feedback loops, thereby suppressing ferroptosis trigger waves. Consistent with previous findings, we observed that a high cell density inhibits ferroptosis and its propagation in A549 and U-2 OS cells (see our response to **P2**). As also mentioned in our response to **P2**, we now show that enhancing the Fenton-mediated ROS feedback loop can induce ferroptosis waves in populations of both of these cell lines (new Extended Data Fig. 13). This result suggests that cells gaining resistance to ferroptosis at high densities can be rendered ferroptosis-sensitive by enhancing their ROS positive feedback loops, thereby enabling ferroptosis propagation across the cell population. We now discuss these results in the context of our model (Fig. 3a) in page 9, lines 208-216 of our revised manuscript.

Based on these findings, we suspect that extracellular signaling molecules can prime tightly packed cells *in vivo* to exhibit ferroptosis waves by modulating their ROS feedback loops. In particular, signaling molecules (e.g., morphogens) can form concentration gradients in space that lead to region-specific expression patterns of ferroptosis regulators, thereby priming distinct cell populations to exhibit ferroptosis waves (as discussed in our response to **P10a**). Indeed, our study demonstrates that this is a feasible scenario, with large-scale cell death being spatially-restricted in the central zeugopodial region of the limb. We show that PUFA levels are higher in the central region of the limb zeugopod (Fig. 5h), which can facilitate ferroptosis propagation along this specific area (new Fig. 5e). Moreover, in a previous study, a ferroptosis regulator like GPX4⁵⁶ was shown to display lower expression in the interdigital tissue of the developing limb, where ROS levels were also found to be elevated. Apart from during development, pathological stimuli (e.g., ischemia) can also prime cells to undergo large-scale ferroptotic cell death¹⁴. Therefore, we suspect that developmental signals or pathological stresses can prime specific tissue regions to undergo ferroptosis during development and disease pathologies.

Minor:

1. It is unclear what cells are being examined in extended data figure 1a and b.

The cells being examined here are RPE-1 cells. We now state this in the figure legend of new Extended Data Figure 1a & 1b.

2. Extended data Figure 2. What does a “technical repeat” mean in this context? Measuring the same image multiple times?

Technical repeats are independent samples from one experiment. This has been stated in the Methods (Reproducibility and statistical analysis).

3. It would be important to demonstrate that all small molecule inhibitors used here are not acting as RTAs except when that is the known/desired effect. E.g., is the PI3K or TK inhibitor an RTA at the concentrations employed, and could this explain the observed effects?

We have now tested the potential radical trapping antioxidant activity of the small molecule inhibitors for PI3K (LY294002), TKs (dasatinib), and NOX1/4 (GKT137831) at the highest concentrations used in our experiments. Using the 2,2-diphenyl-1-picrylhydrazyl (DPPH) antioxidant assay (Dojindo), the antioxidant potential was measured as the decrease in absorbance at 517 nm of the stable free radical DPPH. As shown in Extended Data Figure 11, in contrast to the antioxidant Trolox that reduced the absorbance at 517 nm, all three inhibitors exhibit similar absorbance with the solvent control (DMSO), indicating their lack of antioxidant activity. We now mention this result in page 8, lines 193-194 of the revised manuscript.

4. Figure 5f the correct formula to use is oxidized / (reduced + oxidized) to normalize for probe uptake. These data need to be re-analyzed and, possibly, this will alter the interpretation.

We have now re-analyzed the result using the formula

$$\frac{\text{Oxidized C11-B}}{\text{Oxidized C11-B + Reduced C11-B}}$$

The new quantification yields qualitatively identical results, confirming the validity of our previous interpretation.

The results have been updated accordingly in new Figure 5h & 5i.

5. Quantification of data from multiple limbs/animals in Fig 5a,b seems warranted.

In our original manuscript, we used Nile Blue staining to detect cell death in the embryonic limb (original Fig. 5a). To use a better-defined molecular assay for cell death detection, we have now performed TUNEL staining in the embryonic limb. TUNEL staining revealed abundant cell death in the central region of the limb zeugopod along the proximo-distal axis (Fig. 5d), consistent with our findings from Nile Blue staining. We have now replaced the Nile Blue staining data with the new TUNEL staining data. This new data and the data in original Figure 5b (now Fig. 5c) are consistent across multiple biological repeats. We have now quantified cell death and 4-HNE signals in multiple limbs (n = 3) and present them in new Extended Data Fig. 19.

6. The avian limb experiments are intriguing but preliminary. 50 μM Fer-1 is a very high concentration of inhibitor to use.

We have now used a lower concentration of Fer-1 (10 μM), which also elicited a significant decrease in the cell death area (new Fig. 5f). In addition, we also found that other ferroptosis inhibitors also suppressed cell death in the embryonic limb (see our response to **P9** for details).

Referee #2 (Remarks to the Author):

In this interesting paper, the authors have discovered that ferroptosis spreads from cell to cell in culture via trigger waves, which spread over substantial distances (millimeters) without slowing or diminishing in amplitude. This phenomenon can be seen in several different cell types and in response to a variety of ferroptotic induction strategies. The waves can propagate across a physical gap of up to about 150-200 μm , showing that some diffusible species is responsible for the local coupling of the cells, and inhibitor studies show the species is probably some ROS. The authors then characterize various perturbations that can make the wave speed faster (e.g. Fe³⁺-citrate, ERK2 overexpression) or slower (NOX inhibitor, PI3K inhibitor, Src/Abl inhibitor), which implicates various known mediators of ferroptosis in the positive feedback that underpins the trigger waves. Finally, they show that ferroptotic cell death occurs in the developing avian limb bud, and that it appears to spread at speeds comparable to those seen in cell culture.

Figures 1-3 and the related videos and supplementary figures are terrific. They convincingly establish that ferroptosis spreads via trigger waves, show that a diffusible species is responsible for the local coupling, and implicate various regulators in wave generation. The work is original, convincing, thorough, and important. Figure 4 is a bit more problematic. I think there are other equally plausible explanations for how photoinduction changes the system to make it better able to generate/propagate ferroptotic waves. And Figure 5, which argues that the cell culture work is relevant to limb remodeling in chick development, may be important, but I do not know the field well enough to assess it adequately. I would gladly defer to a developmental biologist on this point.

We thank reviewer 2 for these encouraging remarks. Below are our point-by-point responses.

Major criticisms:

P1. Fig 4. I am not completely convinced of the authors' interpretation of this experiment. Isn't it possible that the system is bistable even prior to photoinduction, but the cells are just more stuck on the lower stable branch of the S-shaped response curve? Perhaps the photoinduction pulls the unstable branch down toward the stable lower branch rather than changing a single-valued response function to a multi-valued response function. This would be another way that ferroptotic stress could, in principle, prime the cells to propagate ferroptotic trigger waves.

We thank reviewer 2 for these insightful comments that have allowed us to further investigate the redox system during ferroptosis stress. In the statement "...system is bistable even prior to photoinduction, but the cells are just more stuck on the lower stable branch of the S-shaped response curve", the reviewer suggests the possibility that redox bistability could be an inherent characteristic of human cells, i.e., prior to photoinduction and in the absence of erastin treatment. In the language of dynamical systems theory, if we consider the potential landscape of the system (Fig. R14), the potential barrier required to transition to the upper steady state is much greater in the absence of ferroptosis stress compared to when ferroptosis stress is present (e.g., under the erastin treatment condition). Thus, the cells appear to be "more stuck".

Figure R14. Potential curves for ROS in erastin-treated and untreated (control) cells. The stable steady states (SSS) and unstable steady states (USS) are indicated as solid and open circles, respectively.

We explored this scenario using our mathematical model by modulating the ROS positive feedback loop strength parameter ($k_{positive_fb}$). By increasing $k_{positive_fb}$ (36%), we simulated the alternative model proposed by reviewer. Compared to the current model we proposed, whereby increasing the erastin concentration lead to a bifurcation of ROS from the monostable to bistable states (left panel, Fig. R15), this alternative model exhibits ROS bistability in the absence of erastin treatment (right panel, Fig. R15). Given that cells can exhibit wide variations in cellular redox states, it is possible that the strength of the ROS positive feedback loop is broadly tunable, so both types of cellular redox systems (Fig. R15) are plausible.

Figure R15. Increasing the ROS positive feedback strength parameter ($k_{positive_{fb}}$) results in bistability in the absence of erastin. Simulated ROS steady state as a function of erastin concentration when $k_{positive_{fb}} = 1.63$ (right panel) and $k_{positive_{fb}} = 1.2$ (left panel).

A direct method to definitively test the presence of ROS bistability in the absence of erastin would be to probe for ROS irreversibility or hysteresis. However, since erastin is an irreversible inhibitor, it is currently not possible to reduce the erastin dose after applying a high erastin dose (e.g., by washing out erastin). Therefore, to probe for ROS bistability in the absence of erastin (right panel, Fig. R15), we used an alternative approach by exogenously introducing ROS into cells. We anticipated that if redox bistability exists in the absence of erastin, introducing exogenous ROS above the threshold would trigger the ROS bistable switch from low to high steady states. We introduced ROS exogenously by: 1) applying varying intensities of photoinduction (greater than that used in Fig. 4); and 2) adding different doses of the ROS, hydrogen peroxide (H_2O_2).

As shown in Extended Data Fig. 16, untreated cells maintained their ROS levels at low steady states regardless of the photoinduction intensity. In stark contrast, the erastin-treated cells exhibited a switch-like elevation of cellular ROS to high steady states after photoinduction. Similarly, exogenous addition of increasing doses of H_2O_2 to erastin-treated cells elevated cellular ROS from low to high steady states, but that was not the case for untreated cells (Fig. R16), indicating that cellular ROS is monostable in the absence of ferroptosis stress. Note that addition of $>100 \mu M H_2O_2$ and applying even stronger light intensities caused cell death that is not mediated by ROS. That is, cellular ROS levels were not elevated and ROS scavengers could not inhibit the resulting cell death, **implying that we had already introduced the highest possible physiological levels of ROS**. These results provide supporting evidence for our model, whereby ferroptosis stress leads to a monostable to bistable bifurcation in ROS steady states in RPE cells (Fig. 4). We have now incorporated these results (i.e., from applying higher photoinduction intensities) in Extended Data Fig. 16, and discuss them in page 11, lines 257-258 of the revised manuscript.

Figure R16. Exogenous H_2O_2 addition does not elevate cellular ROS in the absence of erastin treatment.

a, Representative images of ROS fluorescence signal, as indicated by CellROX staining, in erastin-treated (10 μM) and untreated cells 1 hour after addition of different concentrations of H_2O_2 . **b**, Single-cell ROS steady states were measured in erastin-treated and untreated cells ($n = 40$) after addition of different concentrations of H_2O_2 .

P2a. [For clarity, we have broken this point down into two parts and answered them separately.] Fig. 5 is beyond my textbook-level understanding of the roles of cell death in development. My impression had been that phenomena like limb remodeling are due to coordinated apoptosis, but perhaps, as the authors imply, in at least some instances it is due to ferroptosis instead. Certainly there is massive cell death in the limb, and it is accompanied by lipid peroxidation, as the authors have convincingly demonstrated.

We appreciate reviewer 2 for raising this important aspect of our study. We agree with the reviewer's point of view that apoptosis has been well-recognized as the most prominent form of cell death during embryonic development. We have already provided our perspective on this sentiment shared by all reviewers (see our "Overall response to all reviewers" point B above). Contrary to the conventional view of apoptosis being the predominant form of cell death, several studies have also indicated that an as-yet-unknown mechanism could act collaboratively and/or synergistically with apoptosis to coordinate developmental cell death events. In fact, our new data (outlined in "Overall responses to all reviewers" point A above) directly challenges this conventional view by demonstrating a functional role for ferroptosis in muscle remodeling during limb development.

P2b. But as to the issue of whether the limb remodeling presented here is spreading from cell to cell via trigger waves, I'm not certain. The authors make some estimates of speeds, and the speeds do fit reasonably well with what is seen in culture, but overall this does not look like such a clear-cut example of a constant-speed front of cell death. Maybe it is starting from multiple foci, making it hard to define wave fronts, or maybe flows as well as diffusion are involved in the local coupling. Perhaps the authors could sum up in what ways the process in the limb resembles the ferroptotic spread seen in culture, and in what ways it differs. I do think it is admirable and useful that the authors have gone the "next step" from cell culture to in vivo development, but I am not sure how similar the in vivo and cell culture phenomena are to each other.

We thank reviewer 2 for this question and suggestion. We agree with the reviewer on the importance of pointing out the similarities and differences in cell death propagation observed in cell culture and in the embryonic limb, which have now been incorporated into our revised manuscript (page 16, lines 376-392).

Similar to the multiple spontaneous death initiation events detected in cell culture, we also observed variable numbers of death initiation events in embryonic limbs, which could be attributable to the stochastic nature of the cellular redox system (Extended Data Fig. 2). Multiple death initiation sites were present in the specific limb we showed in our original manuscript (original Fig. 5c, now Supplementary Video 11) and, as astutely pointed out by the reviewer, this poses a challenge to clearly define wave fronts in this specific limb sample. Accordingly, we sought a different limb that exhibited a single death initiation site (new Fig. 5e, also new Supplementary Video 10). Using this limb, we could better visualize the progression of cell death. The speed of propagation was measured as 1.15 $\mu\text{m}/\text{min}$ across a 1 mm distance, consistent with our previous measurement. For three embryonic limbs, cell death propagated with a speed of $1.30 \pm 0.13 \mu\text{m}/\text{min}$ (mean \pm SD), i.e., the same order of magnitude to the wave speeds measured in our cell-based assay. These measurements indicate a common biochemical system underlying the bistable media in different embryonic limbs and in cell culture.

Nevertheless, the ferroptosis priming signal is distinct in cell culture and in the embryonic limbs. Whereas ferroptosis propagation in our cell culture results from erastin-mediated cellular priming, ferroptosis waves in the limb arises from as-yet-uncharacterized developmental signals. These distinct priming signals may account for the different spatial areas encompassed by cell death waves. In contrast to eliminating the entire cell population in culture, ferroptosis waves in the embryonic limb are confined to a specific region. One type of spatially-encoded information derived from the priming signal

is the level of PUFAs (Fig. 5g-i), the distribution of which constrains ferroptosis waves in the embryonic limb.

Therefore, given the limitations of the embryonic limb for long-distance measurements of cell death propagation, we share the reviewer's concern with concluding that ferroptosis propagates as trigger waves in the embryonic limb. Instead, we deem it more appropriate to describe cell death propagation in the limb as wave-like spread of ferroptosis.

Whether "flow" is involved in mediating the large-scale cell death in the embryonic limb remains to be investigated. Since a circulatory system is present in the avian limb, it is a valid hypothesis that the cell death propagation can be regulated by blood flow *in vivo*, where diffusive ROS can be advected by blood flow. Given that there is negligible blood flow in the *ex vivo* dissected limb culture, flow likely plays only a minor role in ferroptosis spread *ex vivo*. However, we acknowledge that the exact role of flow *in vivo* warrants further investigation by detailed comparison of cell death kinetics *in vivo* and *ex vivo* in the future.

Minor criticisms:

1. Line 27: Please spell out what xCT stands for (cystine-glutamate transporter).

Apologies. We have now defined xCT in line 27.

2. Line 49: I suggest replacing "had remained unknown" with "is unknown".

Now changed as recommended.

3. Line 67: Please mention that you are looking at several different cell types in Supplementary Video 1—I think that is an important point to convey in the text.

Agreed. We now highlight the general occurrence of ferroptosis waves in different cell types in page 3, lines 71-74.

4. Fig 2: Please mention in the text that the gaps in the sheets of cells were produced by scratching with glass needles. I know it's in the methods section, but I think it's helpful for the reader to know without having to search for the info.

We now mention that the gaps were created by scratching the bottom of the plate with needles of different tip sizes in the legend of Figure 2 (lines 516-517).

5. Line 131: The estimate for the diffusion coefficient of the ferroptotic ROS signal across the cell gap assumes there is no degradation of the ROS species. Is this a reasonable assumption? If not, perhaps the authors should just skip this analysis, or repeat the analysis with a range of plausible degradation rates in a reaction/diffusion model.

We apologize for the confusion. In our original manuscript, the diffusion equation we used was $\frac{\partial c}{\partial t} = D \frac{\partial^2 c}{\partial x^2}$, which did not consider the degradation rate of the diffusive molecule. By incorporating a constant rate of degradation (K_{deg}), the diffusion equation is $\frac{\partial c}{\partial t} = D \frac{\partial^2 c}{\partial x^2} - K_{deg}c$, and the solution of this diffusion equation is $c(x, t) = \frac{M}{\sqrt{4\pi Dt}} \exp(-\frac{x^2}{4Dt} - K_{deg}t)$.

The ratio (r) of the diffusive molecule concentrations at positions x [$c(x,t)$] and x_0 [$c(x_0,t)$] (assuming $x_0 = 0$) at time t , can be expressed as:

$$r = \exp\left(\frac{x_0^2 - x^2}{4Dt}\right)$$

and the diffusion coefficient (D) of this molecule is:

$$D = \frac{x_0^2 - x^2}{4t \times \ln r}$$

Therefore, incorporating a constant rate of degradation does not change our previous computation for the diffusion coefficient. However, we now think that our estimate can be oversimplified due to the potential limited detection range of the cellular ROS indicator. We have now removed our previous statement on the estimate of the diffusion coefficient from our manuscript.

6. Extended Data Figure 10. Please indicate what-is-what better in panel A. E.g., is the top blot probed with ERK1/2 antibodies? Is the second blot probed with pERK1/2 antibodies? Does ERK2-PA co-migrate with ERK1?

Apologies for the confusion. We have now modified the labels of our Western blots to indicate the proteins we probed for: phospho-ERK1/2 and phospho-ERK2-P2A in the top blot, and ERK1/2 and ERK2-P2A in the second blot. ERK2-PA co-migrates with ERK1. Original Extended Data Figure 10 is now Extended Data Figure 12.

Referee #3 (Remarks to the Author):

In this work, the authors define mechanisms of large-scale, propagating waves of cell death triggered via ferroptosis. Several cell culture assays support the range of speed of these waves, their reliance on ROS for propagation, including across gaps in cell contact, and their experiments support a threshold at which ROS steady state levels bifurcate from monostable to bistable states.

This reviewer was specifically asked to comment on Figure 5 and the ferroptosis propagation in limb development. The data presented look convincing.

We thank the reviewer for his/her encouraging remarks on our findings. We are grateful for the reviewer's careful assessment of our work and constructive feedback. We present our point-by-point responses to particular comments below.

P1. A detail as a note to authors is that the Nile Blue pictures in Figure 5a appear to be viewed dorsally and the rest of the pictures show the opposite outline (ventral view?); it would be helpful to show the Nile Blue from the same aspect as the rest of the data as it is whole mount, and to define this.

We apologize for the confusion. In our original manuscript, we stated in the legend of Figure 5a that ventral views of the Nile Blue-stained limbs are shown. This is the same aspect as the rest of the data. In our revised manuscript, we have now employed TUNEL staining to the embryonic limbs, which is a better-defined molecular assay for detecting cell death. As shown in new Figure 5d, TUNEL staining revealed abundant cell death in the central region of the limb zeugopod along the proximo-distal axis, consistent with our original Nile Blue staining data. Therefore, we have replaced the previous Nile Blue staining data with this new TUNEL staining data (new Fig. 5d and Extended Data Fig. 18). Furthermore, to avoid confusion, we have now labeled the images as "ventral view".

P2. The most well known spatially resolved region of cell death in the limb occurs in the interdigital mesenchyme, which is not examined here. As it is such a well studied and clear phenomenon where the entirety of the tissue disappears, the reviewer can't help but wonder if this was examined. It may be appropriate to at least comment on this.

[Text redacted]

[Figure redacted]

[Text redacted]

[Figure redacted]

[Text redacted]

P3. The authors show distal to proximal cell death that appears to begin at or near the central carpal region and extend proximally at least through the zeugopod according to the authors' picture. John Saunders' 1966 publication was not accessible to this reviewer during the review period (requiring archival access, not online) but other publications note four areas of cell death in chick limbs: the anterior necrotic zone, posterior necrotic zone, interdigital necrotic zone and the opaque patch (Fernandez-Teran, et al, Dev Dyn, 2006; Montero, et al, Dev Dyn, 2021). One presumes this may be the opaque patch? It will be important that the authors put this cell death area in the context of accessible literature and established limb developmental biology and further define this. Importantly also, this reviewer is unaware of anything in the central chick limb that 'disappears' via cell death. The interdigital mesenchyme, of course, leads to the complete disappearance of the tissue between the digits, but the data this reviewer found (only Fernandez-Teran showed scant data at HH31-32) did not suggest cell death across a continuous swath of tissue in the central limb. The radius and ulna initiate as separate cartilage condensations and the tissue between the radius and ulna does not disappear, leaving this reviewer wondering how the chick limb accommodates such a large swath of continuous cell death in the center of the limb, leaving no empty space. It is encouraged that the authors contextualize this cell death further for the readers.

We are grateful to reviewer 3 for raising these questions and the very constructive feedback that has helped us to enhance the quality and impact of our work. We apologize for not giving sufficient background relevant to this developmental event in our original submission. We have now added substantial **new** data that place our original findings in an important context, i.e., that ferroptosis and its propagation in the ectodermal layer of the limb zeugopod plays a functional role in muscle remodeling during limb development (see our detailed response in "Overall responses to all reviewers", point A above). Based on all of the reviewers' suggestions, we now discuss these results in the context of relevant literature on muscle remodeling in page 13-14, lines 300-325 of our revised manuscript.

As mentioned by the reviewer, other publications note four areas of cell death—anterior necrotic zone, posterior necrotic zone, interdigital necrotic zone, and the opaque patch—all of which occur in the limb

mesodermal layer. We wish to clarify that the large-scale cell death we observed in the limb zeugopod does **not** correspond to the opaque patch. In contrast to the large-scale cell death we observed in the **ectodermal layer** of the embryonic limb at **embryonic stage HH31-32**, cell death in the opaque patch happens at the **inner mesodermal layer** and at an earlier **embryonic stage (HH23)**. The cell death event relevant to muscle remodeling that we examined here was previously noted by Rodriguez-Guzman et al. (2007)¹, which was also cited by the specific publication mentioned by the reviewer (Fig. 2 of Montero et al. 2021)⁶⁰. In the study by Rodriguez-Guzman et al. (2007), they described large-scale cell death occurring in the central limb zeugopod that separates the foot and shank muscles. Intriguingly, cell death was found not only to encompass the muscular layer but also the outer ectodermal layer (Fig. 2a of Rodriguez-Guzman et al., 2007). As elaborated on above in our “Overall responses to all reviewers” (point A), we found that these large-scale cell death events involve ferroptosis. Similar to the observation of Rodriguez-Guzman et al. (2007), we found that active caspase-3 co-localizes with 17% of degenerating muscles (Fig. R2). Moreover, our study reports that the ferroptosis indicator (4-HNE) co-localizes with ~80% of degenerating muscles (Fig. R1d, e, also new Extended Data Fig. 18d, e). It is possible that apoptosis and ferroptosis co-exist and can act collaboratively to mediate muscle cell death during development (as elaborated on above in our “Overall responses to all reviewers”, point B).

In another publication mentioned by the reviewer (Fernandez-Teran et al. 2006)⁶¹, the authors identified cell death events in the chick limb at different embryonic stages via TUNEL staining. That study employed TUNEL staining in longitudinal and transverse sections of the limb at the same embryonic stage we examined (i.e., HH31-32) (Fig. 2 of Fernandez-Teran et al., 2006). Whereas our results highlight extensive TUNEL staining signal in the **zeugopodial region** of the limb (Fig. 5d, Extended Data Fig. 18b), the limb tissue sections presented in the Fernandez-Teran et al. (2006) study does not show the zeugopodial region, but only features the **autopodial region** of the limb to emphasize interdigit tissue cell death. Since they focused on the autopodial region of the limb, it may explain why the cell death area we examined was not reported in their study.

References

- 1 Rodriguez-Guzman, M. *et al.* Tendon-muscle crosstalk controls muscle bellies morphogenesis, which is mediated by cell death and retinoic acid signaling. *Dev Biol* **302**, 267-280, doi:10.1016/j.ydbio.2006.09.034 (2007).
- 2 Kardon, G. Muscle and tendon morphogenesis in the avian hind limb. *Development* **125**, 4019-4032, doi:10.1242/dev.125.20.4019 (1998).
- 3 Van Coillie, S. *et al.* Targeting ferroptosis protects against experimental (multi)organ dysfunction and death. *Nat Commun* **13**, 1046, doi:10.1038/s41467-022-28718-6 (2022).
- 4 Mishima, E. *et al.* A non-canonical vitamin K cycle is a potent ferroptosis suppressor. *Nature* **608**, 778-783, doi:10.1038/s41586-022-05022-3 (2022).
- 5 Amthor, H., Christ, B., Weil, M. & Patel, K. The importance of timing differentiation during limb muscle development. *Curr Biol* **8**, 642-652, doi:10.1016/s0960-9822(98)70251-9 (1998).
- 6 Maroto, M. *et al.* Ectopic Pax-3 activates MyoD and Myf-5 expression in embryonic mesoderm and neural tissue. *Cell* **89**, 139-148, doi:10.1016/s0092-8674(00)80190-7 (1997).
- 7 Garcia-Martinez, V. *et al.* Internucleosomal DNA fragmentation and programmed cell death (apoptosis) in the interdigital tissue of the embryonic chick leg bud. *J Cell Sci* **106 (Pt 1)**, 201-208, doi:10.1242/jcs.106.1.201 (1993).
- 8 Lindsten, T. *et al.* The combined functions of proapoptotic Bcl-2 family members bak and bax are essential for normal development of multiple tissues. *Mol Cell* **6**, 1389-1399, doi:10.1016/s1097-2765(00)00136-2 (2000).
- 9 Hurlle, J. M., Ros, M. A., Climent, V. & Garcia-Martinez, V. Morphology and significance of programmed cell death in the developing limb bud of the vertebrate embryo. *Microsc Res Tech* **34**, 236-246, doi:10.1002/(SICI)1097-0029(19960615)34:3<236::AID-JEMT6>3.0.CO;2-N (1996).

- 10 Ke, F. F. S. *et al.* Embryogenesis and Adult Life in the Absence of Intrinsic Apoptosis Effectors BAX, BAK, and BOK. *Cell* **173**, 1217-1230 e1217, doi:10.1016/j.cell.2018.04.036 (2018).
- 11 Nishizawa, H. *et al.* Lipid peroxidation and the subsequent cell death transmitting from ferroptotic cells to neighboring cells. *Cell Death Dis* **12**, 332, doi:10.1038/s41419-021-03613-y (2021).
- 12 Do, Q. & Xu, L. How do different lipid peroxidation mechanisms contribute to ferroptosis? *Cell Reports Physical Science* **4**, doi:10.1016/j.xcrp.2023.101683 (2023).
- 13 Cheng, X. & Ferrell, J. E., Jr. Apoptosis propagates through the cytoplasm as trigger waves. *Science* **361**, 607-612, doi:10.1126/science.aah4065 (2018).
- 14 Linkermann, A. *et al.* Synchronized renal tubular cell death involves ferroptosis. *Proc Natl Acad Sci U S A* **111**, 16836-16841, doi:10.1073/pnas.1415518111 (2014).
- 15 Riegman, M. *et al.* Ferroptosis occurs through an osmotic mechanism and propagates independently of cell rupture. *Nat Cell Biol* **22**, 1042-1048, doi:10.1038/s41556-020-0565-1 (2020).
- 16 Wang, Y. *et al.* Neuronal gap junctions are required for NMDA receptor-mediated excitotoxicity: implications in ischemic stroke. *J Neurophysiol* **104**, 3551-3556, doi:10.1152/jn.00656.2010 (2010).
- 17 Perez-Garijo, A., Fuchs, Y. & Steller, H. Apoptotic cells can induce non-autonomous apoptosis through the TNF pathway. *elife* **2**, e01004, doi:10.7554/eLife.01004 (2013).
- 18 Potts, M. B. & Cameron, S. Cell lineage and cell death: *Caenorhabditis elegans* and cancer research. *Nat Rev Cancer* **11**, 50-58, doi:10.1038/nrc2984 (2011).
- 19 Raj, A. & van Oudenaarden, A. Nature, nurture, or chance: stochastic gene expression and its consequences. *Cell* **135**, 216-226, doi:10.1016/j.cell.2008.09.050 (2008).
- 20 Ozbudak, E. M., Thattai, M., Kurtser, I., Grossman, A. D. & van Oudenaarden, A. Regulation of noise in the expression of a single gene. *Nat Genet* **31**, 69-73, doi:10.1038/ng869 (2002).
- 21 Wu, J. *et al.* Intercellular interaction dictates cancer cell ferroptosis via NF2-YAP signalling. *Nature* **572**, 402-406, doi:10.1038/s41586-019-1426-6 (2019).
- 22 Doll, S. *et al.* FSP1 is a glutathione-independent ferroptosis suppressor. *Nature* **575**, 693-698, doi:10.1038/s41586-019-1707-0 (2019).
- 23 Yang, W. S. *et al.* Regulation of ferroptotic cancer cell death by GPX4. *Cell* **156**, 317-331, doi:10.1016/j.cell.2013.12.010 (2014).
- 24 Magtanong, L. *et al.* Context-dependent regulation of ferroptosis sensitivity. *Cell Chem Biol* **29**, 1409-1418 e1406, doi:10.1016/j.chembiol.2022.06.004 (2022).
- 25 Van Kessel, A. T. M., Karimi, R. & Cosa, G. Live-cell imaging reveals impaired detoxification of lipid-derived electrophiles is a hallmark of ferroptosis. *Chem Sci* **13**, 9727-9738, doi:10.1039/d2sc00525e (2022).
- 26 McGrath, C. E., Tallman, K. A., Porter, N. A. & Marnett, L. J. Structure-activity analysis of diffusible lipid electrophiles associated with phospholipid peroxidation: 4-hydroxynonenal and 4-oxononenal analogues. *Chem Res Toxicol* **24**, 357-370, doi:10.1021/tx100323m (2011).
- 27 Sies, H. & Jones, D. P. Reactive oxygen species (ROS) as pleiotropic physiological signalling agents. *Nat Rev Mol Cell Biol* **21**, 363-383, doi:10.1038/s41580-020-0230-3 (2020).
- 28 Yan, B. *et al.* Membrane Damage during Ferroptosis Is Caused by Oxidation of Phospholipids Catalyzed by the Oxidoreductases POR and CYB5R1. *Mol Cell* **81**, 355-369 e310, doi:10.1016/j.molcel.2020.11.024 (2021).
- 29 Hendricks, J. M. *et al.* Identification of structurally diverse FSP1 inhibitors that sensitize cancer cells to ferroptosis. *Cell Chem Biol* **30**, 1090-1103 e1097, doi:10.1016/j.chembiol.2023.04.007 (2023).
- 30 Mao, C. *et al.* DHODH-mediated ferroptosis defence is a targetable vulnerability in cancer. *Nature* **593**, 586-590, doi:10.1038/s41586-021-03539-7 (2021).
- 31 Soula, M. *et al.* Metabolic determinants of cancer cell sensitivity to canonical ferroptosis inducers. *Nat Chem Biol* **16**, 1351-1360, doi:10.1038/s41589-020-0613-y (2020).
- 32 Espinosa-Diez, C. *et al.* Antioxidant responses and cellular adjustments to oxidative stress. *Redox Biol* **6**, 183-197, doi:10.1016/j.redox.2015.07.008 (2015).

- 33 Novera, W. *et al.* Cysteine Deprivation Targets Ovarian Clear Cell Carcinoma Via Oxidative Stress and Iron-Sulfur Cluster Biogenesis Deficit. *Antioxid Redox Signal* **33**, 1191-1208, doi:10.1089/ars.2019.7850 (2020).
- 34 Alvarez, S. W. *et al.* NFS1 undergoes positive selection in lung tumours and protects cells from ferroptosis. *Nature* **551**, 639-643, doi:10.1038/nature24637 (2017).
- 35 Chang, D. E. *et al.* Building biological memory by linking positive feedback loops. *Proc Natl Acad Sci U S A* **107**, 175-180, doi:10.1073/pnas.0908314107 (2010).
- 36 Venturelli, O. S., El-Samad, H. & Murray, R. M. Synergistic dual positive feedback loops established by molecular sequestration generate robust bimodal response. *Proc Natl Acad Sci U S A* **109**, E3324-3333, doi:10.1073/pnas.1211902109 (2012).
- 37 Huang, J. H., Co, H. K., Lee, Y. C., Wu, C. C. & Chen, S. H. Multistability maintains redox homeostasis in human cells. *Mol Syst Biol* **17**, e10480, doi:10.15252/msb.202110480 (2021).
- 38 Zhang, H. L. *et al.* PKC β 1 phosphorylates ACSL4 to amplify lipid peroxidation to induce ferroptosis. *Nat Cell Biol* **24**, 88-98, doi:10.1038/s41556-021-00818-3 (2022).
- 39 Doll, S. *et al.* ACSL4 dictates ferroptosis sensitivity by shaping cellular lipid composition. *Nat Chem Biol* **13**, 91-98, doi:10.1038/nchembio.2239 (2017).
- 40 Ferrell, J. E., Jr. Bistability, bifurcations, and Waddington's epigenetic landscape. *Curr Biol* **22**, R458-466, doi:10.1016/j.cub.2012.03.045 (2012).
- 41 Pomerening, J. R., Sontag, E. D. & Ferrell, J. E., Jr. Building a cell cycle oscillator: hysteresis and bistability in the activation of Cdc2. *Nat Cell Biol* **5**, 346-351, doi:10.1038/ncb954 (2003).
- 42 Celia-Terrassa, T. *et al.* Hysteresis control of epithelial-mesenchymal transition dynamics conveys a distinct program with enhanced metastatic ability. *Nat Commun* **9**, 5005, doi:10.1038/s41467-018-07538-7 (2018).
- 43 Li, Q. *et al.* Inhibition of neuronal ferroptosis protects hemorrhagic brain. *JCI Insight* **2**, e90777, doi:10.1172/jci.insight.90777 (2017).
- 44 Balla, A. *et al.* A Novel Ferroptosis Inhibitor UAMC-3203, a Potential Treatment for Corneal Epithelial Wound. *Pharmaceutics* **15**, doi:10.3390/pharmaceutics15010118 (2022).
- 45 Hybertson, B. M., Connelly, K. G., Buser, R. T. & Repine, J. E. Ferritin and desferrioxamine attenuate xanthine oxidase-dependent leak in isolated perfused rat lungs. *Inflammation* **26**, 153-159, doi:10.1023/a:1016511611435 (2002).
- 46 Chautan, M., Chazal, G., Cecconi, F., Gruss, P. & Golstein, P. Interdigital cell death can occur through a necrotic and caspase-independent pathway. *Current Biology* **9**, 967-970, doi:10.1016/S0960-9822(99)80425-4 (1999).
- 47 Obinata, A. & Akimoto, Y. Effects of retinoic acid and Gbx1 on feather-bud formation and epidermal transdifferentiation in chick embryonic cultured dorsal skin. *Dev Dyn* **241**, 1405-1412, doi:10.1002/dvdy.23834 (2012).
- 48 Dhouailly, D., Hardy, M. H. & Sengel, P. Formation of feathers on chick foot scales: a stage-dependent morphogenetic response to retinoic acid. *J Embryol Exp Morphol* **58**, 63-78 (1980).
- 49 Duprez, D. M., Coltey, M., Amthor, H., Brickell, P. M. & Tickle, C. Bone morphogenetic protein-2 (BMP-2) inhibits muscle development and promotes cartilage formation in chick limb bud cultures. *Dev Biol* **174**, 448-452, doi:10.1006/dbio.1996.0087 (1996).
- 50 Jakaria, M., Belaidi, A. A., Bush, A. I. & Ayton, S. Vitamin A metabolites inhibit ferroptosis. *Biomed Pharmacother* **164**, 114930, doi:10.1016/j.biopha.2023.114930 (2023).
- 51 Sun, Y. *et al.* All-trans retinoic acid inhibits the malignant behaviors of hepatocarcinoma cells by regulating ferroptosis. *Genes Dis* **9**, 1742-1756, doi:10.1016/j.gendis.2022.04.011 (2022).
- 52 Wang, X. J., Hayes, J. D., Henderson, C. J. & Wolf, C. R. Identification of retinoic acid as an inhibitor of transcription factor Nrf2 through activation of retinoic acid receptor alpha. *Proc Natl Acad Sci USA* **104**, 19589-19594, doi:10.1073/pnas.0709483104 (2007).
- 53 Conte da Frota, M. L., Jr. *et al.* All-trans retinoic acid induces free radical generation and modulate antioxidant enzyme activities in rat sertoli cells. *Mol Cell Biochem* **285**, 173-179, doi:10.1007/s11010-005-9077-3 (2006).

- 54 Jiang, Q. *et al.* NOX4 mediates BMP4-induced upregulation of TRPC1 and 6 protein expressions in distal pulmonary arterial smooth muscle cells. *PLoS One* **9**, e107135, doi:10.1371/journal.pone.0107135 (2014).
- 55 Yang, W. H. *et al.* The Hippo Pathway Effector TAZ Regulates Ferroptosis in Renal Cell Carcinoma. *Cell Rep* **28**, 2501-2508 e2504, doi:10.1016/j.celrep.2019.07.107 (2019).
- 56 Schnabel, D. *et al.* Expression and regulation of antioxidant enzymes in the developing limb support a function of ROS in interdigital cell death. *Dev Biol* **291**, 291-299, doi:10.1016/j.ydbio.2005.12.023 (2006).
- 57 Cordeiro, I. R. *et al.* Environmental Oxygen Exposure Allows for the Evolution of Interdigital Cell Death in Limb Patterning. *Dev Cell* **50**, 155-166, doi:10.1016/j.devcel.2019.05.025 (2019).
- 58 Hernandez-Martinez, R., Castro-Obregon, S. & Covarrubias, L. Progressive interdigital cell death: regulation by the antagonistic interaction between fibroblast growth factor 8 and retinoic acid. *Development* **136**, 3669-3678, doi:10.1242/dev.041954 (2009).
- 59 Hernandez-Martinez, R. & Covarrubias, L. Interdigital cell death function and regulation: new insights on an old programmed cell death model. *Dev Growth Differ* **53**, 245-258, doi:10.1111/j.1440-169X.2010.01246.x (2011).
- 60 Montero, J. A., Lorda-Diez, C. I., Sanchez-Fernandez, C. & Hurle, J. M. Cell death in the developing vertebrate limb: A locally regulated mechanism contributing to musculoskeletal tissue morphogenesis and differentiation. *Dev Dyn* **250**, 1236-1247, doi:10.1002/dvdy.237 (2021).
- 61 Fernandez-Teran, M. A., Hinchliffe, J. R. & Ros, M. A. Birth and death of cells in limb development: a mapping study. *Dev Dyn* **235**, 2521-2537, doi:10.1002/dvdy.20916 (2006).

Reviewer Reports on the First Revision:

Referees' comments:

Referee #1 (Remarks to the Author):

The authors have been able to address my points in most cases. I congratulate them on an impressive amount of effort and for the clarity and depth of the responses. I do think that this work could now be suitable for publication in Nature. I would just like to clarify and offer opinions on a few points, below, that I feel are important to address in the writing of the manuscript:

1. From the way the RPE-1 experiments are initially described, it sounds like you add erastin first, then initiate ferroptosis with blue light (line 99). However, subsequent experiments do not always make it clear whether this erastin + blue light condition is used, or just blue light alone (e.g., in lines 105, and then for later experiments). This just needs to be clearer in all cases (e.g., maybe call it “erastin + blue light stimulation” clearly in every case). This may be especially relevant for the conditioned medium experiment described lines 144-46. If the collected “conditioned” medium contains the original small molecule erastin that was added to cells, then it is perhaps not surprising that it can induce ferroptosis when transferred to new cells, and the search for a diffusible species will perhaps be confounded, no?

1. If I am understanding the new results obtained by the authors, the covalent GPX4 inhibitor RSL3 does not induce trigger waves in 2/3 tested cancer cells lines and shows evidence for triggers waves in the one remaining cell line (RPE-1) at only a single dose, but not at higher or lower doses. These are interesting and important data. They suggest to me that the trigger wave phenomenon is not a universal feature of “ferroptosis”. Rather, this indicates that trigger waves are perhaps a feature of ferroptosis when this process is triggered by cystine deprivation (or blue light exposure). Indeed, this makes sense, as the models advanced in this paper about ROS bistable switches incorporate GSH, which is only affected by cystine deprivation and not GPX4 inhibition.

However, the authors argue against making this distinction clearer in the revised manuscript to avoid “render[ing] the study less accessible to a broad readership”. I disagree, strongly, for reasons I will elaborate on below.

First, GPX4 inhibition is perhaps the most common means of inducing ferroptosis in the literature. GPX4 is also a target for the development of new pro-ferroptosis therapies. Thus, people are very interested in the function of GPX4 and what happens when GPX4 is inhibited. It would not be helpful to confuse people that GPX4 inhibition also commonly can induce trigger waves.

Second, as cited by the authors, previous work has already suggested that GPX4 inhibition does not lead to waves of ferroptosis. So, it will not be surprising that, here, this is also observed in most cases, and this should be reinforced.

Third, there is much emerging evidence that cystine deprivation and GPX4 inhibition do not cause the same “form” or type of ferroptosis (e.g., see a summary by Dixon & Pratt, 2023 in Mol Cell). Indeed, ferroptosis induced by system xc- inhibition and ferroptosis induced by direct GPX4 inhibition appear quite distinct in many ways, including cell death kinetics, genetic regulation, and so on. So, I would

strongly suggest that the authors, in fact, “lean into” this distinction in their new work here. It is crucial to make this clear in this manuscript that trigger waves may be more specific to one “form” of ferroptosis, as opposed to a general phenomenon. This will help further differentiate these two forms of ferroptosis, stimulate new efforts to understand these differences, and also constrain the types of inputs that will be explored when searching for the (unknown) *in vivo* trigger.

Without this clear discussion, this manuscript will give the (unintentional) impression that trigger waves are a universal ferroptosis mechanism when, based on the models and evidence presented, they clearly are not. To not make this point clear in the manuscript will not help readers or the field, it will hurt (and confuse) readers and others in the field. It will also seem like the authors are over-stating their conclusions.

Should the authors still be on the fence on this point, and need additional motivation to re-consider, I would point to the distinction that is drawn in the apoptosis field between the “intrinsic” and “extrinsic” mechanisms of apoptosis. Delineating these distinct yet overlapping mechanisms has not diminished the apoptosis field. In fact, it has proven to be essential for our understanding. The same logic applies to ferroptosis. We need to distinguish between the different modes of ferroptosis that are observed, and trigger waves may provide a new and exciting way to do this.

2. In Supplemental Video 6 it is claimed that there is “ferroptosis deceleration” upon the addition of DFO. DFO may take time to enter the cell (via fluid phase endocytosis) so this might be consistent, but it is a rather inelegant experiment for that reason. This is just a note; there is no need to do anything on this point.

Minor points:

1. In the response to Reviewer #2, the authors state that “erastin is an irreversible inhibitor, it is currently not possible to reduce the erastin dose after applying a high erastin dose (e.g., by washing out erastin).” This is not true. Erastin is a reversible (non-covalent) inhibitor and can be washed out of cells. The problem is that cells exposed to erastin do not necessarily live very long and if one wants to perform washout experiments they have to be done carefully. But, technically, this is possible. I leave it to others to determine whether this is ultimately an important note or not.

2. Line 29 of the Abstract. Is “media” the correct word here? It does not seem like the right word. I see it used again in Line 57 and 59, which suggests perhaps that this is a term of art in the trigger wave field. Perhaps this is the only word to use. But, it is also possible that the manuscript would benefit from a careful definition of what this means since, to many cancer cell biologists, the “media” is simply the fluid that a cultured cell grows in, not something that is “excitable and bistable”. Indeed, the Author’s use media in this latter sense themselves in Line 96 (where grammatically this should perhaps be ‘medium’, the singular, not ‘media’, the plural).

Referee #1 (Remarks on code availability):

I am not a software person. I do not know how to evaluate code.

Referee #2 (Remarks to the Author):

The authors have strengthened this manuscript substantially. In particular I found the new data on muscle sculpting in Fig 5 to be stronger than what was in the original manuscript, although, admittedly, my knowledge of limb developmental biology is pretty minimal. In both the paper and the rebuttal, the authors more emphatically and convincingly make the case that something other than apoptosis contributes to programmed cell death during development, and that in limb development ferroptosis is important.

I have some minor criticisms of the early parts of the paper, which I delineate below, but overall I think this is important work. It provides convincing cell culture evidence that ferroptosis can spread from cell-to-cell via a diffusible factor that is released from a one ferroptotic cell and then gets amplified up by a not-yet ferroptotic target cell, leading to constant speed, constant amplitude trigger waves of ferroptosis propagating through the culture. And it puts this cell culture work into a physiological context via the evidence for ROS and ferroptotic waves in limb development.

Specific minor criticisms:

1. I do not think readers are going to be able to understand the data presented in Extended Data Figure 1. The authors need to explain what the vector fields, polar distribution plots, and entropies mean, and explain what differences in these measures would be expected if you have ferroptosis spreading from a discrete source versus ferroptosis occurring independently in each cell.
2. Lines 152-161. I agree that the diffusible mediator of ferroptosis is almost certainly a free radical other than hydrogen peroxide (Extended Figure 8), but I do not find the speculation that it/they is/are a lipid peroxide compelling. It could be a membrane-bound lipid peroxide, but it could also be a protein or anything in or on an organelle. Perhaps just lay out the various possibilities.
3. Figure 3: Do MEK or ERK inhibitors block the increase in wave speed (or perhaps block the production of ferroptotic waves altogether) seen after ERK overexpression? I don't insist on this experiment, but if you have the data it would be nice to mention it.
4. Line 244: Please clarify: do the authors mean that their model predicts that below the critical concentration of 0.63 μM , increasing the erastin concentration exerts a relatively mild effect on the ROS level of the low-ROS steady state? Or that above the critical concentration of 0.63 μM there is little effect of the erastin concentration on the high-ROS steady state? Either way, I would think it's not just a prediction (Fig 4a), but an experimental observation (Fig 4b).

Referee #3 (Remarks to the Author):

This revised manuscript, "Emergence of large-scale cell death via trigger waves of ferroptosis" has been substantially improved in revision. This reviewer was asked to comment on limb development portion, but unlike the previous version, I was able to follow the rationale, data, and conclusions through the first four figures as well. Will leave to other experts to make appropriate scientific comments, but readability of manuscript is excellent.

The additional experiments and data in Figure 5 (and associated Suppl Figs 18-20) provide excellent proof of principle in vivo in a way that will be appreciated by people in the field.

One remaining comment, which is meant only to reflect this reviewer's curiosity (and perhaps others from this field), if there is room in the discussion/end of manuscript, how do the authors think about both apoptosis and ferroptosis occurring in these biologically relevant areas? It is clear from their data this is occurring, but why both and what could distinguish cells that are presumably going through the same developmentally regulated process (cell death) via both of these distinct mechanisms?

Author Rebuttals to First Revision:

We appreciate all reviewers' positive remarks and careful assessments that have helped improve the quality of our work. We are also delighted that all reviewers found our study to be interesting and important. We have addressed each of their concerns and questions in the point-by-point responses below.

Detailed Responses to the Reviewers' Comments:

Referee #1 (Remarks to the Author):

The authors have been able to address my points in most cases. I congratulate them on an impressive amount of effort and for the clarity and depth of the responses. I do think that this work could now be suitable for publication in Nature. I would just like to clarify and offer opinions on a few points, below, that I feel are important to address in the writing of the manuscript:

We are grateful for the reviewer's positive comments and constructive remarks that have greatly helped enhance the impact and rigor of our work. To address the concerns raised by this reviewer, we have made textual clarifications, as elaborated below.

P1. From the way the RPE-1 experiments are initially described, it sounds like you add erastin first, then initiate ferroptosis with blue light (line 99). However, subsequent experiments do not always make it clear whether this erastin + blue light condition is used, or just blue light alone (e.g., in lines 105, and then for later experiments). This just needs to be clearer in all cases (e.g., maybe call it "erastin + blue light stimulation" clearly in every case). This may be especially relevant for the conditioned medium experiment described lines 144-46. If the collected "conditioned" medium contains the original small molecule erastin that was added to cells, then it is perhaps not surprising that it can induce ferroptosis when transferred to new cells, and the search for a diffusible species will perhaps be confounded, no?

We thank the reviewer for this suggestion. We now state that blue-light irradiation was performed on erastin-treated cells in line 103 of the manuscript (original line 105), and in the figure legends of the relevant experiments (Fig. 1-3 & Extended Data Fig. 3-6).

For the conditioned medium experiment, the conditioned medium does not contain erastin. As mentioned in the Methods section ("Characterization of conditioned media from ferroptotic cells"), following 4 hours of erastin treatment, erastin was washed out and replaced with erastin-free medium, i.e., conditioned medium. To avoid confusion, we now state that the conditioned medium is erastin-free in page 6, line 143, and in the figure legend (Extended Data Fig. 5b).

P2. If I am understanding the new results obtained by the authors, the covalent GPX4 inhibitor RSL3 does not induce trigger waves in 2/3 tested cancer cells lines and shows evidence for triggers waves in the one remaining cell line (RPE-1) at only a single dose, but not at higher or lower doses. These are interesting and important data. They suggest to me that the trigger wave phenomenon is not a universal feature of "ferroptosis". Rather, this indicates that trigger waves are perhaps a feature of ferroptosis when this process is triggered by cystine deprivation (or blue light exposure). Indeed, this makes sense, as the models advanced in this paper about ROS bistable switches incorporate GSH, which is only affected by cystine deprivation and not GPX4 inhibition.

However, the authors argue against making this distinction clearer in the revised manuscript to avoid "render[ing] the study less accessible to a broad readership". I disagree, strongly, for reasons I will elaborate on below.

First, GPX4 inhibition is perhaps the most common means of inducing ferroptosis in the literature. GPX4 is also a target for the development of new pro-ferroptosis therapies. Thus, people are very interested in the function of GPX4 and what happens when GPX4 is inhibited. It would not be helpful to confuse people that GPX4 inhibition also commonly can induce trigger waves. Second, as cited by the authors, previous work has already suggested that GPX4 inhibition does not lead to waves of ferroptosis. So, it will not be surprising that, here, this is also observed in most cases, and this should be reinforced. Third, there is much emerging evidence that cystine deprivation and GPX4 inhibition do not cause the same “form” or type of ferroptosis (e.g., see a summary by Dixon & Pratt, 2023 in Mol Cell). Indeed, ferroptosis induced by system xc- inhibition and ferroptosis induced by direct GPX4 inhibition appear quite distinct in many ways, including cell death kinetics, genetic regulation, and so on. So, I would strongly suggest that the authors, in fact, “lean into” this distinction in their new work here. It is crucial to make this clear in this manuscript that trigger waves may be more specific to one “form” of ferroptosis, as opposed to a general phenomenon. This will help further differentiate these two forms of ferroptosis, stimulate new efforts to understand these differences, and also constrain the types of inputs that will be explored when searching for the (unknown) *in vivo* trigger.

Without this clear discussion, this manuscript will give the (unintentional) impression that trigger waves are a universal ferroptosis mechanism when, based on the models and evidence presented, they clearly are not. To not make this point clear in the manuscript will not help readers or the field, it will hurt (and confuse) readers and others in the field. It will also seem like the authors are over-stating their conclusions.

Should the authors still be on the fence on this point, and need additional motivation to reconsider, I would point to the distinction that is drawn in the apoptosis field between the “intrinsic” and “extrinsic” mechanisms of apoptosis. Delineating these distinct yet overlapping mechanisms has not diminished the apoptosis field. In fact, it has proven to be essential for our understanding. The same logic applies to ferroptosis. We need to distinguish between the different modes of ferroptosis that are observed, and trigger waves may provide a new and exciting way to do this.

We thank the reviewer for emphasizing the importance of highlighting the difference between the ferroptosis induced by GPX4 inhibition and xCT suppression. Whereas xCT suppression leads to a general phenomenon of ferroptosis propagation in different cell lines, GPX4 inhibition did not lead to ferroptosis propagation in two out of the three cell lines we tested. As our findings may potentially inspire future studies to develop therapies harnessing ferroptosis propagation and to identify *in vivo* triggers for ferroptosis, it is indeed crucial to make it clear that ferroptosis propagation is only a general feature of the ferroptosis induced by xCT suppression, but not by GPX4 inhibition. We have now incorporated new results showing the absence of ferroptosis propagation in the two tested cell lines (Extended Data Fig. 1k), and highlight this point in page 3, lines 71-75.

P3. In Supplemental Video 6 it is claimed that there is “ferroptosis deceleration” upon the addition of DFO. DFO may take time to enter the cell (via fluid phase endocytosis) so this might be consistent, but it is a rather inelegant experiment for that reason. This is just a note; there is no need to do anything on this point.

We thank the reviewer for sharing this insight. It is apparent in Supplementary Video 6 that it took ~ 1 hour after DFO addition before the speed of ferroptosis trigger waves decreased. The delay in the effect of DFO on the trigger wave speed is consistent with the time-scale of DFO uptake via fluid phase endocytosis.

Minor points:

P1. In the response to Reviewer #2, the authors state that “erastin is an irreversible inhibitor, it is currently not possible to reduce the erastin dose after applying a high erastin dose (e.g., by washing out erastin).” This is not true. Erastin is a reversible (non-covalent) inhibitor and can be washed out of cells. The problem is that cells exposed to erastin do not necessarily live very long and if one wants to perform washout experiments they have to be done carefully. But, technically, this is possible. I leave it to others to determine whether this is ultimately an important note or not.

We thank the reviewer for raising this point. Our statement that erastin is an irreversible inhibitor was based on a previous study highlighting the irreversibility of xCT inhibition by erastin ¹. That study showed that transient exposure of cells to erastin (5 minutes), followed by washing-out, could lead to prolonged (24 hours) suppression of cystine uptake, albeit the mechanism remains elusive. We agree with the reviewer that since erastin is a non-covalent inhibitor, it is technically possible to wash-out erastin. Nevertheless, the proposed experiment to address the previous comment by reviewer 2 requires complete and immediate reversal of xCT inhibition. Considering the observations made in that previous study ¹, it would be challenging to guarantee complete reversal of xCT inhibition and obtain conclusive results.

P2. Line 29 of the Abstract. Is “media” the correct word here? It does not seem like the right word. I see it used again in Line 57 and 59, which suggests perhaps that this is a term of art in the trigger wave field. Perhaps this is the only word to use. But, it is also possible that the manuscript would benefit from a careful definition of what this means since, to many cancer cell biologists, the “media” is simply the fluid that a cultured cell grows in, not something that is “excitable and bistable”. Indeed, the Author’s use media in this latter sense themselves in Line 96 (where grammatically this should perhaps be ‘medium’, the singular, not ‘media’, the plural).

The term “bistable media” is widely used in various research areas in physics, particularly in studying travelling waves ^{2,3} and pattern formation ^{4,5}. Here, “media” refers to ensembles of spatially-coupled entities over which signals can propagate as waves. In the context of ferroptosis trigger waves, the bistable “media” are “cell populations”, which comprise spatially-coupled bistable systems, i.e., individual cells, over which ferroptosis propagates. To avoid confusion, we have made the following textual changes:

1. In line 29 of the revised Abstract, we stated that the bistable media is the “media over which ROS propagates”.
2. In line 54, we stated that the bistable media is the “media over which signals are regenerated and transmitted”.
3. In line 56, we replaced bistable “media” with bistable “cell populations” in the context of cell death propagation.
4. In lines 255-256, we state that the bistable medium is the medium “over which ROS can propagate as trigger waves”.
5. In line 390-391, we have replaced “media” with “cell populations”.

In line 94 where “media” refers to the culture media in which cells are grown, we have now changed “media” to “medium”.

Referee #1 (Remarks on code availability):

I am not a software person. I do not know how to evaluate code.

Referee #2 (Remarks to the Author):

The authors have strengthened this manuscript substantially. In particular I found the new data on muscle sculpting in Fig 5 to be stronger than what was in the original manuscript, although, admittedly, my knowledge of limb developmental biology is pretty minimal. In both the paper and the rebuttal, the authors more emphatically and convincingly make the case that something other than apoptosis contributes to programmed cell death during development, and that in limb development ferroptosis is important.

I have some minor criticisms of the early parts of the paper, which I delineate below, but overall I think this is important work. It provides convincing cell culture evidence that ferroptosis can spread from cell-to-cell via a diffusible factor that is released from a one ferroptotic cell and then gets amplified up by a not-yet ferroptotic target cell, leading to constant speed, constant amplitude trigger waves of ferroptosis propagating through the culture. And it puts this cell culture work into a physiological context via the evidence for ROS and ferroptotic waves in limb development.

We are grateful for the reviewer's positive assessment about our work's quality and impact. To address the reviewer's concerns, we have made textual changes and incorporated a graphical illustration to enhance readability, as detailed below.

Specific minor criticisms:

P1. I do not think readers are going to be able to understand the data presented in Extended Data Figure 1. The authors need to explain what the vector fields, polar distribution plots, and entropies mean, and explain what differences in these measures would be expected if you have ferroptosis spreading from a discrete source versus ferroptosis occurring independently in each cell.

We thank the reviewer for this suggestion. To help readers understand the data presented in Extended Data Figure 1, we have made a graphical illustration on how different death kinetics impact the vector fields, polar distribution plots and entropies (Extended Data Fig. 1c). We have also incorporated a brief description of the vector field, polar distribution plots and entropies in the figure legend. Due to the word limit for the Main Text, detailed descriptions of these data have been placed in the Methods.

P2. Lines 152-161. I agree that the diffusible mediator of ferroptosis is almost certainly a free radical other than hydrogen peroxide (Extended Figure 8), but I do not find the speculation that it/they is/are a lipid peroxide compelling. It could be a membrane-bound lipid peroxide, but it could also be a protein or anything in or on an organelle. Perhaps just lay out the various possibilities.

We thank the reviewer for the suggestion. In addition to lipid peroxides, we also mentioned the possibility of lipid peroxide byproducts being the potential diffusive molecule in our original manuscript. Extracellular vesicles and/or their cargoes have recently been found to spread between cells during erastin treatment⁶. Proteins that generate ROS (e.g., NADPH oxidase)⁷ can also be released from ferroptotic cells to mediate the spread of cell death. We now state this possibility regarding other signaling molecules in the revised manuscript (page 7, lines 158-160).

P3. Figure 3: Do MEK or ERK inhibitors block the increase in wave speed (or perhaps block the production of ferroptotic waves altogether) seen after ERK overexpression? I don't insist on this experiment, but if you have the data it would be nice to mention it.

We currently do not have this data. We thank the reviewer for raising this question, which can be interesting to explore in the future.

P4. Line 244: Please clarify: do the authors mean that their model predicts that below the critical concentration of 0.63 μM , increasing the erastin concentration exerts a relatively mild effect on the ROS level of the low-ROS steady state? Or that above the critical concentration of 0.63 μM there is little effect of the erastin concentration on the high-ROS steady state? Either way, I would think it's not just a prediction (Fig 4a), but an experimental observation (Fig 4b).

We apologize for the confusion. Our model predicts that when the erastin concentration is increased from 0 to 10 μM , the low ROS steady state is only mildly affected. Unlike this mild effect on the low ROS steady state, increasing the erastin concentration accelerated the wave speed. We think that this is an interesting feature of the system, i.e., the change in the ROS system with respect to increasing levels of erastin is not very apparent unless ROS perturbations (e.g., photoinduction) are introduced to probe for the existence of an alternative steady state (i.e., high steady state) that allows trigger wave occurrence. This scenario represents a testable prediction derived from our model (Fig. 4a) that was experimentally validated (Fig. 4b). To avoid confusion, we have now clarified this point in page 11, lines 236-239.

Referee #3 (Remarks to the Author):

This revised manuscript, "Emergence of large-scale cell death via trigger waves of ferroptosis" has been substantially improved in revision. This reviewer was asked to comment on limb development portion, but unlike the previous version, I was able to follow the rationale, data, and conclusions through the first four figures as well. Will leave to other experts to make appropriate scientific comments, but readability of manuscript is excellent.

The additional experiments and data in Figure 5 (and associated Suppl Figs 18-20) provide excellent proof of principle in vivo in a way that will be appreciated by people in the field.

We thank the reviewer for the positive remarks and questions that have helped us to enhance the depth and rigor of our work.

One remaining comment, which is meant only to reflect this reviewer's curiosity (and perhaps others from this field), if there is room in the discussion/end of manuscript, how do the authors think about both apoptosis and ferroptosis occurring in these biologically relevant areas? It is clear from their data this is occurring, but why both and what could distinguish cells that are presumably going through the same developmentally regulated process (cell death) via both of these distinct mechanisms?

We thank the reviewer for this inspiring question. A number of recent studies have shown the occurrence of both apoptosis and ferroptosis in various pathological contexts, such as during acute kidney injury⁸, intracerebral hemorrhage⁹, and age-related macular degeneration¹⁰. In this study, we show that both apoptosis and ferroptosis occur for muscle mass remodeling during limb development. However, how and why both apoptosis and ferroptosis pathways occur in these contexts remain elusive.

There are multiple possible scenarios explaining how and why both apoptosis and ferroptosis occur during development. One possibility is that both apoptosis and ferroptosis are activated by the same developmental signal, and whether cells undergo apoptosis or ferroptosis depends on which pathway is activated predominantly or more rapidly. Another possibility is that the ferroptosis and apoptosis pathways buffer each other, such that when one cell death pathway fails to be activated, the other cell death pathway is activated as a compensatory mechanism. Both above-mentioned mechanisms would enable robust control of cell death by the developmental signal.

In terms of what could distinguish cells undergoing both apoptosis and ferroptosis from cells undergoing conventional programmed cell death (apoptosis), this is an intriguing question. It is tempting to speculate that different developmental signals regulating cell death might determine whether both cell death pathways or a single pathway is activated. For example, developmental signals that rely on ROS to induce cell death might activate both the apoptosis and ferroptosis pathways.

These are some exciting questions worth exploring in the near future. Although we are enthusiastic about discussing these points in the manuscript, we have been unable to incorporate them into the revised manuscript due to space constraints. Nevertheless, we hope that our study will prompt future studies on this subject.

References

- 1 Sato, M. *et al.* The ferroptosis inducer erastin irreversibly inhibits system x(c)- and synergizes with cisplatin to increase cisplatin's cytotoxicity in cancer cells. *Sci Rep* **8**, 968, doi:10.1038/s41598-018-19213-4 (2018).
- 2 Tyson, J. J. & Keener, J. P. Singular Perturbation-Theory of Traveling Waves in Excitable Media. *Physica D* **32**, 327-361, doi:Doi 10.1016/0167-2789(88)90062-0 (1988).
- 3 Yde, P., Mengel, B., Jensen, M. H., Krishna, S. & Trusina, A. Modeling the NF-kappaB mediated inflammatory response predicts cytokine waves in tissue. *BMC Syst Biol* **5**, 115, doi:10.1186/1752-0509-5-115 (2011).
- 4 Meron, E. Pattern-Formation in Excitable Media. *Phys Rep* **218**, 1-66, doi:Doi 10.1016/0370-1573(92)90098-K (1992).
- 5 Sakurai, T., Mihaliuk, E., Chirila, F. & Showalter, K. Design and control of wave propagation patterns in excitable media. *Science* **296**, 2009-2012, doi:10.1126/science.1071265 (2002).
- 6 Wang, L. L., Mai, Y. Z., Zheng, M. H., Yan, G. H. & Jin, J. Y. A single fluorescent probe to examine the dynamics of mitochondria-lysosome interplay and extracellular vesicle role in ferroptosis. *Dev Cell* **59**, 517-528 e513, doi:10.1016/j.devcel.2024.01.003 (2024).
- 7 Block, K. & Gorin, Y. Aiding and abetting roles of NOX oxidases in cellular transformation. *Nat Rev Cancer* **12**, 627-637, doi:10.1038/nrc3339 (2012).
- 8 Hu, Z. *et al.* VDR activation attenuate cisplatin induced AKI by inhibiting ferroptosis. *Cell Death Dis* **11**, 73, doi:10.1038/s41419-020-2256-z (2020).
- 9 Li, Q. *et al.* Inhibition of neuronal ferroptosis protects hemorrhagic brain. *JCI Insight* **2**, e90777, doi:10.1172/jci.insight.90777 (2017).
- 10 Chen, C., Chen, J., Wang, Y., Liu, Z. & Wu, Y. Ferroptosis drives photoreceptor degeneration in mice with defects in all-trans-retinal clearance. *J Biol Chem* **296**, 100187, doi:10.1074/jbc.RA120.015779 (2021).